# Bone morphogenetic protein (BMP) signaling determines neuroblastoma cell fate and sensitivity to retinoic acid

Min Pan [1,10] ✉, Yinwen Zhang[1,10], William C. Wright [1], Xueying Liu [1], Barbara Passaia [1], Duane Currier [2], Jonathan Low[2], Richard H. Chapple [1], Jacob A. Steele [3,4], Jon P. Connelly [3,4], Bensheng Ju[1], Emily Plyler[1], Meifen Lu[5], Allister J. Loughran [3,4], Lei Yang [2], Brian J. Abraham [1], Shondra M. Pruett-Miller [3,4], Burgess Freeman III [6], George E. Campbell [7], Michael A. Dyer [8,9], Taosheng Chen [2], Elizabeth Stewart [7], Selene Koo [5], Heather Sheppard [5], John Easton [1] ✉ & Paul Geeleher [1] ✉

Retinoic acid (RA) is a standard-of-care neuroblastoma drug thought to be effective by inducing differentiation. Curiously, RA has little effect on primary human tumors during upfront treatment but can eliminate neuroblastoma cells from the bone marrow during post-chemo maintenance therapy—a discrepancy that has never been explained. To investigate this, we treat a large cohort of neuroblastoma cell lines with RA and observe that the most RA-sensitive cells predominantly undergo apoptosis or senescence, rather than differentiation. We conduct genome-wide CRISPR knockout screens under RA treatment, which identify bone morphogenic protein (BMP) signaling as controlling the apoptosis/senescence vs differentiation cell fate decision and determining RA's overall potency. We then discover that BMP signaling activity is markedly higher in neuroblastoma patient samples at bone marrow metastatic sites, providing a plausible explanation for RA's ability to clear neuroblastoma cells specifically from the bone marrow, by seemingly mimicking interactions between BMP and RA during normal development.

Retinoic acid (RA) is a metabolite of vitamin $A_1$ (retinol) that regulates several normal physiological processes, including cell differentiation and development[1,2]. RA's potential as an anti-cancer drug was first described in the 1980s when it was shown to have therapeutic activity in acute promyelocytic leukemia (APL)[3], a hematologic cancer characterized by fusion of the retinoic acid receptor alpha (*RARA*) and promyelocytic leukemia (*PML*) genes[4,5]. This fusion causes a

pathogenic block in myeloid cell differentiation that can be overcome with pharmacological concentrations of RA[6]. RA was shown to induce differentiation and maturation of APL cells[7], ultimately leading to their elimination and providing the first curative therapy for APL[8].

The unprecedented success of RA in APL generated interest in other cancers, including neuroblastoma, the most common pediatric solid tumor, where only ~50% of high-risk patients survive. Initial

[1]Department of Computational Biology, St. Jude Children's Research Hospital, Memphis, TN 38105, USA. [2]Department of Chemical Biology and Therapeutics, St. Jude Children's Research Hospital, Memphis, TN 38105, USA. [3]Department of Cell and Molecular Biology, St. Jude Children's Research Hospital, Memphis, TN 38105, USA. [4]Center for Advanced Genome Engineering, St. Jude Children's Research Hospital, Memphis, TN 38105, USA. [5]Department of Pathology, St. Jude Children's Research Hospital, Memphis, TN 38105, USA. [6]Preclinical Pharmacokinetic Shared Resource, St. Jude Children's Research Hospital, Memphis, TN 38105, USA. [7]Cellular Imaging Shared Resource, St. Jude Children's Research Hospital, Memphis, TN 38105, USA. [8]Department of Developmental Neurobiology, St. Jude Children's Research Hospital, Memphis, TN 38105, USA. [9]Howard Hughes Medical Institute, Chevy Chase, MD 20815, USA. [10]These authors contributed equally: Min Pan, Yinwen Zhang. ✉e-mail: min.pan@stjude.org; john.easton@stjude.org; paul.geeleher@stjude.org

studies in cell lines[9,10] and animal models[11] demonstrated that RA could also induce differentiation of neuroblastoma cells—suggesting a mechanism similar to that in APL. Subsequent clinical trials assessing the efficacy of RA in neuroblastoma reported striking improvements in event-free survival among high-risk neuroblastoma patients in post-chemotherapy maintenance therapy: to 46 ± 6% for patients receiving 13-cis-retinoic acid, compared to 29 ± 5% for no further therapy[12].

Although RA is now established as standard-of-care for neuro-blastoma, its precise mechanism of action has not been proven experimentally. Surprisingly, unlike in APL, *RARA* fusions have never been identified in neuroblastoma[13], and while RA can certainly induce differentiation in neuroblastoma cell lines in vitro[10], its use in the maintenance therapy setting (typically minimal residual disease) makes it difficult to prove that differentiation is the primary mechanism of its activity in patients. Indeed, several early clinical studies showed that disseminated neuroblastoma cells could be eliminated from bone marrow metastatic sites following RA treatment, possibly due to clearance of those cells through some mechanism other than terminal differentiation[11,14]. Indeed, more recent work has also indicated that RA-induced differentiation in neuroblastoma cells is likely reversible[15]. Additionally, while clinical trials have shown that RA significantly improves outcomes as maintenance therapy, it has little upfront activity against established tumors in patients and mouse models[16]—behavior that is not typical of a cancer drug and has never been explained[12]. Interestingly, previous clinical trials are also suggestive that RA may potentiate the activity of anti-GD2 immunotherapy[17], but this sequential combination also predominantly clears cells from the bone marrow[18] and the reason for this unusual site-specific activity remains unknown.

Here, we exploited several recent technological advances to tease apart the mechanisms underpinning RA's diverse and site-specific effects in neuroblastoma. Specifically, we performed genome-wide CRISPR modifier screens in RA-treated hyper-sensitive neuroblastoma cell lines identified from recent large-scale drug screening datasets[19]. Our results indicate that RA's strongest anti-neoplastic effects in neuroblastoma arise from either apoptosis or senescence, rather than differentiation, activities we show are mediated by bone morphogenetic protein (BMP) signaling, a process highly active at bone marrow metastatic sites in neuroblastoma patients. These results provide new insight into the molecular basis of RA's therapeutic efficacy in neuroblastoma and have implications for the development of novel combination therapies, potentially including immunotherapies. Notably, the interplay between RA and BMP signaling is well-documented in developmental biology, where the coordinated activities of these pathways regulate cell fate decisions, including differentiation, apoptosis, and senescence[20–24]. Such pharmacological "hijacking" of a normal organismal developmental process has not, to our knowledge, previously been described as an effective anti-cancer mechanism.

## Results

### BMP signaling is required for RA response in a hyper-sensitive NB cell line

All-trans retinoic acid (ATRA; the most common biologically active form of RA) was recently screened across 923 cancer cell lines in the Genomics of Drug Sensitivity in Cancer (GDSC1) study[19]. Curiously, a small number of the pediatric cancer cell lines in this dataset responded dramatically to RA treatment within the short 3-day exposure, exhibiting extreme outlier $IC_{50}$ values compared to other lines of the same lineage (Fig. 1A). For example, the $IC_{50}$ of the RA hyper-sensitive neuroblastoma line NB13 was 0.26 μM, more than 100-fold lower than the median $IC_{50}$ (35.2 μM) for all NB cell lines.

We rescreened 16 cell lines (12 neuroblastoma cell lines and four cell lines from other cancers) with ATRA for 6 days (Fig. S1A, Supplementary Data 1), a more clinically relevant treatment duration given the long-lived pharmacokinetic profile of RA in vivo[25]. We found that some cell lines exceeded even NB13 in sensitivity with this longer

treatment; these lines included the neuroblastoma cell line CHP-134 and the medulloblastoma cell line D283 Med. Since no differentiation was observed in 3 days (Fig. S1B, C), we hypothesized that apoptosis, rather than differentiation, might account for these extreme RA responses. Indeed, ATRA induced substantial apoptosis, as shown by Annexin V+ cells and cleaved PARP expression (Fig. 1B, C, Fig. S1D), in NB13, CHP-134, and D283 Med cells after only 3 days of treatment. Thus, in the most ATRA-responsive cell lines, apoptosis may be the dominant cell fate decision.

While some evidence that RA can induce apoptosis in neuro-blastoma has been reported previously[26], it is not known what tips the scale between a cell fate choice of differentiation vs. apoptosis. In search of the determinants of the ATRA-induced apoptotic response, we performed a genome-wide CRISPR knockout screen using the hyper-sensitive neuroblastoma cell line, CHP-134, comparing enriched and depleted gene knockouts in 6-day ATRA-treated cells vs. mock-treated controls (Fig. S1E). As expected, the knockout of retinoic acid receptor *RARA* was the strongest ATRA resistance hit (Fig. 1D, E, Supplementary Data 2; Fig. S1F; hits ranked by RRA scores from MAGeCK[27], see Methods), supporting the validity of the CRISPR screen. Notably, however, functional analysis of the top hits (resistance and sensitization) showed that the top enriched pathways were surprisingly related to bone morphogenetic protein (BMP) signaling (Fig. 1D; Fig. S1G). Cells with knockout of BMP type I and type II receptors (*ACVR1*, *BMPR1A*, and *BMPR2*) and the targets of these receptors (*SMAD4* and *SMAD9*) were all significantly enriched in cells surviving RA treatment (Fig. 1E, left panel), with the estimated magnitude of *ACVR1's* effect similar to that of *RARA* knockout (2.2 vs 1.5 log₂ fold change for *ACVR1* vs *RARA* respectively). Consistent with this, knockout of BMP suppressors sensitized cells to ATRA (Fig. 1E, right panel). This included *FKBP1A*, a broad-spectrum type I BMP receptor repressor that can bind directly to ACVR1, inhibiting its activity[28]; *SMAD6*, an inhibitory SMAD that prevents phosphorylation and activation of regulatory SMADs[29]; and *SMURF1* and *SMURF2*, negative regulators of BMP signaling that promote ubiquitination and proteolysis of SMADs[30]. We repeated this CRISPR screen with CHP-134 cells treated for 3 days with ATRA and obtained similar results (Fig. S1H), with *FKBP1A* knockout remaining the most potent sensitizer and *ACVR1* knockout causing resistance comparable to *RARA* knockout.

To confirm the high throughput CRISPR screen, we knocked down the top resistance hit, BMP type I receptor *ACVR1*, with three independent shRNAs, and consistent with the screen, CHP-134 became approximately 10-fold more resistant to ATRA (Fig. S1I, J). Due to functional redundancy, silencing a single gene is usually not sufficient to fully inactivate BMP signaling. To inhibit BMP signaling more efficiently, we treated CHP-134 cells with ATRA combined with the selective BMP inhibitor K02288, an agonist of type I BMP receptors[31]. After 3 days of treatment with K02288, BMP signaling activity became undetectable as estimated by phosphorylation levels of SMAD1/5/9 (Fig. S1K), though this did not affect CHP-134 viability (Fig. S1L). However, the combination with K02288 markedly diminished CHP-134 sensitivity to ATRA, causing a 100-fold decrease in $IC_{50}$ at 6 days (Fig. 1F). K02288 completely blocked ATRA-induced cleaved PARP (Fig. 1G) and significantly decreased the percentage of Annexin V+ cells (Fig. 1H), indicating that blocking BMP signaling severely blunted the apoptotic response and sensitivity to ATRA. Thus, BMP signaling is necessary for RA-induced apoptosis in CHP-134 cells.

### BMP signaling potentiates RA response in a broad range of neuroblastoma cell lines

To test whether the effect of BMP signaling activity on ATRA response is generalizable beyond CHP-134 (and therefore potentially clinically relevant), we next examined the RNA-seq data available from GDSC and DepMap for the 16 cell lines we had screened with 6-day single-agent ATRA (Fig. S1A, Supplementary Data 1). We used these data to assess

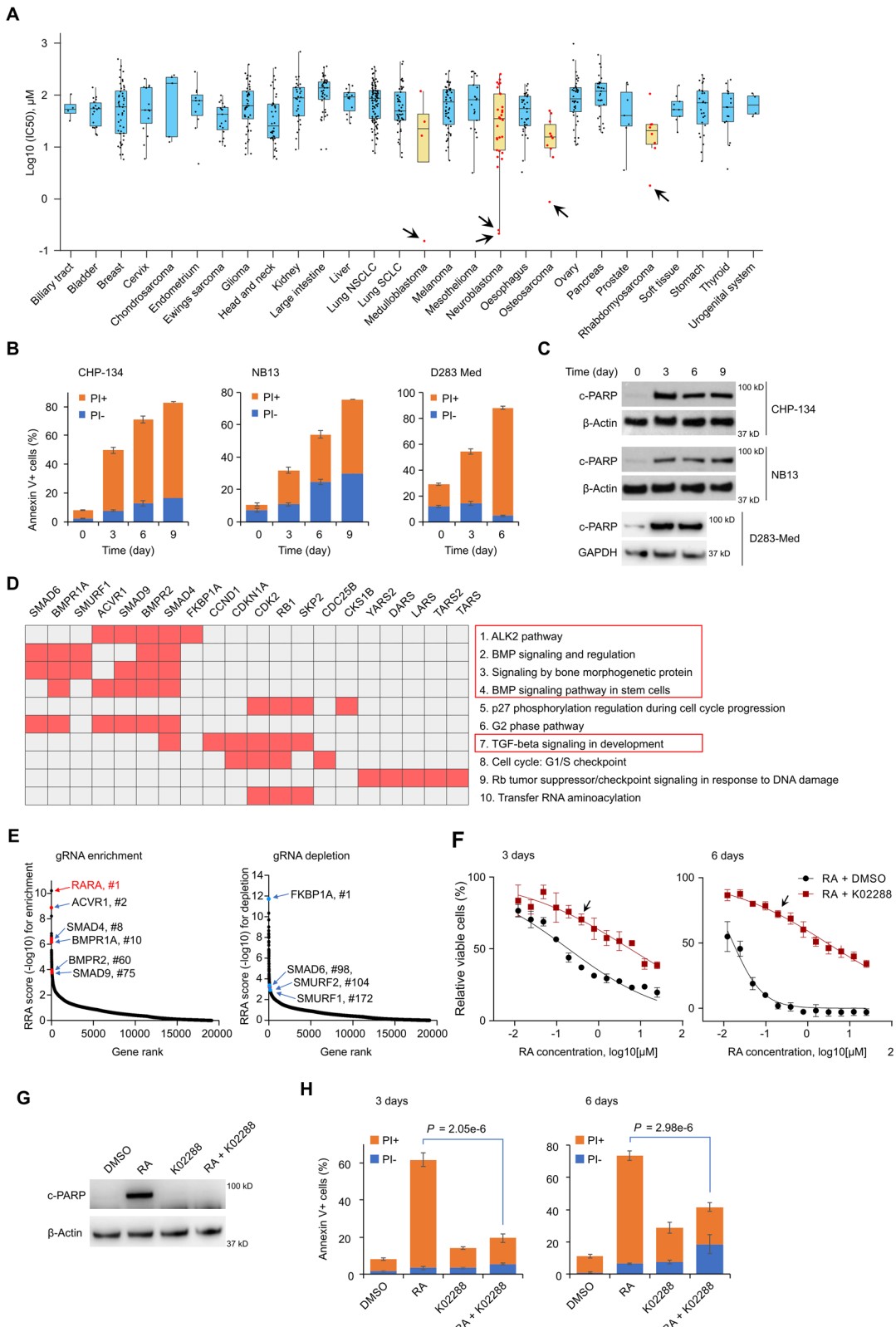

the correlations between ATRA response and the expression of all genes. Remarkably, the gene most correlated with ATRA $IC_{50}$ in GDSC was *SMAD9* (Fig. 2A). *SMAD9* is a regulatory SMAD and transcription factor that acts as one of the main downstream effectors of BMP signaling, which is highly expressed in neuroblastoma (Fig. S2A). Indeed, the ATRA $IC_{50}$s were almost perfectly correlated with *SMAD9* expression levels in both the GDSC ($R = -0.92$, $P = 4.2 \times 10^{-6}$; Fig. 2B) and

DepMap ($R = -0.81$, $P = 4.8 \times 10^{-4}$; Fig. 2A, B), with *SMAD9* expression also prognostic in neuroblastoma patients (Fig. S2B, C). This correlation was not evident for the other canonical BMP R-SMADs, *SMAD1* and *SMAD5* (Fig. 2A, Fig. S2D, E). To test whether *SMAD9's* association with ATRA response is partially causal, we knocked out *SMAD9* in two hypersensitive cell lines (CHP-134 and TGW; Fig. S2F). In both cell lines, *SMAD9* knockout diminished the response to ATRA treatment (Fig. S2G,

**Fig. 1 | BMP signaling is required for RA sensitivity in a hyper-sensitive NB cell line. A** ATRA IC$_{50}$ values for all non-hematological cancer cell lines screened by GDSC1. Each dot represents a cell line (*n* = 759 in total). Box bounds show 25th and 75th percentiles, horizontal lines within the interquartile range (IQR) boxes indicate the median values for each group, and whiskers represent 1.5*IQR from the IQR values. Cancer types with ≤ 3 cell lines are not included. **B** Quantified flow cytometry results showing apoptosis in CHP-134, NB13, and D283 Med cell lines treated with ATRA (2 μM). Apoptotic cells were stained positive with annexin V and dead cells were stained positive with PI. Samples at day 9 for CHP-134 *P* = 1.67 × 10$^{-7}$, NB13 *P* = 3.32 × 10$^{-6}$; samples at day 6 for D283 Med *P* = 1.35 × 10$^{-6}$. **C** Western blots for cleaved PARP (c-PARP) expression in CHP-134, NB13, and D283 Med cell lines treated with ATRA (2 μM). **D** Enrichr clustergram of CRISPR screen results (for BioPlanet 2019 pathways and GO Molecular Function 2021). Input: top 1% of genes with lowest negative and positive RRA score (191 + 191 genes). The genes and their directionality are listed in Supplementary Data 2. Red indicates screen hits.

**E** Scatter plot highlighting top CRISPR screen hits involved in BMP signaling, ranked by positive RRA score (left, top knockouts cause resistance to ATRA) or negative RRA score (right, top knockouts sensitize to ATRA). *RARA* is also highlighted. **F** Viability of CHP-134 cells under 3-day (left) or 6-day (right) treatment with ATRA in the presence of DMSO or 0.5 μM K02288. Cell viability was measured with MTS. Data represent the mean ± SD of 4 independent replicates. For arrow indicated points, *P* = 3.89 × 10$^{-6}$ for 3 days, *P* = 1.91 × 10$^{-8}$ for 6 days. **G.** Western blots for c-PARP expression in CHP-134 treated with indicated compounds for 3 days (2 μM ATRA and 0.5 μM K02288). **H** Quantified flow cytometry results showing apoptosis in CHP-134 treated as in (G) for 3 days (left) and 6 days (right). For panel B and H, data represent the mean ± SD of 3 independent replicates. For panels C and G, GAPDH and β-Actin were used as loading control, the representative results of three independent experiments are shown. Two-sided t-test was used for panels B, F, and H. Source data are provided as a Source Data file.

H), but *SMAD1* knockout had limited effect (Fig. S2I, J). This generalizable association between this key BMP signaling effector and ATRA IC$_{50}$ across a sizable cohort of cell lines suggests that the effect of BMP signaling activity on RA response is broadly relevant.

To further probe the generality of these relationships, we examined the data from a 655-gene targeted CRISPR knockout screen in 18 drug-treated cell lines[32], including 10 ATRA-treated neuroblastoma cell lines (all independent of CHP-134). Among those cell lines, knockout of the BMP pathway repressor *FKBP1A* (the top sensitizer in our CHP-134 genome-wide screen) was, on average, the 4th most potent sensitizer to ATRA in the neuroblastoma cell lines (Fig. 2C; ranked by MAGeCK RRA score[27]), further indicating that perturbation of BMP signaling broadly affects ATRA response. Interestingly, several small molecules can repress FKBP1A's activity as negative regulators of type I BMP receptors. These include FK506 (which tethers FKBP1A to calcineurin) and rapamycin (which tethers FKBP1A to mTOR). Thus, these molecules sequester FKBP1A away from type I BMP receptors (e.g., ACVR1), thereby enhancing BMP signaling activity[33]. We treated ATRA-sensitive neuroblastoma cell lines (CHP-134, NB-13, TGW, SK-N-SH) with FK506 or rapamycin for four hours, which in all cases resulted in increased phosphorylation of SMAD1/5/9, indicating an increase in BMP signaling activity (Fig. 2D). To investigate the effect of these small molecules on ATRA response at scale, we used high-throughput robotic handling to perform a dense large-scale drug combination screen in 10 neuroblastoma cell lines and 6 cell lines from other tissues, treating with the combination of ATRA plus FK506 or rapamycin. We generated a 10 × 10 matrix of drug concentrations for each combination (Fig. 2E), and calculated synergy as previously described[34] (Fig. 2F). ATRA showed significant synergistic effects with both compounds in all but three neuroblastoma cell line screens (NGP treated with ATRA + FK506 or rapamycin, SK-N-AS with ATRA + FK506, and SK-N-FI with ATRA + rapamycin)(Fig. 2G, Fig. S2K, L). The synergy was particularly pronounced in cell lines that have high baseline *SMAD9* expression and exhibit reasonable sensitivity to single-agent ATRA (e.g., CHP-134, TGW, SK-N-SH, and NB-13) (Fig. S2M, N, Supplementary Data 1), where concentrations at which neither drug alone had a marked effect on cell viability reduced cell viability by over 70% when used in combination (Fig. S2K, L). Little to no synergy was observed in non-neuroblastoma cell lines, except for the ATRA-sensitive D283 Med (Fig. 2G, Fig. S2O, P). The max synergy scores for RA + FK506 and RA+Rapamycin were significantly correlated across tested cell lines (Fig. 2H), suggesting that, while both FK506 and rapamycin affect other cellular processes, their shared common effect on amplifying BMP signaling contributed to the synergy with ATRA.

We performed MTS assays to confirm the results of our high-throughput screen in CHP-134 and TGW cells, where FK506 was again highly synergistic with nanomolar concentrations of ATRA (Fig. 2I, J). The synergistic effect of ATRA and FK506 was severely diminished in a knockdown of the BMP Type I receptor *ACVR1* (Fig. 2K), consistent

with our expectation that FK506 can amplify BMP signaling by sequestering FKBP1A away from type I BMP receptors. Collectively, these data suggest that the sensitivity of neuroblastoma cells to RA is broadly dependent on BMP signaling activity and it is possible to amplify this effect pharmacologically.

## High BMP signaling activity prevents cell differentiation while promoting apoptosis in cooperation with ATRA

We showed in Fig. 1 that the hyper-sensitive cell line CHP-134 underwent BMP-mediated ATRA-induced apoptosis. Given the apparent generalizability of BMP's effect on ATRA response (Fig. 2), we set out to determine whether BMP always functions to promote apoptosis and if/how these cells avoid differentiation. Indeed, testing the expression of mature neuron markers (MAP2 and NEFH) in CHP-134 and NB13 cells and measuring neurite outgrowth after 6 days of ATRA treatment revealed clear evidence of differentiation in the surviving cell (<10% for CHP-134 and ~20% for NB13; Fig. 3A, DMSO vs ATRA; Fig. S1B, C), but we did not observe these differentiated features in untreated cells. While this outcome seems to indicate that differentiation also contributes to the anti-neuroblastoma effect, it could also be consistent with differentiation acting as an apoptosis escape mechanism, which could diminish ATRA effectiveness overall and is thus worth further investigation.

We showed in Fig. 1F–H that the BMP inhibitor K02288 blocked ATRA-induced apoptosis, causing ATRA-resistance in CHP-134; thus, we wondered whether blocking BMP signaling would similarly block the differentiation response. Remarkably, however, rather than inhibit differentiation, single-agent K02288 markedly *induced* cell differentiation (Fig. 3A, B). The combination of ATRA and K02288 induced differentiation even more intensely than RA or K02288 alone, producing marked expression of MAP2 after 3 days of treatment (Fig. 3B) and more neurites after 6 days (Fig. 3A). These changes were accompanied by a large *increase* in the proportion of surviving cells (Fig. 1F; cell viability was 1.9% when treated with 0.2 μM ATRA for 6 days, but addition of K02288 increased this to 72.2%). Induction of differentiation by K02288 was also observed in another hypersensitive cell line NB13 (Fig. 3B, Fig. S3A), and similar to CHP-134, K02288 treatment substantially attenuated ATRA-induced apoptosis in NB13 (Fig. 3C, D). Another BMP inhibitor, noggin (NOG)[35], exhibited weaker but similar effects to K02288 (Fig. 3E, F). We observed similar behaviors in *SMAD9* knockout cells, which showed elevated expression of the differentiation marker MAP2 (Fig. S3B) and expressed much higher levels of this neuronal marker protein than control cells after ATRA treatment (Fig. 3G). In contrast to differentiation, apoptosis was diminished by *SMAD9* knockout (Fig. 3G). These observations suggest that differentiation is *inhibited*—not potentiated—by the high BMP activity in ATRA hyper-sensitive neuroblastoma cell lines.

This also suggests the synergy between ATRA and BMP activators could be broadly explainable by BMP signaling enhancing RA's ability to induce apoptosis. Indeed, the combination of ATRA and BMP activator

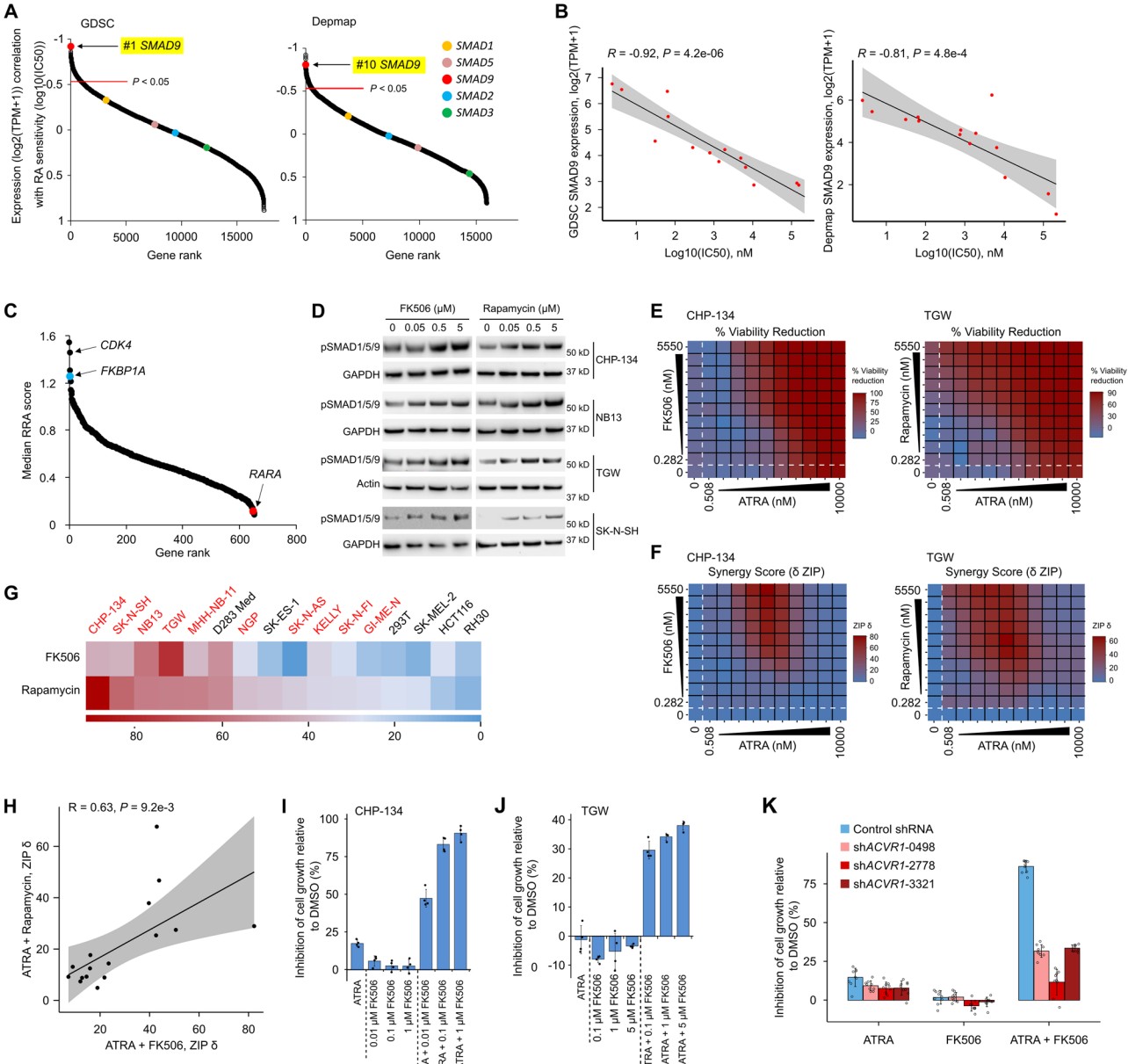

**Fig. 2 | BMP signaling potentiates RA response in a broad range of neuroblastoma cell lines. A** Pearson correlations of the expression of all genes with ATRA sensitivity in the cell lines we rescreened for 6 days. Gene expression data were from GDSC (left, $P = 4.2 \times 10^{-5}$ for *SMAD9*) and Depmap (right, $P = 4.8 \times 10^{-4}$ for *SMAD9*). *P*-values are from two-sided Pearson's correlation tests. **B** Scatter plot of *SMAD9* RNA expression from GDSC (left) and Depmap (right) plotted against ATRA 6-day $IC_{50}$ values. *P*-values were calculated with two-sided Pearson correlation tests. The shaded area represents the Pearson correlation standard error. **C** Scatter plot of median RRA score of the 655 druggable genes from targeted CRISPR screens in 10 neuroblastoma cell lines. **D** Western blots showing BMP signaling activity (level of phosphorylated SMAD1/5/9) in neuroblastoma cell lines after FK506 and rapamycin treatment. Cells were treated for 4 h. GAPDH and β-Actin were used as a loading control, and the representative results of three independent experiments are shown. **E** Heatmaps of percent inhibition of cell viability in CHP-134 cells (left) conferred by drug-only treatments (values within dotted lines) and combination treatments (all other values) with ATRA and FK506. Ten-point doses were used at indicated concentrations with 1:3-fold dilutions for each drug. Each matrix represents the average of three independent experiments. Data were normalized as a function of percent cell viability using inter-plate controls. ATRA and rapamycin combination in the TGW cell line was shown on the right side. **F** Heatmap matrices of synergy scores derived from respective cell

death values in (E). Synergy scores "ZIP δ" were calculated based on the zero interaction potency (ZIP) model, which can be interpreted similarly to BLISS synergy. **G** Heatmap showing the maximum ZIP δ synergy scores (color scale) of ATRA and FK506 combinations or ATRA and rapamycin combinations in 16 cell lines. Neuroblastoma cell lines are highlighted in red. **H** Correlation of max ZIP synergy scores of ATRA + FK506 (x-axis) and ATRA + rapamycin (y-axis) screens in the cell lines shown in (G). The *P*-value was calculated with two-sided Pearson correlation tests. The shaded area represents the Pearson correlation standard error. **I** Bar plot confirming synergy between ATRA and FK506 in CHP-134. Cells were treated for 6 days. Cell viability was measured with MTS assay and normalized to DMSO-treated samples, which were defined as 0% inhibition of cell growth. Data represent the mean ± SD of 4 independent replicates. Two-sided t-test for ATRA vs ATRA + 1 μM FK506, $P = 1.29 \times 10^{-7}$. **J** Like (I) but in the TGW cell line. Two-sided t-test for ATRA vs ATRA + 5 μM FK506, $P = 4.71 \times 10^{-6}$. **K** Barplot showing synergy between ATRA and FK506 in CHP-134 upon *ACVR1* knockdown. Cells were transduced with lentiviral control shRNA or 3 independent *ACVR1* shRNAs. Cells were treated with 0.05 μM ATRA and 0.1 μM FK506 for 6 days. Inhibition of cell growth was calculated as in (I). Data represent the mean ± SD of 4 independent replicates. In ATRA + FK506 group, two-sided test for control vs 0498, $P = 3.22 \times 10^{-15}$; control vs 2778, $P = 4.7 \times 10^{-14}$; control vs 3321, $P = 1.24 \times 10^{-16}$. Source data are provided as a Source Data file.

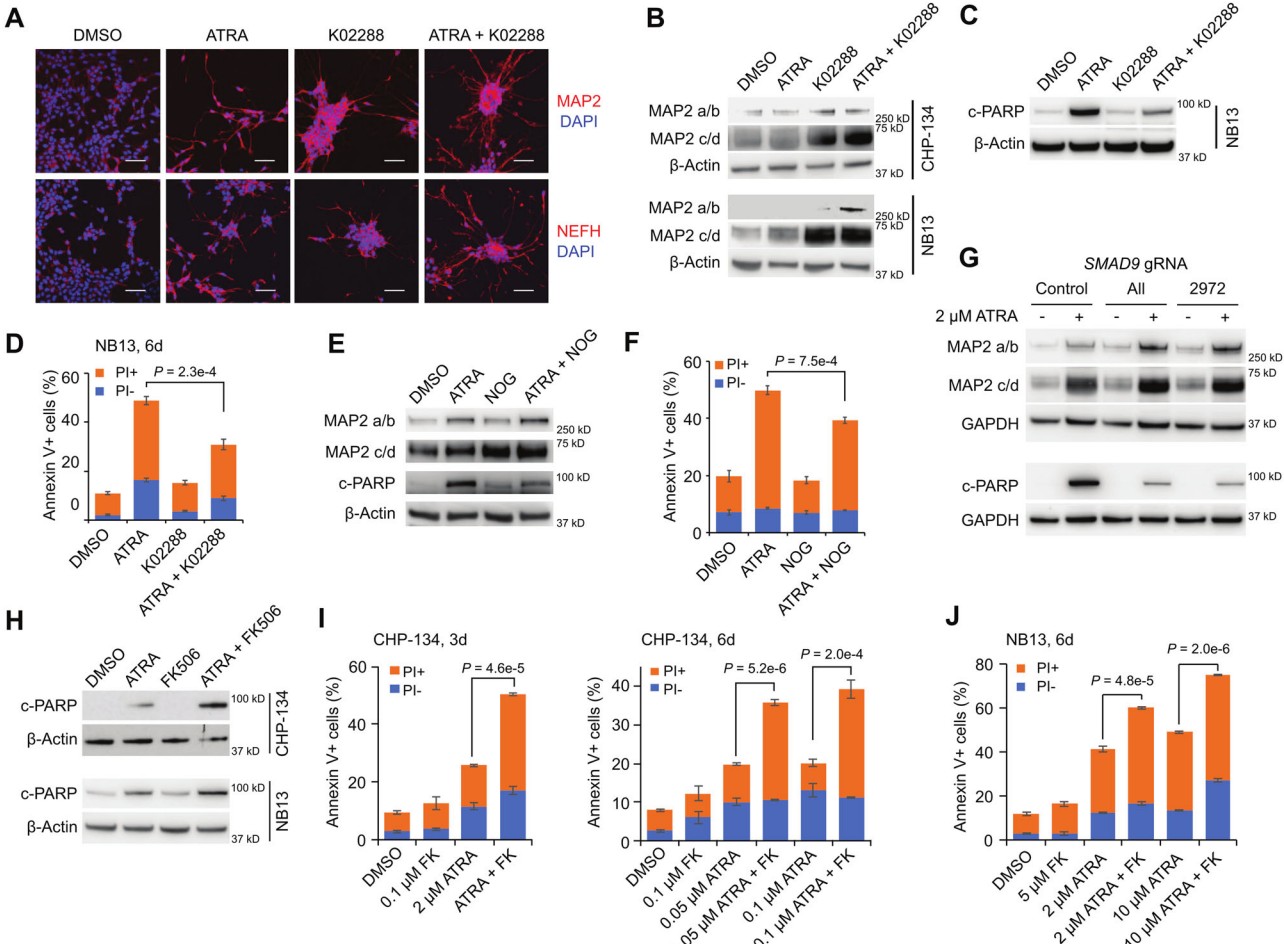

**Fig. 3 | BMP signaling prevents cell differentiation and promotes apoptosis in the presence of ATRA. A** Immunofluorescent staining for CHP-134 cells treated with indicated compounds (2 μM ATRA, 0.5 μM K02288) for 6 days. MAP2 and NEFH expression and neurite structures are shown in red. Nuclei were stained with DAPI (blue). Representative results of 3 independent replicates are shown. Scale bar = 50 μm. **B** MAP2 expression in CHP-134 and NB13 cells treated for 3 days (2 μM ATRA, 0.5 μM K02288). Four isoforms of MAP2 (labeled a/b and c/d) are shown. **C** c-PARP expression in NB13 treated for 3 days (2 μM ATRA and 0.1 μM K02288 for NB13). **D** Quantified flow cytometry results showing apoptosis in NB13 treated for 6 days (2 μM ATRA, 0.1 μM K02288). **E** MAP2 and c-PARP expression in CHP-134 treated for 6 days (2 μM ATRA, 3 μg/ml noggin (NOG)). All samples were derived from the same experiment and all antibodies were incubated with the same membrane. **F.** Quantified flow cytometry results showing apoptosis in CHP-134 treated the same as in (**E**). **G** MAP2 and c-PARP expression in CHP-134 with *SMAD9* CRISPR knockout (*SMAD9*

KO). Cells were treated with DMSO or 2 μM ATRA (6 days for MAP2 and 3 days for c-PARP). The samples for MAP2 proteins and c-PARP are from 6-day and 3-day treated cells, respectively. The 6-day treated samples were ran in one gel and the 3-day treated samples were ran in another gel. Control, negative control gRNA with no target in the human genome; All, a mixture of 3 independent gRNAs targeting *SMAD9*; 2972, a single gRNA targeting *SMAD9*. **H** c-PARP expression in CHP-134 and NB13 treated for 3 days (2 μM ATRA and 0.1 μM FK506 for CHP-134, 0.05 μM ATRA and 5 μM FK506 for NB13). **I** Quantified flow cytometry results showing apoptosis in CHP-134 treated for 3 days (left) and 6 days (right). FK, FK506. **J** Quantified flow cytometry results showing apoptosis in NB13 treated for 6 days. FK, FK506. For panels **B**, **C**, **E**, **G**, and **H**, β-Actin and GAPDH were used as loading control. Representative results of 3 independent replicates are shown. For panels **D**, **F**, **I**, and **J**, data represent the mean ± SD of 3 biologically independent replicates. *P*-values were calculated with two-tailed t-tests. Source data are provided as a Source Data file.

FK506 induced much more apoptotic cell death than ATRA alone in CHP-134 and NB13 cell lines in 3 days (Fig. 3H, J), while cell differentiation was not changed, or slightly decreased (Fig. S3C). Collectively, these results indicate that in the most sensitive neuroblastoma cell lines, the dominant anti-neoplastic effect is apoptosis and that low BMP signaling, leading to differentiation, results in a greater number of viable cells. These behaviors are consistent with the interactions between these pathways during normal development (see Discussion)[20,22].

## BMP signaling can also promote senescence, while preventing differentiation, in the presence of ATRA

Surprisingly, ATRA treatment did not induce apoptosis in every hyper-sensitive neuroblastoma cell line, including TGW and SK-N-SH (Fig. S3D). Curiously, however, the BMP inhibitor K02288 still caused these two cell lines to differentiate (Fig. 4A, B, DMSO vs. K02288, Fig. S3A). K02288 also caused TGW to become more resistant ATRA

(Fig. 4C), similar to the behavior in apoptotic cell lines. This suggests that ATRA can repress neuroblastoma cell viability through a third mechanism distinct from both apoptosis and differentiation, and that this activity is also mediated by BMP signaling.

To understand this further, we analyzed our targeted CRISPR screens in the 10 neuroblastoma cell lines (Fig. 2A), noticing the cell cycle regulator *CDK4* knockout was the #2 ranked sensitization hit. In addition, cell cycle-related pathways were also generally enriched in the CHP-134 genome-wide CRISPR screen (Fig. 1D, pathways 5, 6, 8, and 9). Thus, we speculated ATRA may sometimes induce cell cycle arrest. Given that differentiation was associated with more viable cells (Fig. 1F, Figs. 3B, 4B, C, ATRA + K02288 induced more differentiation and led to more viable cells than ATRA alone), we hypothesized these effects may be mediated by senescence[36,37]. We investigated senescence in TGW and SK-N-SH cells using the standard X-gal based β-Galactosidase (β-gal) activity assay (see Methods).

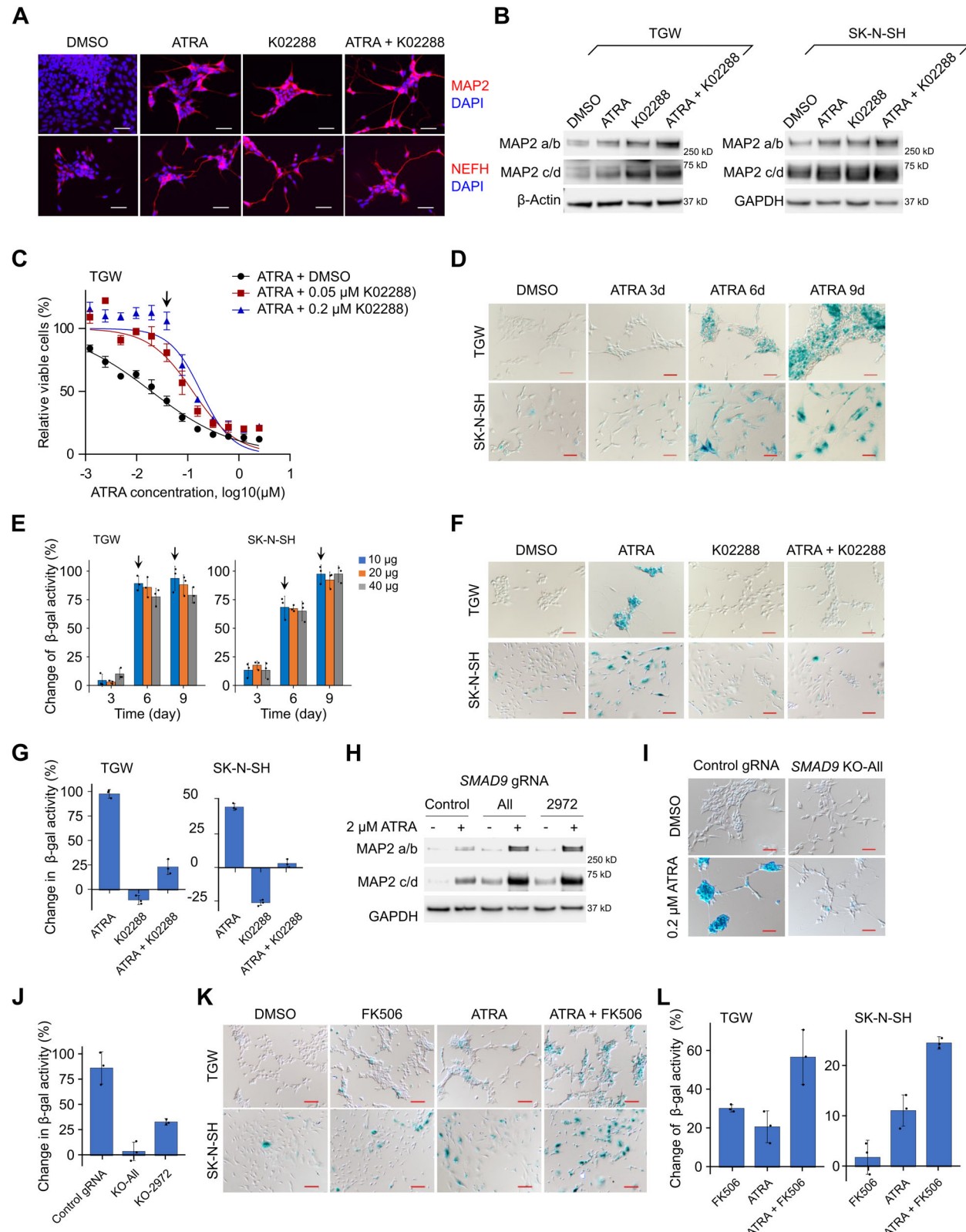

Strikingly, senescent cells accumulated after 6 days of ATRA treatment, and most of the cells became positive for β-gal activity after 9 days (Fig. 4D; senescent cells form blue crystals in this assay). We validated this result with another β-gal activity-based assay, using β-gal substrate 4-Methylumbelliferyl β-D-galactopyranoside (4-MUG) which can be hydrolyzed to a fluorescent product that can be quantified by fluorescence intensity. Consistent with the cell staining assay, cell lysates from ATRA-treated cells showed markedly increased fluorescent signal after 6 days (Fig. 4E). Importantly, we did not observe senescence in the apoptosis-dominant cell lines CHP-134 and NB13 after ATRA treatment (Fig. S3E).

Remarkably, when we blocked BMP signaling in these cells using K02288, ATRA could no longer induce senescence (Fig. 4F, G)—mirroring the effect of blocking BMP on apoptosis (Fig. 3). Similarly, *SMAD9*

**Fig. 4 | BMP signaling prevents cell differentiation and promotes senescence in the presence of ATRA. A** Immunofluorescent staining for TGW cells treated for 5 days (0.2 µM ATRA, 0.5 µM K02288). MAP2 and NEFH expression and neurite structures are shown in red. Nuclei were stained with DAPI (blue). Representative results of 3 biologically independent replicates are shown. Scale bar = 50 µm. **B** MAP2 expression in TGW and SK-N-SH cells treated for 6 days (0.02 µM ATRA and 0.5 µM K02288 for TGW, 5 µM ATRA and 0.5 µM K02288 for SK-N-SH). Four isoforms of MAP2 are shown. **C** Viability of TGW cells treated with ATRA for 6 days in the presence of DMSO or K02288. Cell viability was measured with MTS. Data represent the mean ± SD of 4 independent replicates. At the arrow indicated point, $P = 7.51 \times 10^{-5}$ for 0.05 µM K02288, $P = 3.99 \times 10^{-6}$ for 0.2 µM K02288. **D** Senescent cells form blue crystals in TGW and SK-N-SH cells treated with 0.2 µM and 2 µM ATRA, respectively. Scale bar = 50 µm. **E** Senescence levels quantified by intensity of fluorescence. TGW and SK-N-SH cells were treated for 0 (DMSO), 3, 6, and 9 days with 0.2 µM and 2 µM ATRA, respectively. Cell lysates containing 10, 20, and 40 µg protein from each time point were collected for β-gal activity analysis. P values for the arrow indicated samples are $2.11 \times 10^{-5}$, $1.18 \times 10^{-4}$, $4.38 \times 10^{-4}$, $6.03 \times 10^{-5}$ (from left to right). **F** Senescence in TGW and SK-N-SH treated for 6 days (0.2 µM ATRA and 0.5 µM K02288 for TGW, 5 µM ATRA and 0.5 µM K02288 for SK-N-SH). Scale bar = 100 µm. **G** Senescence levels quantified by intensity of fluorescence in TGW and SK-N-SH cell lines treated the same as in (**F**). P values for ATRA vs ATRA + K02288 are $1.32 \times 10^{-4}$ for TGW, $6.08 \times 10^{-5}$ for SK-N-SH. **H** MAP2 expression in TGW with *SMAD9* CRISPR knockout (*SMAD9* KO). Cells were treated with DMSO or 0.2 µM ATRA for 6 days. Control, negative control gRNA with no target in the human genome; All, a mixture of 3 independent gRNAs targeting SMAD9; 2972, a single gRNA targeting *SMAD9*. **I** Senescence in TGW with *SMAD9* KO. Cells were treated with 0.2 µM ATRA for 6 days. Scale bar = 50 µm. **J** Senescence levels quantified by intensity of fluorescence in TGW with *SMAD9* KO. Cells were treated with 0.2 µM ATRA for 6 days. 20 µg protein from each sample was used. P values are $1.46 \times 10^{-3}$ for KO-All, $4.75 \times 10^{-3}$ for KO-2972. **K** Senescence in TGW and SK-N-SH treated for 6 days (0.05 µM ATRA and 1 µM FK506 for TGW, 0.5 µM ATRA and 1 µM FK506 for SK-N-SH). Scale bar = 100 µm. **L** Senescence levels quantified by intensity of fluorescence in TGW and SK-N-SH cell lines treated the same as in (**K**). P values for ATRA vs ATRA + K02288 are 0.019 for TGW, $2.24 \times 10^{-3}$ for SK-N-SH. For panel **B** and **H**, β-Actin and GAPDH were used as loading control. Representative results of 3 independent replicates are shown. For panels **D**, **F**, **I**, and **K**, the representative results of 3 independent replicates are shown. For panels **E**, **G**, **J**, and **L**, data represent the mean ± SD of 3 independent replicates. *P*-values were calculated with two-tailed t-tests. Source data are provided as a Source Data file.

knockout also diminished senescence and potentiated cell differentiation, shown by increased MAP2 expression (Fig. 4H, Fig. S3B). After ATRA treatment, *SMAD9* knockout cells also showed lower β-gal activity than wild-type cells. (Fig. 4I-J). Additionally, lower concentrations of ATRA (0.05 µM for TGW and 0.5 µM for SK-N-SH), which induced only marginal senescence alone, could be markedly amplified when combined with FK506 (Fig. 4K-L). Intriguingly, programmed cell senescence also occurs naturally during tissue remodeling in embryonic development and can also be mediated by BMP[23,38,39]. Overall, these data suggest that BMP signaling regulates neuroblastoma cell differentiation and apoptosis/senescence in opposing ways when cells are exposed to retinoic acid, modifying these cell fate decisions and mirroring natural developmental processes.

## Downstream BMP and RA transcription factors cooperate to determine cell fate, following exposure to RA

We next wondered what determines these diverse BMP-mediated cell fate outcomes upon exposure to RA. We hypothesized the explanation may stem from the binding preferences and interactions between BMP and RA signaling transcription factors, the downstream effectors of these pathways. Thus, we evaluated the ChIP-seq binding profiles of SMAD4, SMAD9, and RARA proteins in baseline and ATRA-treated cells, profiling CHP-134, TGW, and BE(2)-M17 cell lines, representing the spectrum of dominant cell fate decisions (apoptosis, senescence, and differentiation[40] respectively; see Supplementary Data 3 for data summary; Fig. S4 for additional ChIP-seq summary/QC plots).

Interestingly, in untreated CHP-134 cells, we observed strongly overlapping binding profiles between all three transcription factors, with e.g., RARA overlapping 87% of SMAD9 peaks and 98% of SMAD9 peaks overlapping with SMAD4 peaks (Fig. 5A, B, Figs. S4F and S5A), indicating these TFs regulate similar target genes and processes. We evaluated the pathway enrichment for the genes closest to the binding loci using Gene Set Enrichment Analysis (GSEA). Remarkably, SMAD4 and SMAD9's strongest enrichment was in pathways related to apoptosis (such as regulation of apoptosis by FSH, T cell receptor regulation of apoptosis, and Interleukin-4 regulation of apoptosis) (Fig. 5C, Fig. S5B–D, Supplementary Data 4). Thus, the baseline binding profile of BMP signaling effectors SMAD4/9 is indicative of CHP-134's primary cell fate decision following ATRA exposure. Following 1 day of ATRA treatment, new RARA peaks were evident at a subset of baseline SMAD4 peaks, and apoptosis pathways were strongly enriched among these genes, suggesting RARA is directly regulating the acute activation of apoptosis (Fig. S5E). Over 6 days of exposure to 2 µM of ATRA, most CHP-134 cells die, but a small proportion (<10%) survive and

acquire differentiated neuron-like morphological features (Fig. 3A, Fig. S1C). Interestingly, the change in the binding pattern of these TFs is also reflected in this cell state transition, with SMAD9 gradually losing peaks associated with apoptosis-related genes (Fig. 5D), over 6 days of ATRA treatment (Fig. S5A, F). RARA and SMAD4 gain new peaks primarily located on genes with functions related to neuron development, which are evident after only 1 day of ATRA treatment, indicating the differentiation program is co-activated very quickly after ATRA treatment (Fig. S5G, H, Supplementary Data 4).

To determine whether these TF binding profiles were reflected in gene expression changes, we also performed RNA-seq analysis over the same treatment time course in CHP-134. The differentially expressed genes following ATRA (Fig. 5D) were significantly enriched for RARA and SMAD4 co-bound genes in untreated cells, with stronger enrichment observed in 1-day and 3-day ATRA treated cells compared to 6-day treatment (Fig. 5E, Supplementary Data 5). This is consistent with early expression changes resulting from the direct dysregulation of SMAD-bound genes by RARA. We investigated the differential expression patterns of genes co-bound by both RARA and SMAD4 in untreated cells, using the functional groups of apoptosis, cell cycle, and differentiation (see Methods). Consistent with the ChIP-seq profiles, apoptosis-related genes were primarily upregulated in the early time points (1-3 day ATRA) but were more likely to be repressed at 6 days, where strong induction of a differentiation program was evident in the cells that avoided apoptosis (Fig. 5F).

Notably, the expression of the 'Inhibitor of DNA binding' (*ID*) family of genes was almost completely lost in CHP-134 over the 6-day ATRA treatment (Fig. 5G; Fig. S5I), as was phosphorylation of SMAD1/5/9 (Fig. S5J). *ID* genes are the canonical downstream targets of BMP signaling, were among the strongest bound by SMAD4 and SMAD9 in untreated cells (Fig. S4E), and their mRNA levels can provide a reliable readout of BMP signaling activity[41,42]. *ID* genes' canonical function is to inhibit the activity of basic helix loop helix (bHLH) transcription factors[43] such as NEUROD1, a key transcriptional regulator of neuronal differentiation[44]. Interestingly, *NEUROD1* was the #1 ranked motif enrichment at RARA binding regions (1-day ATRA treatment; Fig. 5H, $P = 1.6 \times 10^{-210}$; Supplementary Data 6) and knockout of *NEUROD1* was the #8 ranked RA *sensitizer* in the CRISPR screen (Fig. S5K; $P = 2.6 \times 10^{-7}$). Indeed, overexpression of *ID2* potentiated ATRA-induced apoptosis and increased CHP-134 sensitivity to ATRA (Fig. 5I, J, Fig. S5L). This suggests that when highly active, BMP signaling simultaneously promotes acute RA-induced apoptosis and actively blocks RA-induced differentiation, but these activities are lost in the CHP-134 cells surviving ATRA treatment.

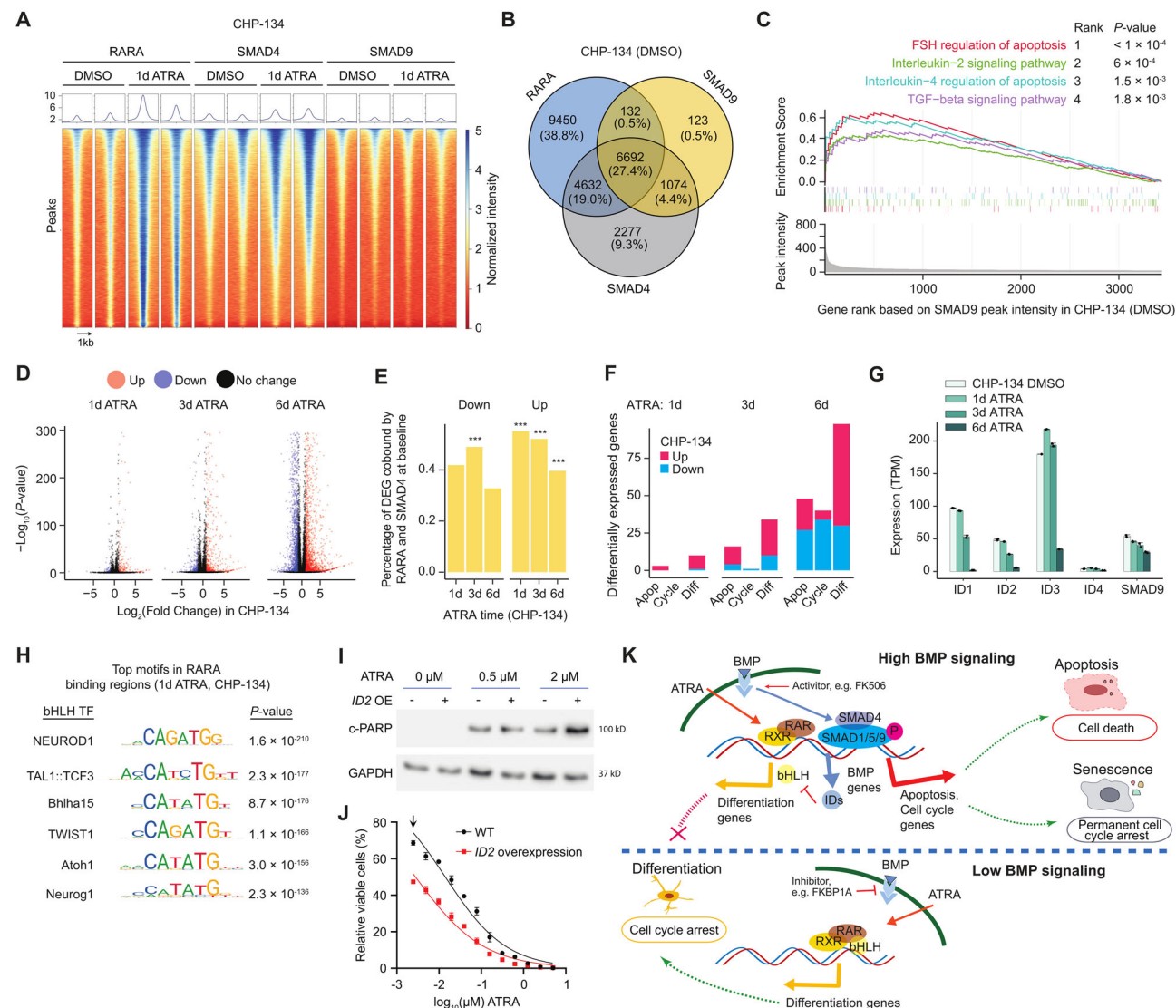

**Fig. 5 | Downstream BMP and RARA transcription factors cooperate to determine cell fate, following exposure to RA. A** Heatmap showing ChIP-seq peaks for RARA, SMAD4, and SMAD9 genome-wide in CHP-134 cells following treatment with DMSO control or 2 μM ATRA for 1 day. The color scale represents binding peak intensity, scaled between 0 and 5 (see Methods). Each genomic region (y-axis) was centered on the binding peak, displaying 1 kb upstream and downstream, sorted by average peak intensity across all transcription factors. Two biologically independent replicates are shown. **B** Venn diagram showing the number and proportion of overlapped binding regions between RARA, SMAD4, and SMAD9 in DMSO-treated CHP-134 cells. **C** GSEA plot showing the top 4 enriched pathways (defined in Bio-Planet 2019) of SMAD9 bound genes that gain RARA peaks following 2 μM ATRA treatment for 1 day in CHP-134 cells (see Methods). Running Enrichment Scores (colored lines). The vertical-colored lines indicate the members of each gene set, ranked by baseline SMAD9 binding intensity (y-axis, bottom panel). *P*-values were calculated by a one-sided empirical permutation test. **D** Volcano plot showing the differentially expressed genes (red or blue points) after 1, 3, or 6 days of 2 μM ATRA exposure in CHP-134 cells. Differential expression cutoffs: log₂ fold change > 1 or < -1 and *FDR* < 0.05. **E** Bar plot showing the percentage of up or down-regulated genes that are bound by SMAD4 and RARA at 1, 3, and 6 days (x-axis) following 2 μM ATRA exposure in CHP-134 cells. *P*-values indicate the enrichment of RARA and SMAD4

bound genes among the differentially expressed genes, calculated using a two-sided Fisher's exact test. *P*-values from left to right are $3.78 \times 10^{-8}$, $5.98 \times 10^{-8}$, $7.53 \times 10^{-16}$, and $1.60 \times 10^{-8}$. **F** Bar plot showing the number of up or down (blue) regulated genes (y-axis), bound by both RARA and SMAD4, following at 1-, 3-, and 6-days exposure 2 μM ATRA. Genes are grouped by their functional annotation (see Methods) in processes related to apoptosis (apop), cell cycle (cycle), and differentiation (diff). **G** Bar blot showing the expression (y-axis) of selected BMP signaling pathway genes (x-axis) in CHP-134 cells treated with ATRA for 1, 3, or 6 days (colors). Data represents the mean ± SD of two independent replicates. **H** Top-ranked consensus motif enrichments in the RARA binding regions of 1-day ATRA-treated CHP-134 cells, which are all basic helix-loop-helix (bHLH) factors. **I** Western blots (upper panel) of c-PARP expression in wild-type (−) and *ID2*-overexpressed (+) CHP-134 cells following 3 days of ATRA treatment at concentrations indicated. A representative result of three independent experiments is shown. GAPDH was used as a loading control. **J** Viability of wild-type (WT) and *ID2*-overexpressing CHP-134 cells after ATRA treatment for 6 days. Data represents the mean ± SD of three independent replicates. $P = 1.43 \times 10^{-5}$ at the arrow indicated point (two-sided t-test). **K** A conceptual model for how cells with high or low BMP signaling are directed into different fates upon RA exposure. \**P* < 0.05, \*\**P* < 0.01, \*\*\**P* < 0.001. Source data are provided as a Source Data file.

In the TGW cell line, which undergoes senescence during ATRA exposure (Fig. 4A, D), the baseline RARA binding patterns were sparser compared to CHP-134 (Fig. S6A). However, 89% of RARA peaks still overlapped SMAD4 in TGW (Fig. S6A, B), with co-binding genes notably enriched in cell cycle and differentiation-related genes

(Fig. S6C, Supplementary Data 4). After ATRA exposure, decreasing SMAD4 binding genes revealed a high enrichment in cell cycle pathways, including control of G1 to S and mitotic G2-G2/M phases, again aligning with the cell fate of TGW (Fig. S6D, E, Supplementary Data 4).

In the comparatively ATRA-resistant BE(2)-M17 cell line, which undergoes differentiation upon exposure to ATRA[45], RARA and SMAD4 baseline binding patterns still strongly overlapped, with 82% of RARA peaks overlapping with SMAD4, but they had fewer binding loci overall (Fig. S7A, B). The co-binding genes were weakly enriched in neuronal development processes but showed no significant enrichment in apoptosis or cell cycle (Fig. S7C, Supplementary Data 4). RARA and SMAD4 binding profiles showed moderate gains after 1 day of ATRA treatment (Fig. S7D), with SMAD4 peaks enriched in developmental genes (Fig. S7E). In data from the related cell line BE(2)-C, after 12 days of ATRA treatment, where BE(2)-C cells differentiate[15], the cells showed a significant increase in both the number and intensity of RARA binding sites (Fig. S7F). Thus, in comparatively ATRA-resistant cell lines BE(2)-M17 and BE(2)-C, acute ATRA-induced apoptotic/senescence is severely diminished, though the cells still eventually differentiate.

Based on our data, we propose a conceptual model to explain how neuroblastoma cells are directed toward remarkably diverse cell fates upon RA exposure: in cells with high BMP signaling activity, BMP target genes (e.g. *ID* genes) inhibit differentiation, which would typically be induced by RA. In parallel with this, the co-binding of RARA with active SMAD transcription factors activates apoptosis or senescence, depending on baseline binding patterns of BMP effector SMADs (Fig. 5K, upper panel). When BMP signaling activity is lower, RARA is less prone to acutely activate apoptosis or senescence-related programs but can still bind and eventually activate a neuron-like differentiation program (Fig. 5K, lower panel).

## BMP signaling activity in neuroblastoma cells varies dramatically depending on the tumor site but is highly active in disseminated bone marrow metastases

RA's activity against neuroblastoma in vivo currently poses several unexplained contradictions, with clinical benefit in maintenance therapy, but little activity against established human tumors or in existing mouse models. To investigate whether BMP signaling activity could explain these discrepancies, we first implanted our hypersensitive CHP-134 cells in mice, then treated them with a pharmacologically relevant dosage of 13-*cis* RA (see Methods). However, consistent with previous studies, RA did not affect survival (Fig. 6A left;), nor tumor volumes (Fig. S8A), which is highly peculiar given the extreme sensitivity of CHP-134 to RA in vitro. We repeated these experiments by implanting the BE(2)-C cell line, again observing no tumor responses (Fig. 6A right, Fig. S8B). We confirmed tumor drug penetration and high concentrations of ATRA using HPLC (see Methods), thus neither drug formulation nor pharmacokinetics could explain this lack of activity (Fig. S8C, D).

To understand this collapse of CHP-134 response to RA in vivo, we performed RNA-seq on our vehicle-treated control xenograft tumors, comparing their gene expression profiles to CHP-134 cells growing in vitro (Fig. S8E–I). Remarkably, GSEA showed BMP signaling activity decreased dramatically in the xenografts (Fig. 6B, left panel; $P = 0.01$ for loss of *ID* gene expression signaling; Fig. S8J, K), with the expression of the BMP-target *ID1-4* family genes decreasing 50-fold (Fig. 6C). A Western blot for phosphorylated SMAD1/5/9 confirmed an almost complete loss of BMP signaling activity in the xenografts (Fig. 6D). However, while RA treatment had no measurable effect on survival, nor tumor volumes, the RNA-seq also indicated that neuronal differentiation-related gene expression was enriched in the vehicle-treated CHP-134 xenograft tumors, compared to the same cells untreated in vitro, suggesting the xenografts had begun to differentiate (Fig. 6E; Fig. S8L). This was mirrored by increased protein abundance of neuronal markers Tau (MAPT), MAP2, and DCX in the xenograft tumors (Fig. S8M). These data are consistent with the loss of BMP signaling activity in CHP-134 xenografts tipping the cell fate decision from apoptosis to differentiation, resulting in severely diminished RA activity in xenograft models, relative to in vitro culture.

These mouse data could plausibly indicate that the extreme/acute RA responses observed in cell lines are not relevant in vivo (nor clinically), which could arise if, hypothetically for example, BMP signaling is artificially inflated in vitro. Thus, we next investigated the activity of BMP signaling in neuroblastoma patient tumors, cell lines, and mouse models using single-cell RNA-seq (Fig. 6F). We assembled 5 datasets, including primary human tumors ($n = 20$)[46,47], orthotopic neuroblastoma patient-derived xenografts (PDXs) ($n = 17$), neuroblastoma cell lines ($n = 13$), and human neuroblastoma bone marrow metastases[48] ($n = 16$). We analyzed the data using automatic consensus nonnegative matrix factorization (acNMF)[49].

We estimated the activity of BMP signaling using the ID family of BMP-target genes[41,42] and compared these summary estimates across each of the relevant datasets. Mirroring the CHP-134 xenografts, BMP signaling activity was much lower in neuroblastoma cells in primary patient tumors and PDXs compared to cell lines (Fig. 6F; Supplementary Data 7). Remarkably however, the activity of BMP signaling in metastatic neuroblastoma cells in the bone marrow was much higher than in primary tumors, with an 8-fold higher expression of *ID1*– reaching levels comparable to CHP-134 cells in vitro (Fig. 6F, Fig. S9A–G). This result was consistent in a 2nd independent dataset, which isolated metastatic neuroblastoma cells from bone marrow aspirates by cell sorting based on GD2 expression and compared to primary tumors from the same patients using bulk RNA-seq[50] (Fig. S9H–M). This suggests that higher BMP signaling activity determines RA's unusual site-specific activity against disseminated neuroblastoma cells in bone marrow.

To confirm the site-specific BMP signaling activity, we obtained primary tumor and paired bone marrow biopsies from 6 neuroblastoma patients (Supplementary Data 8). We performed immunofluorescence (see Methods), co-labeling each sample for pSMAD1/5/9 (marking BMP signaling activity), PHOX2B (marking neuroblastoma cancer cells), and vimentin (marking vasculature). We calculated automated H-scores (a quantitative measure of staining intensity; see Methods) for pSMAD1/5/9 in PHOX2B positive neuroblastoma cells in bulk tumors, invasive regions, and disseminated single cancer cells in the bone marrow (Fig. 6G). The median H-score in neuroblastoma cells of the bulky primary tumors was 6.7, increasing markedly to 35.3 at invasive edges/regions (Fig. 6G, H; upper panels; Fig. S10A). However, in the bulkier tumors in the bone marrow the median H-score was 50.5, rising to 71.2 in the disseminated single neuroblastoma cells in the bone marrow, 10.6-fold higher than the median H-score in bulk primary tumors (Fig. 6G, H; lower panels; Fig. S10A). Collectively, these results suggest that BMP signaling activity is higher in disseminated neuroblastoma cells, especially at bone marrow metastatic sites.

We wondered why BMP signaling activity was higher in disseminated bone marrow metastases. We further explored the bone marrow single-cell RNA-seq data, identifying "bone marrow stromal cells" (mSigDb cell types database, $OR = 184.56$, $P = 2.6 \times 10^{-57}$) as overexpressing *BMP2*, *BMP4* and *BMP5* (Fig. 6I; the only bone marrow cell type identified overexpressing BMP ligands). We tested whether neuroblastoma cells respond to these specific BMP ligands, directly treating CHP-134 and TGW cell lines with recombinant BMP. We tested BMP2 and BMP4, as well as BMP7, which was not overexpressed in these bone marrow stromal cells, and BMP1, which is a protease (not a BMP ligand) and serves as a negative control. In CHP-134 cells, we found that BMP2 and BMP4 induced strong phosphorylation of SMAD1/5/9, BMP7 had a milder effect, and BMP1 did not activate BMP signaling (Fig. 6J). In the presence of BMP2, BMP4, and BMP7, the IC50 of ATRA decreased nearly 100-fold reaching very low picomolar concentrations (Fig. 6K), but BMP1 treatment had no effect. Consistent with this, BMP2, BMP4, and BMP7 also promoted strong expression of ATRA-induced cleaved PARP (Fig. 6J) and cell death (Fig. 6L)). In TGW cells, recombinant BMP2 and BMP4 also induced strong phosphorylation of SMAD1/5/9 (Fig. 6J) and led to a clear increase in

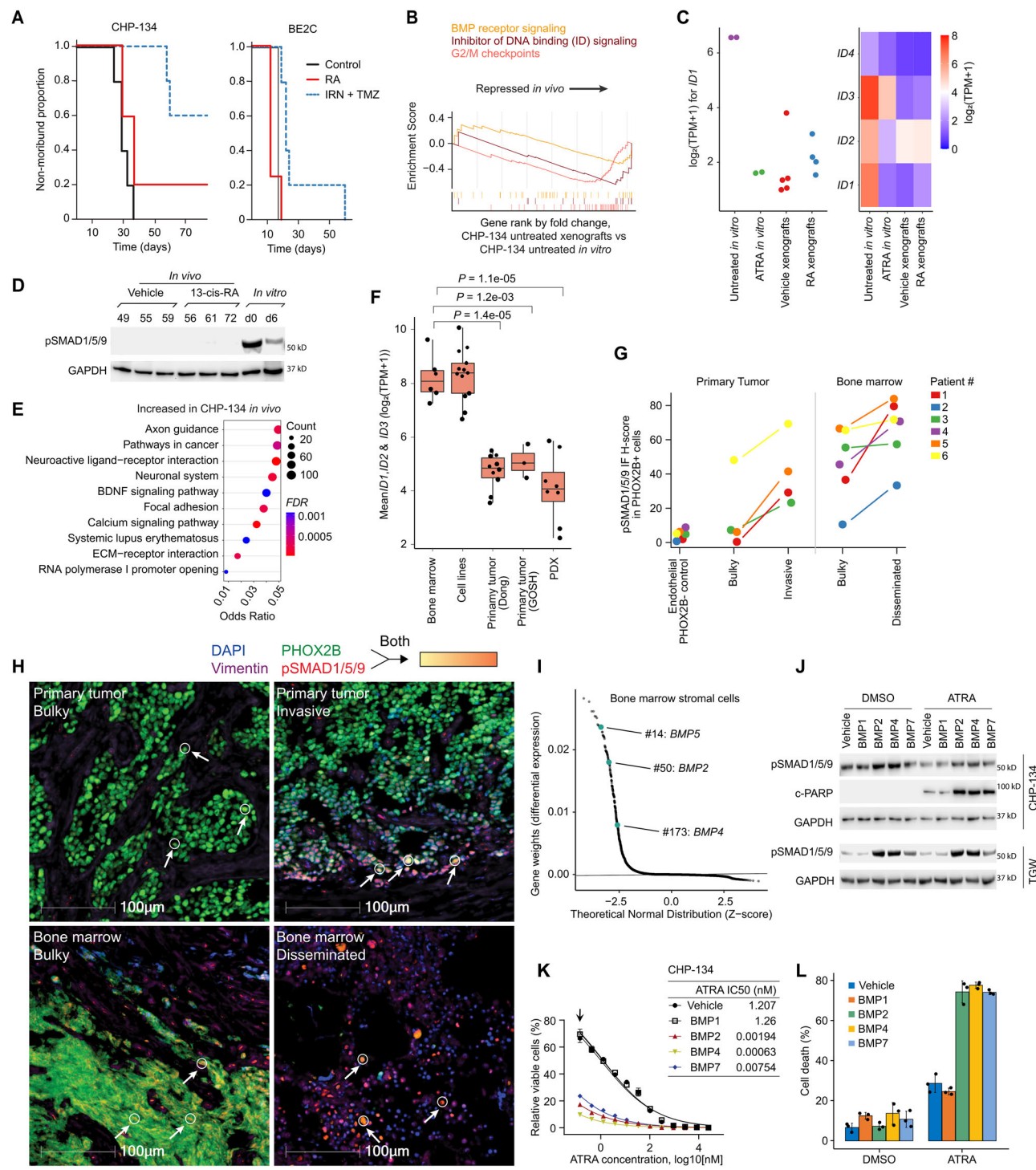

senescence and decreased viability when co-administered with ATRA (Fig. S10B, C), while BMP1 and BMP7 had a much weaker effect.

Overall, these data imply that disseminated neuroblastoma cells are responding to BMP ligands manufactured in the bone marrow microenvironment. This can result in sensitization to pharmacological concentrations RA and the clearance of these cells. This model provides a clear mechanism to explain RA's site-specific activity and clinical utility in neuroblastoma maintenance therapy.

## Discussion

The reason only a subset of neuroblastoma patients benefit from retinoic acid (RA) treatment is not understood, nor have RA's diverse and tumor-site-specific activities ever been explained. This limits our ability to stratify treatment or to develop drug combinations to improve the activity of this drug. Indeed, this lack of clarity around RA's mechanism, combined with a relatively small number of sufficiently powered clinical studies, has fueled recent debate about RA's effectiveness in neuroblastoma[51]. In this study, we discovered that in the most RA-sensitive neuroblastoma cell lines the primary mechanism of anti-neoplastic activity is acute apoptosis or senescence and that these activities are mediated by BMP signaling. We showed that BMP signaling activity in neuroblastoma heavily depends on the cellular context but is high in metastatic cells in the bone marrow and in cell lines in vitro. This context specificity is likely partially determined by access to BMP ligands[52] and provides an explanation for the site-specific activity of retinoic acid during

**Fig. 6 | BMP signaling activity varies dramatically depending on neuroblastoma tumor site but is highly active in disseminated bone marrow metastases. A** Kaplan−Meier curves showing the survival of mouse. 5 mice were used in each group. Each mouse was implanted with luciferase-labeled CHP-134 cells (left panel) or BE(2)-C cells (right panel) by subcutaneous injection. RA, 13-cis retinoic acid; IRN, irinotecan; TMZ, temozolomide. The IRN + TMZ group is intended as a standard-chemo positive control. **B** GSEA plots summarizing relevant differentially expressed pathways in vehicle treated CHP-134 tumors in mouse versus untreated CHP-134 cells in vitro. Running Enrichment Scores are shown by colored lines. **C** Stripchart (left panel) showing *ID1* expression (y-axis) in CHP-134 cells growing under different in vitro and in vivo conditions (x-axis). Heatmap (right panel) showing the expression of *ID1-ID4* under the same conditions. **D** Western blot for phospho-SMAD1/5/9 in CHP-134 cells under various in vitro and in vivo RA treatment conditions. GAPDH was used as a loading control. The representative results of 3 independent replicates are shown. **E** Dot plot showing the top enriched pathways for differentially expressed genes in vehicle-treated CHP-134 mouse xenografts to vehicle treated CHP-134 cells in vitro. *P*-values (color) and odds ratios (x-axis) were calculated by hypergeometric test. Exact *P*-values are provided in the Source Data file. **F** Boxplots estimating BMP signaling activity based on the average TPM expression of *ID1-ID3* (y-axis) in clusters of neuroblastoma cancer cells (dots), identified using acNMF from single-cell RNA-seq in different datasets (x-axis). *P*-values were calculated with two-sided Wilcoxon tests. **G** H scores (y-axis; see Methods) quantifying the IF staining intensity of phospho-SMAD1/5/9 in cells that also stain positive for PHOX2B (marking neuroblastoma cancer cells) in six different neuroblastoma patients (n = 6, each color indicates a patient). The x-axis shows the different conditions/sites. Note: the endothelial controls cells are PHOX2B

negative. **H** Representative IF images from Patient #3 showing merged marker staining in bulk primary tumor (left upper panel), a lymphovascular invasion region of primary tumor (right upper panel), bulk metastatic tumor in the bone marrow (left lower panel) or disseminated cells in the bone marrow (right lower panel). White arrows and circles indicate representative cells positively stained with both PHOX2B and pSMAD1/5/9 (these cells appear yellow/orange, see scale bar, which is shown for illustrative purposes). Additional detailed images are included in Fig. S10A. **I** Scatter plot highlighting the overexpression of BMP ligands in 'bone marrow stromal cells' identified using single-cell RNA-seq on bone marrow aspirates from metastatic neuroblastoma patients. Dataset from Fetahu et al. **J** Expression of phosphorylated SMAD1/5/9 and c-PARP in CHP-134 and TGW. Cells were treated with DMSO or 0.03 μM ATRA in the presence of vehicle control or 10 ng/ml BMP recombinant proteins. CHP-134 was treated for 3 days and TGW was treated for 24 h. For CHP-134, all antibodies were incubated with the same membrane. GAPDH was used loading control. The representative results of 3 independent replicates are shown. **K** Viability of CHP-134 cells (left) treated with ATRA for 6 days in the presence of vehicle control or 10 ng/ml BMP recombinant proteins. Cell viability was measured with CellTiter-Glo. ATRA $IC_{50}$ for CHP-134 (right) in the presence of each BMP was calculated with GraphPad Prism 9. Data represent the mean ± SD of 3 independent replicates. P values at the arrow indicated point for BMP1, 2, 4, and 7 are 0.37, $1.55 \times 10^{-5}$, $7.42 \times 10^{-6}$, and $2.83 \times 10^{-5}$ (two-sided t-test). **L** Cell counting for CHP-134 treated as in (**J**). Dead cells were determined by trypan blue staining. Data represent the mean ± SD of 3 independent replicates. P values (DMSO vs ATRA) for vehicle, BMP1, 2, 4, and 7 are $1.6 \times 10^{-3}$, $8.81 \times 10^{-4}$, $3.58 \times 10^{-5}$, $3.96 \times 10^{-5}$, and $1.28 \times 10^{-5}$ (two-sided t-test). Source data are provided as a Source Data file.

neuroblastoma maintenance therapy, as well as the loss of drug response in cell line xenografts.

BMP signaling has very diverse activities during human/neuronal development[53], including specifying sympathoadrenal progenitors from neural crest cells during early development[54]. The activities we observed in neuroblastoma cells upon exposure to BMP are especially consistent with BMP's function as a negative regulator of neural commitment of human embryonic stem cells, where treatment with BMP inhibitors such as noggin causes neuronal differentiation[55]. Additionally, interactions between RA and BMP signaling are well established in developmental biology, where the coordinated activities of these pathways regulate cell fate decisions, including differentiation, apoptosis, and senescence[20–24,53,56–58]. This has been shown most clearly in vertebrate limb development, where high BMP levels in interdigital regions leads to apoptosis, while experimentally blocking BMP leads to differentiation and the development of webbed feet[20]—this specific BMP-dependent activity was subsequently shown to depend on the presence of naturally occurring retinoic acid[22].

Our results suggest that these normal developmental mechanisms can be maintained in neuroblastoma cells—a cancer of the *developing* peripheral nervous system. These mechanisms can be activated by pharmacological concentrations of RA, with the levels and binding profiles of BMP signaling transcription factors determining cell fate decisions. The site-specific activity of BMP signaling in human neuroblastoma bone marrow metastases seems to explain both the previously documented activity of RA in the bone marrow[11,14,18] and the utility of RA in the maintenance therapy setting[12], when bulky tumors have already been eliminated by conventional chemotherapy/surgery. To our knowledge, such a mechanism has not been previously described as an effective means to treat cancer. We propose naming this process "developmental hijacking" and speculate that similar undiscovered vulnerabilities may exist in other cancers, particularly pediatric cancers of developmental origin[59].

The observation that the interaction of BMP signaling and RA can induce senescence in some neuroblastoma cells may represent a particularly interesting avenue for future investigation. Like apoptosis, senescence is an important activity in controlling embryonic stem cell fate[60]. However, the relationship between apoptosis and senescence is multifaceted and still poorly understood[60], and it is likely additional

factors, including *TP53* mutation[61], which is evident in e.g. our TGW cell line, may also contribute to tipping neuroblastoma cells towards this outcome. Interestingly, numerous recent studies have suggested that senescent cancer cells exhibit amplified antigen presentation, which can lead to clearance of these cells by the adaptive immune system via e.g. T cell receptor engagement[62–65]. Additionally, clearance of senescent cells via the innate immune system (e.g. natural killer cells and macrophages) has been long established[66]. Indeed, recent evidence showed that senescence induced by retinoic acid signaling activation can robustly enhance natural killer cell-mediated tumor clearance in prostate cancer[67]. Neuroblastoma is one of the only pediatric cancers where a successful immunotherapy (targeting the ganglioside GD2) has been developed, and it is plausible that RA-induced senescence could potentiate the activity of this or other immunotherapies in some cases. Remarkably, anti-GD2 in combination with RA in neuroblastoma has also recently been shown to have site-specific activity in the bone marrow[18], which could potentially result from an interaction between anti-GD2 and retinoic acid-induced senescence. Notably, several therapies have also been proposed to target senescent cells, which may represent rational combinations with RA in some cases[68].

Additionally, although not examined in detail here, our analyses revealed cell lines from other cancers, including medulloblastoma, exhibiting a dramatic apoptotic response to RA in a short-duration[69]. This suggests aspects of our findings could be generalizable beyond neuroblastoma if a common molecular mechanism underlies RA sensitivity across different cancer types. This means RA may represent a viable treatment option in a broader range of patients than currently appreciated, and that such individuals may be identifiable based on the baseline activity of BMP signaling and RA receptors. Future research may also further detail the molecular steps pushing neuroblastoma towards either apoptosis or senescence upon RA exposure, given high BMP signaling; based on existing data in various model systems, this cell fate decision is likely multifactorial, and context specific[70]. Inevitably, additional factors not examined in this study will further influence response to RA and the interactions RA and BMP signaling. Such factors may include, for example, hypoxia response, which has been shown to influence, and be influenced by, retinoic acid signaling[71]. Furthermore, future studies may benefit from dissecting abundance of specific BMP ligands in primary tumors and various metastatic sites,

and their influence on RA response. Such investigations may also benefit from improved mouse models of bone marrow metastatic neuroblastoma from diverse patient derived cells, features of which may explain further variability in RA response in the clinic. In conclusion, our study reveals a generalizable, role of BMP signaling in determining neuroblastoma cell fate and sensitivity to RA. Our findings provide important insights into RA's mechanism of action in neuroblastoma, provide an explanation for its context-specific activity, and suggest there could be value in further investigating the interplay between RA and BMP signaling in other cancers. Our findings also have significant implications for the development of novel combination therapies, for example co-targeting the RA and BMP signaling pathways, but also including anti-GD2 and other immunotherapies, insights that may ultimately lead to improved outcomes for patients with high-risk neuroblastoma.

## Methods

### Ethics statement

This study complied with all relevant ethical regulations. All animal experiments were carried out in strict accordance with the recommendations in the Guide to Care and Use of Laboratory Animals of the National Institute of Health. The protocol was approved by the Institutional Animal Care and Use Committee at St. Jude Children's Research Hospital (protocol number 648-100602). To minimize suffering, the maximum tumor burden allowed was ~20% of mouse body weight. All mice were housed on a 12 – 12 light cycle (light on 6 am off 6 pm) with temperature- and humidity-controlled conditions and provided food and water *ad libitum*.

### Cell culture, transfection, and transduction

All cells are cultured at 37 °C with 5% $CO_2$. Cell lines were tested negative for mycoplasma using the MycoAlert mycoplasma detection kit (Lonza, LT07-118). Cell culture conditions and sources are listed in Supplementary Data 10.

SMAD9 and SMAD1 All-in-one sgRNAs and non-targeting control (cat# GSGC12066) for CRISPR knockout were purchased from Horizon Discovery (Horizon Discovery, USA). The sgRNA clone numbers are GSGH12181-249556872, GSGH12181-249518932, and GSGH12181-249342972 for SMAD9; GSGH12181-249337252, GSGH12181-249607466, and GSGH12181-249431172 for SMAD1. Lentiviral shRNA plasmids for ACVR1 knockdown were purchased from Horizon Discovery (clone number: RHS4430-200180498, RHS4430-200262778, RHS4430-200263312). The plasmid for ID2 overexpression was from Addgene (Plasmid #83096). To generate lentiviral particles, 293 T cells were seeded in a 6-well plate 1 day before transfection. For one well of a 6-well plate, cells were co-transfected with 0.56 μg of pMD2.G (Addgene plasmid # 12259), 0.83 μg of psPAX2 (Addgene plasmid # 12260), and 1.1 μg of sgRNA or shRNA plasmid, using Lipofectamine 3000 (ThermoFisher, # L3000008). Cell culture medium with lentivirus were collected three times from 48 h to 72 h after transfection. Cells were infected two to three times during 24 h. For SMAD9 and SMAD1 knockout, GFP+ cells were collected using flow cytometry after 2 days of infection. For ACVR1 knockdown, positive cells were selected with puromycin after 2 days of infection.

### CRISPR screen

The Human Brunello CRISPR KO library was a gift from David Root and John Doench (Addgene #73179)[72]. The Center for Advanced Genome Engineering at St. Jude Children's Research Hospital amplified and validated the library using protocols described in the Broad Genome Perturbation Portal (https://portals.broadinstitute.org/gpp/public/resources/protocols), substituting Endura DUOs electrocompetent cells for STBL4. Following expansion, the Brunello library was validated using calc_auc_v1.1.py (https://github.com/mhegde/) and count_spacers.py[73]. The St. Jude Vector Development and Production laboratory produced the lentiviral particles.

CHP-134 cells were transduced with the lentiCas9-Blast lentivirus (Addgene #52962). Media with 7 μg/ml of blasticidin was added 2 days later to select positive cells. Cas9 expression was determined by western blotting with Cas9 antibody (Cell Signaling, #14697). Cas9 activity was tested using the CRISPRtest™ Functional Cas9 Activity Kit for Human Cells (Cellecta, #CRTEST). The Cas9-expressing cells were transduced at day 0 with lentiviral Brunello pooled library[72] at a low MOI to make the transduction efficiency ~30%[74]. 0.5 μg/ml of puromycin was used for selection after 2 days of transduction (day 2). Cells were subcultured on day 4. On day 6, cells were split into two groups, with 500 representatives for each gRNA in each group. On the next day (day 7), one group of cells were treated with 2 μM of ATRA, DMSO was added to the other group. Cells were collected after 3 days (day 10) and 6 days (day 13). Genomic DNA (gDNA) was extracted using the NucleoSpin Blood XL kit (Macherey-Nagel, # 740950.50). The gDNA was amplified according to Broad Genome Perturbation Portal protocols and single end 100 cycle sequencing was performed on a NovaSeq 6000 (Illumina) by the Hartwell Center for Genome Sequencing Facility at St. Jude. Results were analyzed using the the de facto tool MAGeCK ver. 0.5.7[27] using the default parameters (see "Overview of the MAGeCK algorithm" section from Li et al. for a brief description of the algorithm).

### Cell viability assays

Cell viability was determined using either MTS or CellTiter-Glo. For MTS assay, cells were seeded in clear bottom 96-well or 384-well plates and treated with compounds as needed. After 3 days or 6 days of incubation, premixed MTS reagent (Abcam, ab197010) was added to each well of the plates to a final concentration of 10%. Then the plates were incubated in a cell culture incubator for 0.5 to 4 h and absorbance was measured at 490 nm. For CellTiter-Glo (Promega, G9241), cells were treated and cultured in white bottom plates and the reagent was added to the plates to a final concentration of 50%. The plates were then shaken for 2 min to induce cell lysis. After incubation at room temperature for 10 minutes, the luminescence was recorded.

### Western blotting

Generally, cells were washed carefully with cold PBS and lysed in RIPA buffer (ThermoFisher, # 89900) in cell culture plates. RIPA buffer was supplemented with Protease Inhibitor Cocktail (Sigma, #5892970001) and Phosphatase Inhibitor (Thermo Fisher, #A32957). For cleaved PARP detection, cell culture medium was also collected. Protein concentration was determined with BCA protein assay kit (Thermo Fisher, # 23225). After boiling in 1× LDS sample buffer (Thermo Fisher, #NP0007), proteins were separated by SDS-PAGE and transferred to PVDF membranes. After blocking in 5% non-fat milk in TBS with 0.1% Tween-20, membranes were probed with primary antibodies and horseradish-peroxidase-conjugated secondary antibodies. The signal was visualized with SuperSignal™ West Femto Maximum Sensitivity Substrate (ThermoFisher, # 34095) and ChemiDoc Imaging System (Bio-Rad). Unprocessed images are included in the Source Data file. The antibodies used for western blotting in this study are β-Actin (Sigma, # A1978), GAPDH (ThermoFisher, # MA5-15738), cleaved PARP (Cell Signaling, # 5625), PARP (Cell Signaling, # 9542), SMAD9 (ThermoFisher, # 720333), SMAD1 (Cell Signaling, # 6944), pSMAD1/5/9 (Cell Signaling, # 13820S), total SMAD1/5/9 (Abcam, # ab13723), ACVR1 (Abcam, # ab155981), MAP2 (Cell Signaling, # 4542), HRP-linked anti-rabbit IgG (Cell Signaling, # 7074), HRP-linked anti-mouse IgG (Cell Signaling, # 7076).

### Immunofluorescence

Cells were seeded on poly-D-lysine coated coverslips (Neuvitro, # GG-18-PDL) and treated with the necessary reagents. Treated cells were fixed in 4% formaldehyde for 15-20 minutes and permeabilized with 0.3% Triton X-100 for 10 min at room temperature. The cells were

washed with PBS after that and blocked in 5% FBS/PBS for 1 h at room temperature, then incubated with primary antibody diluted in 1% FBS/PBS overnight at 4 °C (MAP2, Cell Signaling # 4542, 1:500 dilution; NEFH, Cell Signaling # 55453, 1:500 dilution). After 3 washes with PBS, coverslips were incubated with a secondary antibody (Alexa fluor 594 conjugate anti-rabbit IgG, Cell Signaling # 8889, 1:1000 dilution) for 1 h at room temperature. ProLong® Gold Antifade Reagent with DAPI (Cell Signaling, #8961) was used as a coverslip mountant. Coverslips were mounted for 24 h at room temperature and subjected to fluorescence microscopy. Images were taken with the Nikon C2 confocal microscope and analyzed with NIS-Elements Viewer imaging software.

## Flow cytometry for apoptosis analysis

For apoptosis, cells were treated in 6-well plates. After incubation for the time needed, cell culture medium and cells were collected and washed in cold phosphate-buffered saline (PBS). The cells were then resuspended in annexin-binding buffer and stained with Alexa Fluor 488 conjugated annexin V (Thermo Fisher, catalog# A13201) and 1 μg/ml Propidium iodide (PI) for flow cytometry analysis. The results were analyzed and quantified with FlowJo.

## Senescence assays

The senescence cell staining kit and fluorescence assay kit were purchased from Cell Signaling (Senescence β-Galactosidase Staining Kit, # 9860; Senescence β-Galactosidase

Activity Assay Kit (Fluorescence, Plate-Based), #23833). Assays were performed following the vendor's instructions. Briefly, for cell staining assay, cells were seeded on poly-D-lysine coated coverslips (Neuvitro, # GG-18-PDL). After treatment, cells were fixed and stained with β-Galactosidase Staining Solution containing X-Gal. The pH value of the staining solution was strictly controlled in the range of 5.9 – 6.1. The coverslips in the staining solution were incubated at 37 °C for 24 h in an incubator without $CO_2$. Then the staining solution was removed and the coverslips were mounted with 70% glycerol. Images were taken with Nikon Eclipse Ni upright microscope. For the fluorescence assay, cells were seeded and treated in 6-well plates. After a wash with cold PBS, cells were lysed with Senescence Cell Lysis Buffer. The total protein concentration of each cell lysate was determined with a BCA protein assay kit (Thermo Fisher, # 23225). For each sample, 50 μl of cell lysate and 50 μl of 2X Assay Buffer containing the

SA-β-Gal Substrate 4-MUG were mixed in a 96-well plated and incubated at 37 °C protected from light for 3 h. The mixture was transferred to a 96-well black opaque plate and fluorescence with excitation at 360 nm and emission at 465 nm was measured with CLARIOstar Plus microplate reader (BMG Labtech).

## Generation of the cell line with HA-tagged SMAD9

Genetically modified cells were generated using CRISPR-Cas9 at the Center for Advanced Genome Engineering at St. Jude Children's Research Hospital. Briefly, 400,000 CHP-134 cells were transiently co-transfected with precomplexed ribonuclear proteins (RNPs) consisting of 100 pmol of chemically modified sgRNA (Synthego), 33 pmol of 3X NLS SpCas9 protein (St. Jude Protein Production Core), 200 ng of pMaxGFP (Lonza), and 3ug of ssODN donor[50]. Transfections were performed using nucleofection (Lonza, 4D-Nucleofector™ X-unit) with solution P3 and program CA-137 in a small (20ul) cuvette according to the recommended protocol from the manufacturer. Single cells were sorted based on transfected cells (GFP + ) five days post-nucleofection into 96-well plates containing prewarmed media and clonally expanded. Clones were screened and verified for the desired modification using targeted deep sequencing and analyzed with CRIS.py as previously described[75]. Final clones tested negative for mycoplasma by the MycoAlert™Plus Mycoplasma Detection Kit (Lonza) and were authenticated using the PowerPlex® Fusion System (Promega) performed at the Hartwell Center at St. Jude Children's Research Hospital.

Editing construct sequences and screening primers are listed in Supplementary Data 10 (the star indicates AltR modifications[50]).

## ChIP-seq

**Cross-linking.** After the desired treatment, cells were incubated with 1% formaldehyde to crosslink the proteins and DNA. Glycine was added to a final concentration of 225 mM to quench the reaction. 20 million cells were used for each sample.

**Chromatin shearing.** The cells were lysed with Covaris Lysis buffer (0.25% Triton X-100, 1 mM EDTA pH 8.0, 140 mM NaCl, 50 mM HEPES pH 7.9, 10% glycerol, 0.5% NP-40). The chromatins were collected by centrifuge and the pellets were washed twice with Covaris Wash Buffer (1 mM EDTA pH 8.0, 10 mM Tris HCl pH 8.0, 200 mM NaCl, 0.5 mM EGTA pH 8.0) and twice with Covaris Shearing Buffer (0.1% SDS, 1 mM EDTA pH 8.0, 10 mM Tris HCl pH 8.0). Then the chromatin was sheared to 200 – 700 bp in Covaris Shearing Buffer with the ultrasonicator Covaris LE220 (Covaris, Woburn, MA).

**Immunoprecipitation.** Protein A/G-coated magnetic beads (protein A-coated beads (Diagenode, # C03010020-150) for rabbit antibody; protein G-coated beads (Diagenode, # C03010021-150) for mouse antibody) were washed with 0.5% BSA in PBS (g/ml) and added to the sheared chromatin to preclean the samples by rotation at 4 °C for 1 h. 100 μl of the beads were used for each sample. The magnetic beads were isolated with a magnetic separation rack and the supernatant was collected for immunoprecipitation. Primary antibodies were added to the samples, then the samples were incubated at 4 °C overnight with rotation. The antibodies used in this study are HA (Cell Signaling, # 3724; use 40 μl of antibody for each sample in 2 ml of buffer (1:50 dilution)), SMAD4 (R&D Systems, # AF2097; use 20 μg per 20 million cells), and RARA (R&D Systems, # PP-H1920-00; use 20 μg per 20 million cells). On the second day, 100 μl of freshly washed protein A/G-coated magnetic beads were added to each sample. Then the samples were incubated at 4 °C for 5 h with rotation.

**Elution and reverse cross-links.** The beads were collected and washed once with cold Low Salt Wash Buffer (0.1% SDS, 1% Triton X-100, 2 mM EDTA, 20 mM Tris pH 8.0, 150 mM NaCl), twice with cold High Salt Wash Buffer (0.1% SDS, 1% Triton X-100, 2 mM EDTA, 20 mM Tris pH 8.0, 500 mM NaCl), once with cold LiCl Wash Buffer (1 mM EDTA, 10 mM Tris pH 8.0, 250 mM LiCl, 1% NP-40, 1% DOC), and twice with cold TE Buffer pH 8.0 (1 mM EDTA, 10 mM Tris pH 8.0). Then the DNA were eluted from the beads with 100 μl of freshly made Elution buffer (0.1 M NaHCO3 and 1% SDS). 4 μl of 5 M NaCl and 2 μl of RNaseA (10 mg/mL) (ThermoFisher, # EN0531) were added to each sample and the samples were then incubated at 65 °C with shaking overnight.

**DNA purification.** Proteins in the samples were removed by adding 5 μg of proteinase K (ThermoFisher, # AM2548) and shaking at 60 °C for 2 h. Afterward, the DNA was purified with Qiagen MinElute PCR Purification Kit (Qiagen, # 28004).

**Sequencing.** ChIP-seq DNA libraries were prepared using the KAPA HyperPrep Kit (Roche, # KK8504) and sequenced on the Illumina NextSeq 500 system. At least 50 million reads were collected for each sample.

## RNA-seq

Cells were cultured and treated as desired. Total RNA was extracted with TRIzol Reagent (ThermoFisher, cat# 15596026). For each sample, 150 ng of total RNA was used for library prep using Illumina® Stranded Total RNA Prep, Ligation with Ribo-Zero Plus (Illumina, cat# 20040529).

For tumor samples, tumor tissues were homogenized in TRIzol Reagent with FastPrep-24™ 5 G (MP Biomedicals, cat# 116005500). RNA-seq libraries were prepared using the KAPA RNA HyperPrep Kit with RiboErase (HMR) (Roche, cat# KK8561). For each sample, 200 ng of RNA were mixed with 4 μl of diluted ERCC RNA Spike-In Mix (Invitrogen, cat# 4456740. 1:1,000 dilution). All libraries were sequenced on the Illumina NovaSeq 6000 system.

## High-throughput drug combination (synergy) screening

ATRA in combination with FK506 or rapamycin were dispensed into 384-well flat-bottom tissue culture-treated solid white plates (Corning, Catalog# 3570) using a Labcyte Echo 555 acoustic Liquid Dispenser (Beckman Coulter Life Sciences, USA). Ten-point dose-response curves were generated using 1:3 dilution intervals for each drug. For each drug combination, three replicates of the 100 combination matrices (all possible combinations of drug concentrations) were used per cell line. The final DMSO concentration of all wells was 0.536%. Cells were plated in volumes of 40 μL in the compound-containing plates using a Multidrop Combi reagent dispenser (ThermoFisher Scientific, USA) and settled for 20 s at $100 \times g$ in an Eppendorf 5810 centrifuge. Plates were then incubated at 37 °C in 5% $CO_2$ for 6 days. Following incubation, plates were equilibrated to room temperature for 20 min before viability assessment. 25 μL of CellTiter-Glo (Promega Catalog# G9241) was then added to each well to measure viability using a Multidrop Combi and the plates were incubated for an additional 25 min at room temperature. Luminescence was then detected with an EnVision 2102 Multilabel Plate Reader (PerkinElmer Life Sciences, USA).

## Drug synergy analysis

Raw luminescent data were imported into the R statistical environment version 4.0.2 (www.r-project.org). Background-subtracted values in raw luminescent units[48] were assigned to the appropriate drugs and concentrations, and each replicate was separated. All replicates were normalized to the mean of their respective inter-plate controls (DMSO for 0% cell death, and the maximum concentration of each drug combined for 100% cell death). Matrices of the percent cell death values were constructed using means of normalized data from each of the three replicates per group as input. Synergy scores were calculated from these normalized values for all tested concentration combinations. Delta scores derived from the Zero Interaction Potency (ZIP) model were used to score synergy across all samples using the SynergyFinder package. The most significant synergy (or antagonism) scores of drug pairs were directly compared to their corresponding single agents by referencing original normalized data of individual replicates and plotting mean values of percent cell death. P-values were calculated with a two-tailed one-sample t-test comparing the replicates of the combination effect to the expected additive value. Synergy needs to meet both criteria: (1) the "combined" value is higher than the red dotted line (additive effect); (2) $P < 0.05$. The highest-scoring concentration pairs were extracted from the resulting synergy matrices to represent the most significant synergy.

## Retinoic acid quantitation in mouse xenograft tumor tissues

Tumor tissues were kept in liquid nitrogen until analysis. To prepare samples for analysis, the tissues were weighed and homogenized in cold PBS (PBS (μl): tissue (mg) = 1:2) with FastPrep-24™ 5 G (MP Biomedicals, cat# 116005500). The samples were processed for bioanalytical analysis immediately after homogenization. To prepare the standards and quality control samples, 10 mM solution of 13-cis-RA (MedChemExpress, HY-15127), ATRA (MedChemExpress, HY-14649), and Etretinate (MedChemExpress, HY-B0797) were dissolved in 100% DMSO. All the stock solutions were protected from light and stored in a -20 °C freezer until use.

The calibration standards and quality control (QC) samples were prepared by using the Echo liquid dispenser (Beckman Coulter Life Sciences, Indianapolis, Indiana). A dose-response (DR) plate was made with 3-fold serial dilution and backfilled with DMSO to achieve final concentrations of 1111.11, 370.37, 123.46, 41.15, 13.72, and 4.57 μM. The QC stocks had concentrations of 4.57, 18.29, 201.19, and 1001.37 μM. 50 nL of calibration standards and QC samples were dispensed from the DR plate into an assay plate. Then, 50 μL of PBS were added to each well in the assay plate to obtain final concentrations of 1111.11, 370.37, 123.46, 41.15, 13.72, and 4.57 nM for the calibration curve. Samples with concentrations of 4.57, 18.29, 201.19, and 1001.37 nM were used as low (LQC), medium1 (MQC1), medium2 (MQC2), and high (HQC) QC samples, respectively.

Sample Processing: To extract the samples, protein precipitation (PP) was performed. 50 μL of spiked samples and unknown tumor tissue homogenate were placed in a 96-well analytical plate and then quenched with 100 μL of acetonitrile containing 100 nM of Etretinate as an internal standard. The plate was sealed and centrifuged at 4000 rpm for 15 minutes. 75 μL of the supernatant were transferred to a new analytical plate and 31.2 μL of MilliQ water was added to each well. Samplers were mixed on a plate shaker at 1000 rpm for 2 minutes. Finally, 10 μL of the samples were injected onto the LC-MS/MS system (Sciex 6500 tripleQ).

LC-MS/MS instrumentation: the analytical system used was a Waters Acquity I class UPLC system with a binary pump, integrated degasser, column oven, and autosampler. The column used was an Acquity UPLC BEH C18 analytical column with dimensions of 1.7 μm, 100 Å, and 2.1 × 100 mm, which was kept at a column oven with the temperature of 60 °C. The mobile phase was composed of 0.1% formic acid–water (solvent A) and 0.1% formic acid–acetonitrile (solvent B), and a gradient was applied according to the following conditions: 0.0 min, 25% A–75% B; 0.2 min, 25% A–75% B; 3.0 min, 5% A–95% B; 3.5 min, 5% A–95% B; and 4.0 min, 25% A–75% B. The flow rate was 0.5 mL/min, and 10 μL of samples were injected using a partial loop with needle overfill (PLNO) mode. The first 0.5 min of eluate was directed to waste to desalt, and the remaining eluates were directed to the Sciex 6500 triple quadrupole mass spectrometer equipped with an electrospray ionization source. LC-MS/MS was performed in positive polarity at 5500 V, with a source temperature of 550 °C, and gas 1 and gas 2 settings for nitrogen set to 60 and 60, respectively. The curtain gas and collision gas were also nitrogen, and set to 30 and 10, respectively. The multiple reaction monitoring transitions were m/z 301.3 to m/z 123.2 for 13 cis retinoic acid, m/z 301.2 to m/z 123.1 for all trans retinoic acid, and m/z 355.134 to m/z 205.1 for the IS (Etretinate). The declustering potentials, entrance potential, collision energy, and collision cell exit potential were set at 60 V, 10 V, 21 V, and 14 V for 13 cis RA; 60 V, 10 V, 23 V, and 15 V for ATRA; and 60 V, 10 V, 17 V, 24 V for Etretinate. The data acquisition was conducted using Analyst 1.6.3 software (SCIEX), and the data processing was carried out using MultiQuant 2.1.1 software (SCIEX).

## Next-generation sequencing data analysis

**ChIP-seq analysis**. ChIP sequencing data of each sample were trimmed by Trimgalore v0.4.4[76] to remove low quality reads and adapter sequences. Then, the trimmed reads were mapped to the hg38 human reference genome using BWA v0.7.17[77]. Mapping quality metrics were generated using the flagstat command of Samtools v1.9[78].To generate genome-wide coverage profiles, the bedtools v2.17.0 "genomecov" command was applied[79], and the output was converted to bigwig files using bedGraphToBigWig for visualization in IGV[80]. MACS2 v2.1.1 was utilized to identify ChIP-seq signal enriched regions (referred to as "peaks" or "binding regions" in this manuscript)[81]. The blacklist provided by the ENCODE project for the human genome (hg38) was used to filter out peaks in blacklisted regions. Additionally, we generated a customized blacklist specifically for neuroblastoma cells to filter out peaks in the problematic highly repetitive regions, such as MYCN region.To compare the peak profiles between samples pulled out by

the same antibody but with different treatment condition, or across different cell types, a consensus set of peak coordinates is needed for comparison of different samples. In each sample, we selected only those peaks that appeared in both technical replicates, referred to as recurrent peaks. To achieve this, we used the bedtools v2.17.0 intersect function to identify overlapping peak regions with at least a 1 base pair (bp) overlap between the two replicates. Then, the recurrent peaks from the different samples being compared were merged to form a consensus peak coordinate using the mergeBed function from bedtools. The mapped read counts (binding intensity) in these consensus regions were then recalculated using bamtobed function from bedtools.

To visualize the ChIP-seq signal across different samples, we utilized the computeMatrix function from deeptools v3.5.0[82] to obtain a binding signal on consensus regions from the coverage profile (bigwig file) of each sample. The generated binding matrix file was fed to the plotHeatmap function from deeptools to generate the heatmap such as that shown in Fig. 5A.

Differential binding events were identified using DESeq2[83], which compared the read counts in the binding regions between drug-treated and untreated samples. Peaks were determined to have different intensities between drug-treated and untreated samples if they exhibited a $\log_2$ fold change > 1 or < -1, with an *FDR* less than 0.05.

The genomic annotation of peak proximity to the nearest transcription start site was calculated using the ChIPseeker R package[84]. Pscan-ChIP v1.3 was used to identify enriched DNA sequence motifs corresponding to RARA binding sites within the peak regions[85] for drug-treated samples.

Enriched biological pathways among genes bound by the peaks were identified using the GSEA and enricher functions (based on hypergeometric test) from the clusterProfiler R package[86] and gene sets used in the annotation were defined in BioPlanet 2019[87].

For GSEA annotation of genes bound by SMAD9, SMAD4 or RARA, genes were ranked by binding intensity of the transcription factor, and pathways that enriched for genes with strong TF binding were identified. The same GSEA annotation procedure was applied to a subset of genes that were bound by both RARA and SMAD9 (or SMAD4).

**Bulk RNA-seq analysis.** The raw sequencing fastq files underwent quality and adapter trimming using trimgalore v0.4.4. The trimmed reads were then mapped to the hg38 human reference genome using STAR v2.7.10a[88]. The abundance of transcripts and genes was estimated from the aligned reads using RSEM v1.3.3[89]. RSEM provided quantification of gene expression metrics including expected read counts, TPM (transcripts per million), and FPKM (fragments per kilobase of transcript per million mapped reads). The expected read counts generated by RSEM were utilized by DESeq2 to identify significantly differentially expressed genes[83]. Genes were considered differentially expressed if they exhibited a $\log_2$ fold change > 1 or < -1, along with an *FDR* < 0.05 when comparing drug-treated and untreated samples. Enriched biological pathways of differentially expressed genes were identified using the GSEA and enricher functions (based on hypergeometric test) from the clusterProfiler R package[86] and gene sets used in the annotation were defined in BioPlanet 2019[87].

**Single cell RNA-seq analysis.** The patient information in this study are published data from Chapple, R. H. et al., 2024[90] and are included in Supplementary Data 7. We downloaded the count matrices of 16 human neuroblastoma-infiltrated bone marrow samples from the NCBI database, identified under GEO number GSE216176[48]. These matrices were integrated into a single dataset using Scanpy, filtering out cells where the percentage of mitochondrial gene read counts was greater than or equal to 10%. We then applied automatic consensus nonnegative matrix factorization (acNMF)91. (https://rchapple2.github.io/acNMF/) to factorize the gene-cell count matrices, resulting

in the clustering of cells into 52 distinct gene expression programs. These programs were annotated based on the expression of canonical adrenergic marker genes in neuroblastoma, as defined by Van Groningen et al[91]. We also accessed the annotated gene expression programs of 21 human primary neuroblastoma tumors (Dong dataset: $n = 16$, GSE137804; GOSH dataset: $n = 5$, EGAD00001008345)[46,47] and 17 neuroblastoma Patient-Derived Xenografts (PDX, GSE228957)[90,92], where human cancer cells were implanted in mice. These processed data were obtained from https://pscb.stjude.org/pscb/.

Additionally, we acquired the read count matrices for single-cell RNA-seq data from 13 neuroblastoma cell lines, the dataset was downloaded from GSE229224[90]. To process this data, we converted the read counts to Transcripts Per Million (TPM) for genes in each single cell. We then calculated the average TPM for all single cells within the same cell line type for each gene, utilizing a custom R script.

**Patient -*omics* data analysis**
Survival curves were generated using the Kaplan-Meier model implemented in the R package survival for a dataset comprising 786 samples (Cangelosi et al. 2020)[93]. The samples were classified into two groups based on their SMAD9 gene expression levels: the low expression group, which included samples ranked within the lower 20% of all samples, and the high expression group, which comprised the remaining samples. Specifically, the high-expression group consisted of 625 samples, while the low-expression group consisted of 156 samples. We employed a multivariate Cox proportional hazards model to analyze the association between overall survival and SMAD9 expression, taking into account additional factors MYCN status, age, and INSS tumor stage. This analysis was conducted using the coxph() function from the survival package in R.

**Animal experiments**
Neuroblastoma cell-line derived xenografts were created by injecting luciferase labeled BE(2)-C and CHP-134 cells into recipient athymic nude mice using a subcutaneous injection technique. All the mice were purchased from Charles River (strain code 553), female, and approximately 12 weeks in age. Mice were weighed and screened weekly by xenogen and the bioluminescence was measured. Mice were enrolled in the study after achieving a target bioluminescence signal of 107 photons/sec/cm2 or a palpable tumor, and chemotherapy was started the following Monday.

Mice were randomized to the following three treatment groups with five mice in each group:

Control, Irinotecan + Temozolomide, Retinoic Acid. The dosing schedule is included in Supplementary Data 10.

Allometric doses were administered on a clinically relevant schedule for 12 weeks (4 courses for Irinotecan + Temozolomide groups, dosage informed by previous pharmacokinetics modeling as part of the Childhood Solid Tumor Network (CSTN)). Xenogen and caliper measurements were obtained weekly throughout the study. Mice were removed from studied and considered to have progressive disease when the tumor burden was ~20% mouse body weight.

**Determining a pharmacologically relevant dosage for 13-cis retinoic acid in mice.** The typical dose of isotretinoin in the clinic for pediatric neuroblastoma is 160 mg/m2/day split into two oral doses (i.e. 80 mg/m2/dose)[94]. Using the recommended weights and body surface areas for children and mice from the FDA, along with an average of the two commonly recommended allometric exponents (0.67 and 0.75), yields a mouse equivalent dose of 25 mg/kg twice daily. Nolting et al.[95] used a similar approach to derive isotretinoin mouse dosing for their preclinical studies in which they used 53 mg/kg/day. Moreover, the pharmacokinetics (PK) of isotretinoin was studied in mice after a 10 mg/kg IV injection, yielding a plasma area under the concentration-time curve (AUC) of 30.9 hr-uM[96]. Assuming an absolute

oral bioavailability of 21%[97] and dose proportional pharmacokinetics, a 25 mg/kg oral dose would give a plasma AUC of 16.2 hr-uM. This compares well with the calculated, clinically achieved isotretinoin AUC reported by Veal et al.[94] of 12.9 hr-uM. In fact, exposures in mice at 25 mg/kg may be slightly higher, and more akin to the 200 mg/m2/day split administration dose. Overall, our selection of isotretinoin 25 mg/kg orally twice daily can be justified by two common approaches for deriving clinically relevant doses in mice – allometry and AUC/Cavg PK equivalency.

### Immunofluorescence assays on patient tumor and paired bone marrow samples

Tissues were fixed in 10% neutral buffered formalin, embedded in paraffin, sectioned at 4 μm, and mounted on positive charged glass slides (Superfrost Plus; 12-550-15, Thermo Fisher Scientific, Waltham, MA) that were dried at 60 °C for 20 minutes, deparaffinized, and used for immunofluorescence (IF). The antibodies used for the assays are included in Supplementary Data 10.

All staining was performed on Ventana Discovery Ultra autostainer (Roche, Indianapolis, IN). All tissue sections used for IF labeling underwent deparaffinization followed by heat-induced antigen retrieval using prediluted Cell Conditioning Solution 1 (ULTRA CC1, 950-224, Ventana Medical Systems, Indianapolis, IN) for 4 minutes. Primary antibodies were serially applied using the U DISCOVERY 5-Plex IF procedure and the following reagents and kits (all from Ventana Medical Systems, Roche): DISCOVERY inhibitor (760-4840), ready to use DISCOVERY OmniMap anti-Rb HRP (760-4311) for PHOX2B and Phospho-SMAD1/5/9 or rabbit anti-mouse secondary (Abcam, ab133469) followed by DISCOVERY OmniMap anti-Rb HRP (760-4311) for vimentin. Visualization of staining was the following: Phospho-SMAD1/5/9 with DISCOVERY Rhodamine 6 G kit (760-244), PHOX2B using a DISCOVERY Red 610 kit (760-245), and Vimentin using a DISCOVERY CY5 Kit (760-238). Coverslips were hand mounted on slides with Prolong Gold Antifade reagent containing DAPI (Thermo Fisher Scientific, P36935). Sections were digitized with a Zeiss AxioVision Scanner (Carl Zeiss Microscopy, LLC) and analyzed using HALO v3.2.1851.354 and the HighPlex FL v4.0 algorithm to segment and count immunopositive and immunonegative cellular subpopulations (both Indica Labs, Albuquerque, NM) and generate automated H-scores and density maps of positive and negative immunolabeling. The H-score was calculated with the following equation and resulted in a value between 0 and 300.

$$H-score = \sum(PI \times I)$$

Where:

PI = Percentage of cells stained at a specific staining Intensity level.
I = Intensity of staining, where 0 = no staining, 1 = weak staining, 2 = moderate staining, 3 = strong staining.

All patient information are provided in Supplementary Data 8. The patient samples were obtained from the Tissue Bank at St. Jude Children's Research Hospital and used under institutional review board (IRB) approval from St. Jude Children's Research Hospital. The patients weren't consented for this study, but all samples were consented for future unspecified research. No participant compensation was provided because this is non-human subjects research so there is no interaction or intervention with the subjects or their identifiable data or biospecimens for this study. Sex and/or gender was determined based on self-report and was not considered in the study design.

### Treatments with 13-cis-RA vs ATRA

We used ATRA for in vitro experiments and 13-cis-RA for our in vivo studies. This decision was carefully considered and is supported by existing literature as well as our own experimental data. Briefly, 13-cis-RA is widely recognized as a pro-drug that undergoes isomerization

and metabolic conversion to its active form, ATRA, in the liver and other tissues[98]. Other metabolites of 13-cis-RA, such as 4-oxo-13-cisRA, have also been shown to accumulate in patient plasma during pharmacokinetics studies[25], and has been argued to have activity in neuroblastoma cells in vitro[99], though conversion of 13-cis-RA to ATRA occurs predominantly within cells[100]. Given that 13-cis-RA requires metabolic activation to exert its effects on retinoic acid receptors (RARs), using it directly in cell culture could introduce variability due to differences in enzymatic activity or the absence of necessary metabolic enzymes. By treating neuroblastoma cell lines directly with ATRA, we eliminated the variable of enzymatic conversion, allowing us to quantify the direct effects of the active compound. This aligns with standard practices in pharmacological research, where active metabolites are often used for in vitro experiments to study direct cellular effects. To confirm that ATRA and 13-cis-RA have equivalent biological activities in our cell models, we treated several neuroblastoma cell lines (CHP-134, TGW, BE(2)-C, and BE(2)-M17) with both compounds at identical concentrations, observing no significant differences in antiproliferative effects between ATRA and 13-cis-RA (Supplementary Fig. S8C). For our in vivo mouse xenograft studies, we administered 13-cis-RA, aligning with its clinical use in neuroblastoma treatment due to its favorable pharmacokinetics[101]. To verify the in vivo conversion of 13-cis-RA to ATRA, we performed high-performance liquid chromatography (HPLC) analysis on tumors harvested from RA-treated mice. The analysis showed that the concentration of ATRA in the tumors were much higher than vehicle-treated mice and were similar to 13-cis-RA concentrations, confirming effective conversion to ATRA in vivo.

### Statistics and reproducibility

To ensure reproducibility, statistics in this study are all derived from three or more biological replicates. No statistical method was used to predetermine the sample size. No data were excluded from the analyses. The Investigators were not blinded to allocation during experiments and outcome assessment.

### Reporting summary

Further information on research design is available in the Nature Portfolio Reporting Summary linked to this article.

## Data availability

The DepMap and GDSC RNA-seq and drug response data were obtained from depmap.org and cancerrxgene.org respectively. The high-throughput RNA-seq and ChIP-seq data generated during this study were deposited in the GEO database under the accession numbers GSE253785 and GSE253786, respectively. Flow cytometry gating strategy is shown in Fig. S11. Western blots showing full-length PARP and total SMAD1/5/9 expression are included in Fig. S12. Replicates for the immunofluorescence images in Figs. 3A and 4A are shown in Fig. S13. Raw immunofluorescence images for the patient samples related to Fig. 6G - H are uploaded to Zenodo (https://zenodo.org/records/13905830) and the processed images are included in Fig. S14. Replicates and raw images for Western blots are shown in Source Data file, and the quantification of the Western blots is shown in Supplementary Data 9. Source data are provided with this paper.

## Code availability

The customized scripts developed for high-throughput data analysis in this study are available on GitHub (https://github.com/geeleherlab/BMP-signaling-mediated-apoptosis-and-senescence).

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

## Acknowledgements
P.G. is supported by an NIGMS R35 award [R35GM138293] an R01 grant from NCI [R01CA260060]; K99/R00 [R00HG009679] from NHGRI; P.G also receives support from ALSAC. The St. Jude Center for Advanced Genome Engineering receives support from NCI [P30CA021765]. Images were acquired at the Cell & Tissue Imaging Center which is supported by SJCRH and NCI [P30CA021765]. The authors thank Sarah K. August for assisting with assembling the text and scientific illustrations. The authors would also like to acknowledge the Comparative Pathology Core Laboratory at St. Jude Children's Research Hospital. The content is solely the responsibility of the authors and does not necessarily represent the official views of the National Institutes of Health.

## Author contributions

M.P. performed the experimental work. Y.Z., W.C.W., X.L., R.H.C., M.P., and P.G. performed computational work. P.G., M.P., and Y.Z. wrote the manuscript. P.G. and J.E. supervised the experimental work. B.P., B.J., and E.P. performed next-generation sequencing for RNA-seq and ChIP-seq, supervised by J.E.. D.C. and J.L. performed drug screening, with W.C.W. analyzing the data. J.A.S., J.P.C., A.J.L., and S.M.P.M generated the cell line with HA-tagged SMAD9 and assisted with CRISPR screening. M.L., G.E.C., S.K., and H.S. performed IF staining, imaging, and image analysis for patient samples. L.Y. performed HPLC for RA concentrations in tumor samples. B.J.A. assisted with ChIP-seq data analysis. B.F. and E.S. provided resources and performed mouse experiments. M.A.D. and T.C. provided resources, insights and supervised additional parts of the study.

## Competing interests
The authors declare no competing interests.
