## [Transparent Peer Review file · Nature Communications]

Bone morphogenetic protein (BMP) signaling determines neuroblastoma cell fate and sensitivity to retinoic acid.

Corresponding Author: Dr Paul Geeleher

Version 0:

Reviewer comments:

Reviewer #1

(Remarks to the Author)

Pan et al. present an interesting study about the role of retinoic acid in neuroblastoma therapy. They integrate data from multiple sources, including Genomics of Drug Sensitivity in Cancer (GDSC1) (Iorio et al., Cell, 2016), DepMap and summarize data from their recent work (Lee et al., Nature Communications, 2023) before performing a genome-wide CRISPR screen in sensitive neuroblastoma cell lines treated with RA. They then investigate several mechanisms downstream of retinoic acid, including high BMP/SMAD signaling and its role in apoptosis/senescence, while recapitulating early findings on its role in differentiating neuroblastoma cells. They contrast this with situations of low BMP signaling, where differentiation predominates. They then explore the ChIP-seq binding profiles of SMAD4, SMAD9, and RARA in baseline and ATRA-treated cells in 3 cell lines, representing apoptosis, senescence, and differentiation. They explore the lack of efficacy of RA for mouse xenografts of neuroblastoma by performing RNA-seq, finding that BMP signaling activity decreased dramatically in xenografts. Finally, they analyze published RNA-seq data and find high BMP signaling in metastases, and confirm these findings in paired primary and bone marrow biopsies in 6 neuroblastoma patients, also finding that BMP2/4/7 enhance apoptosis downstream of ATRA in vitro. One exciting aspect of the study is that it highlights the limitations of purely in vitro gene/drug screens by showing that highly RA-sensitive lines act very differently when xenografted in vivo. Much of the field thinks of xenografts into nude mice as purely a furry test tube and this study provides a compelling example of where that is not the case.

Overall, the authors present a compelling study suited to the broad readership of Nature Communications. However, there are a few details and experimental controls missing, summarized below:

Major comments

- Western blots should show both the unmodified and modified protein. For example, western blots showing cleaved PARP1 are missing the uncleaved PARP1 band (e.g. Fig 1C, etc.) - both should be shown. Similarly, the blots showing pSMAD (Fig 2D, Fig S1G, etc) should also show the unphosphorylated protein.
- There are contradictory statements on the CRISPR screen, such as: "As expected, the knockout of retinoic acid receptor RARA was the strongest ATRA resistance hit [...] enrichment ACVR1 similar to that of RARA knockout (2.2 vs 1.5 log2 fold change for ACVR1 vs RARA respectively)". Could the authors clarify how RARA is the strongest hit but that ACVR1 is stronger (2.2 > 1.5)? There is a need for more precise language than "strongest hit". Strongest how? Fold-change of top gRNA? Fold-change of median gRNA? P-value? Something else?
- Given the emphasis on the CRISPR screen, the Methods given for CRISPR screen analysis ("Results were analyzed using MAGeCK ver. 0.5.7 (Li, Xu et al. 2014)") are inadequate. Full details of preprocessing, normalization, replicate correlation, gRNA ranking and p-value computation should be provided. The main figures for the CRISPR screen should plot median/mean enrichment/depletion or RRA p-value (with appropriate multiple hypothesis testing correction); "RRA score" is not interpretable without more complete method details.
- No QC plots are provided for the CRISPR screen. At a minimum, replicate and sample correlation should be shown. The authors might consider looking at published CRISPR screens to find established methods to demonstrate that the screen is of high-quality (e.g. essential gene dropout).
- The authors move from their genome-wide CRISPR screen to analyzing a previous published CRISPR screen targeting 655 genes (Lee, Wright 2023). Why are these 655 genes important (esp. in light of the results previously presented)? Does this previous screen include among the 655 genes many of the same top-ranked genes from the current work? It would be best if the targeted screen capitalized on the discovered top genes from the genome-wide screen but, absent such a

justification, the authors must provide some rational basis for these 655 genes being chosen for screening in more NB cell lines.

- There are several quantifications missing:
 - o Quantification of blot results and immunofluorescence from the "three independent experiments" should be shown in addition to the representative blot/images.
 - o It is unclear if there are replicates of the CRISPR screens. If so, correlation between replicates should be reported. If not, replicates should be performed.
 - o Many results missing statistics, for example the results presented in Supp Fig 2, others. Sometimes it is unclear what statistical test was performed.

Minor comments

- "unbiased genome-wide CRISPR knockout screen" should just be "genome-wide CRISPR knockout screen". No biological technique is unbiased. In this case: The authors did not design the CRISPR library nor did they systematically measure knock-out efficiency for each gRNA (which certainly differs).
- "hijacking" of a normal developmental process has been described since the very early studies with retinoic acid, where treatment of acute promyelocytic leukemia led to differentiation of leukemic cells into macrophages and neutrophils
- There was not an insignificant amount of apoptotic cells in DMSO treated cells (Supp Fig 1B)
- Additional differentiation markers should be evaluated
- Some experimental/analysis details are missing. What kind of replicates in 4A? How many replicates of bulk RNA-seq? How were single cell counts converted to TPM - is each dot a cell in 5F, Supp Fig 9E-M or how was this analysis done? How old were the mice, what was the sex, and how often were they weighed?
- Addgene plasmid numbers are missing for some plasmids (e.g. lentiCas9-Blast)
- Minor typographical and citation errors
 - o Order of figure panels in text (1E then 1D), Fig 2K should be 2B
 - o "extracted" should be "extracted", "AVCR1" is "ACVR1"
 - o Cangelosi et al. 2020 was used but not cited

Reviewer #2

(Remarks to the Author)

In this study, Pan & Zhang et al. investigated the effects of ATRA in neuroblastoma cells. The authors employed a comprehensive range of methods to identify and examine the impact of BMP signaling on ATRA-induced differentiation and apoptosis/senescence.

We have some comments:

- In the manuscript, Genomics of Drug Sensitivity in Cancer (GDSC1) by Iorio, Knijnenburg et al. 2016 was referenced and NB-13 was pointed out as a retinoic acid sensitive cell line with IC50 of 0.26 μ M. The authors then moved onto analyze 16 cell lines themselves by treating cell lines with ATRA for 6 days, which they state as a more clinically relevant treatment duration. According to Supplementary Figure 1A and Supplementary Table 1, under their 6 days treatment protocol IC50 values of NB-13, CHP-154 and D283 Med are 287.7, 2.507 and 4.399 nM respectively. In the following parts of the manuscript, NB-13, CHP-134 and D283 Med are treated with 2 μ M ATRA, which is 2-3 orders of magnitude higher than the calculated IC50 values. Could this effect or explain the induction of apoptosis over differentiation? Does isotretinoin have a comparable effect? Some experiments could be added testing lower ATRA concentrations.

- pSMAD 1/5/9 protein levels in Western blots are shown as an indicator of activated BMP signaling. To interpret phosphoprotein levels total SMAD1/5/9 levels should have been assessed in parallel.

- The same applies to the analysis of c-PARP as a marker for apoptosis induction. Total PARP levels should be included.

- Although FK506 and rapamycin can enhance BMP signaling activity by tethering FKBP1A to calcineurin and by tethering FKBP1A to mTOR, respectively, both are known to have a broad spectrum of effects. For instance, mTOR inhibitors have been demonstrated to synergize with ATRA in AML, as evidenced by their impact on MYC. This could be discussed.

- To analyze interactions between BMP and RA signaling transcription factors, the authors performed ChIP-seq binding profiles of SMAD4, SMAD9, and RARA baseline and ATRA-treated cells. Their data indicate high amount of overlaps between peaks (such as RARA peaks overlapping 87% of SMAD9 peaks and 98% of SMAD9 peaks overlapping with SMAD4 peaks). In our hands, RARA ChIPs are usually of very low quality, although we never tested the here used antibody. Was the specificity of the antibodies assessed and blacklisted known phantom peaks excluded? One could also assess canonical binding motives or known targets.

- It has been previously shown that fetal calf serum contains high amounts of BMPs. (<https://doi.org/10.1186/1471-2121-10-20>.) Could this be skewing the results authors get in vitro towards a "bone marrow metastasis" like phenotype and the reason why in vitro dynamics cannot be replicated in primary tumor sites?

- In experiments that involve viral transduction (Such as Figure 2K) comparisons are made between control groups, there is no information regarding how empty controls compare to parental cells.

- Considering the author's aim to explain the RA's lack of effect on primary human tumors and its efficacy in eliminating neuroblastoma cells from the bone marrow during post-chemo maintenance, co-culture systems between bone marrow stromal / immune cells and neuroblastoma cells would allow authors to enlighten the effect of bone marrow microenvironment on the apoptosis / differentiation dynamics in neuroblastoma cells.

- In Supplementary Figure 3B what kind of treatment DMSO Day 0 represent?

- In Figure D, the authors show Enrichr clustergram of CRISPR screen results and state that the input is top and bottom 1% genes. The figure could be more clear on which genes are top and bottom.
- In order to confirm their high throughput CRISPR screen, the authors knocked down the BMP type I receptor ACVR1, with three independent shRNAs, and consistent with the screen. When describing their following approach of using selective BMP inhibitor K02288, they state that “due to functional redundancy, a single gene knockout is usually not sufficient to fully inactivate BMP signaling”. While the statement itself might be true, it causes a confusion between CRISPR KO and the technique they actually used (shRNA mediated knockdown).

Reviewer #3

(Remarks to the Author)

Reviewer #4

(Remarks to the Author)

Pan et al., demonstrated that retinoic acid (RA) treatment can induce apoptosis or senescence in neuroblastoma cells. Through a genome-wide CRISPR screen, they identified the bone morphogenetic protein (BMP) signaling pathway as a key regulator of RA-induced apoptosis in these cells. Furthermore, they observed that neuroblastoma cells derived from bone marrow exhibit heightened BMP activity, offering a novel perspective on the mechanisms of action (MoA) of RA in the clinical treatment of neuroblastoma patients.

Major points:

1. The authors observed that certain pediatric cancer cell lines exhibit exceptionally low IC50 values in response to RA treatment, as shown in Figure 1A. It would be beneficial to include clinical data on these cell lines, such as the presence of driver mutations and potential mutations in BMP signaling pathway components.
2. The authors employed K02288 to inhibit BMP signaling in neuroblastoma cells, reporting that BMP activity became undetectable after three days of treatment. However, phosphorylation of SMAD1/5/8 typically peaks within 15 minutes, with downstream targets activated within 2 hours in the presence of BMP ligands. Standard BMP signaling pathway inhibitors also block SMAD1/5/8 phosphorylation within hours, not days. Therefore, to more accurately assess the inhibition of BMP signaling activity, a shorter treatment duration should be used instead of the three-day period implemented in this study.
3. The authors assert that BMP signaling is essential for retinoic acid (RA)-induced apoptosis in neuroblastoma (NB) cells. They demonstrate that inhibition of BMP signaling with K02288 reduces NB cell sensitivity to RA, as illustrated in Figure 1F, and prevents the increase in cleaved PARP levels induced by ATRA, shown in Figure 1G. This observation is noteworthy. The authors should further investigate whether RA treatment activates BMP signaling or coordinates with BMP signaling pathways to facilitate apoptosis in NB cells.
4. FKBP1A is shown to inhibit not only BMP signaling but also TGF β signaling. Consequently, FK506 or rapamycin, which act as FKBP1A inhibitors, enhance both BMP and TGF β signaling. Therefore, in Figure 2D-2H, the author should exercise caution in concluding that BMP signaling is the sole target of these small molecules.
5. The authors utilized SMAD9 knockout (KO) to investigate neuroblastoma (NB) cell differentiation in the context of BMP signaling inhibition, as shown in Figures 3G and S3D. First, the authors should provide evidence of the extent of SMAD9 KO to confirm its effectiveness, given that SMAD9 expression levels are typically low in human cells. Additionally, to achieve more comprehensive inhibition of BMP signaling, it is recommended that the authors also conduct SMAD1/5 KO instead of single gene KO (SMAD1 KO). Alternatively, the authors should present data demonstrating the inhibition of BMP downstream targets following SMAD9 knockout.
6. The authors conducted chromatin immunoprecipitation sequencing (ChIP-seq) to assess the genomic binding patterns of RARA, SMAD4, and SMAD9 in neuroblastoma (NB) cell lines following all-trans retinoic acid (ATRA) treatment. Interestingly, the analysis revealed a significant increase in SMAD4 binding across the genome post-ATRA treatment, contrasting with a slight decrease observed in SMAD9 binding. This discrepancy warrants further elucidation from the authors to provide a comprehensive understanding of the differential regulatory roles of SMAD4 and SMAD9 in response to ATRA.
6. The genome-wide increase in SMAD4 binding does not account for the decreased expression of ID genes following ATRA treatment from the perspective of BMP signaling. The authors should differentiate between the activation or inhibition of BMP and TGF- β signaling pathways. TGF- β signaling is known to inhibit ID gene expression, whereas BMP signaling promotes the expression of these genes in certain cellular contexts. Therefore, the GSEA analysis in Figure 5C should focus specifically on BMP signaling rather than the broader TGF- β superfamily.

7. Transcriptional regulation in BMP signaling typically occurs within hours. In the present study, the authors assessed gene expression changes over several days, which may primarily reflect secondary effects. It is strongly recommended that the authors investigate gene expression alterations over shorter time intervals, such as 1 hour or 4 hours, to capture the more immediate effects of BMP signaling.

Minor points:

1. In Figure 1B, the proportion of Annexin V+/PI- cells decreases on Day 6 in the D283 Med cell line following ATRA treatment, which contrasts with the observations in the other two cell lines. The authors should provide an explanation for this phenomenon.
2. The authors conducted a Western blot analysis to assess cleaved PARP expression following ATRA treatment in two neuroblastoma cell lines and a medulloblastoma cell line, as depicted in Figure 1C. However, it is noteworthy that the loading controls used differ between the neuroblastoma cell lines (β -Actin) and the D283-Med cell line (GAPDH). The authors should provide an explanation for this.
3. The legend of Fig. 1D should indicate the meaning of the panel color (the red color indicates the screen hits?).
4. The authors conducted an unbiased genome-wide CRISPR screen using a neuroblastoma cell line to compare the enrichment and depletion of gene knockouts in cells treated with ATRA for six days versus mock-treated cells. It is recommended that the authors include a schematic diagram to illustrate this experimental strategy.
5. Line 131: SMAD4 and SMAD9 are not downstream targets of BMP signaling. The authors should revise this description accordingly.
6. Figure S2B does not specify the units for the x-axis. Does "time" refer to years? The authors should clarify this in the figure legend.
7. Line 193: If the authors intend to describe all SMAD proteins involved in the BMP signaling pathway, they should also analyze the correlation of Co-SMAD (SMAD4) and I-SMAD (SMAD6/7) with the ATRA drug response. Or the authors want to emphasize the R-SMADs (SMAD1/5/8), they should correct the statements of "BMP SMADs".
8. The authors should use total SMAD1/5/9 as the loading control for detecting pSMAD1/5/9 levels in Figures S1G, 2D, S5I, and 6J.
9. Lines 226-228: The authors demonstrated that FK506 and rapamycin both enhance BMP signaling, contributing to their synergistic effects with ATRA. To further elucidate this, the authors should conduct RNA-seq or qPCR analyses targeting BMP signaling pathways in neuroblastoma (NB) cell lines treated with FK506 + RA and rapamycin + RA. This would allow for a comparative assessment of the shared effects of these treatments on BMP signaling.
10. Line 288-290: The authors demonstrated that the combination of ATRA and K02288 induced differentiation more quickly than K02288 alone in Fig. 3B. They should perform a time-course western blot to check the induction of differentiation between the combination and K02288 alone.
11. The gene symbols should be italic in Fig. 5G.
12. Compared to the DMSO control condition, SMAD9 binding at TLX2 and ID1 appears to increase after 1 day of ATRA treatment, followed by a subsequent decrease over time, as depicted in Figure S4F. The authors should provide an explanation for this dynamic change in BMP signaling activity.
14. In Figure 6F, the authors used the mean expression levels of ID genes to assess BMP signaling activity. However, given that ID1, ID2, and ID3 may exhibit extreme variations in expression across different cells or tissues, calculating mean values (ID1 + ID2 + ID3) may not accurately reflect BMP signaling activity. It is recommended that the expression levels of ID1, ID2, and ID3 be compared individually to provide a more precise analysis.

Reviewer #5

(Remarks to the Author)

Pan et al. report the mechanism of retinoic acid in eliminating post-chemotherapy neuroblastoma cells. In a previous publication, Iorio et al. showed that two neuroblastoma cell lines are >100-fold sensitive to all-trans-retinoic acid compared to other cell lines in in vitro activity testing. In repeat experiments of six-day incubation with ATRA, the authors found two sensitive cell lines were apoptotic. CRISPR knockout screen showed that BMP signaling was one of the enriched pathways associated with ATRA sensitivity. Using BMP inhibitors and knock-down experiments, they showed that BMP contributes to ATRA-induced apoptosis. They also reported that BMP inhibition induced differentiation of neuroblastoma cells by ATRA. However, in the cell-line xenograft model of the same cell line that was utilized for the in vitro studies, the BMP signaling pathway was not enriched. In investigating the reason, the authors reported that the bone marrow stromal cells obtained from patients have high BMP protein expression. Therefore, the ATRA effect in eliminating neuroblastoma cells in bone marrow is due to the tumor microenvironment with high BMP expression. There are major and minor comments on the manuscript to

improve the quality of the study.

1. The authors utilized all-trans-retinoic acid instead of 13-cis retinoic acid for no clear reasons. Although they are enantiomers, they have different metabolic profiles. Also, to make the experiments clinically relevant, 13-cis retinoic acid should have been used. Alternatively, if the use of ATRA was inevitable for experimental convenience, a few key additional experiments should have been included to demonstrate that they are biologically equivalent.
2. It is stated that CHP-134 is more sensitive than NB13 to ATRA, meaning sub micromolar IC50. However, Figure 1F shows that both 3-day- and 6-day-incubation show $>1 \mu\text{M}$.
3. What is the common gene mutational status of the 16 cell lines (or the 12 neuroblastoma cell lines) that were used for the project? SMAD9 is also regulated by MYCN, PHOX2B, etc. Some cell lines responding better to ATRA than others can be also due to several other reasons. It may better explain if other clinical and genetic features of the cell lines are considered. For example, are any of the cell lines retinoic acid naive? What is the status of common mutations of neuroblastoma?
4. Apoptosis by ATRA was only observed in vitro, but not in xenograft tumors even at clinically relevant concentrations of ATRA, and apoptosis in metastatic neuroblastoma by ATRA was not shown, which is the key experiment for the project. 1) Therefore, the title that BMP signaling determines neuroblastoma cell fate and sensitivity to retinoic acid is limited to cell culture and largely speculative in clinical/in vivo settings. 2) Would this be related to the oxygen tension that was used in cell culture? Retinoic acid analogs are known to have different activity depending on oxygen tension (20% vs hypoxia). The apoptotic effect shown in 20% oxygen may diminish when the cells are cultured in hypoxia. This issue is worth investigating, especially based on the data shown in Figure 6F.
5. Figure 2: SMAD9, not other SMADs, was highly correlated with ATRA IC50. However, the scope of the investigation includes SMAD4 in Figure 5 without a plausible explanation (it showed in ChiP-seq, and what is the explanation?). And the summary illustration includes SMAD1/5/9 for apoptosis, but SMAD1 data has not been included. Also, there is no evidence that SMAD4 contributed to apoptosis in response to ATRA.
6. Figure 6: several types of primary samples, cultured cell lines and xenograft models of neuroblastoma were investigated. However, again their genetic and clinical features are not taken into consideration.
7. Line 355 – 356: it is stated that the authors could rule out differentiation, but it is unclear how this was ruled out.

Reviewer #6

(Remarks to the Author)

- What are the noteworthy results?

The manuscript by Pan, Zhang et al. provides experimental evidence of a previously undescribed mechanism-of-action for retinoic acid (RA) maintenance therapy in neuroblastoma (NB). Their data show that despite differentiation occurring in tumor cells subjected to RA (as previously known), induction of apoptosis and/or senescence by RA might be the main cause of its anti-neoplastic effect in specific sites. Moreover, the dependence of the RA effect on BMP signaling provides a plausible explanation for the site-specific effect of RA therapy on metastatic NB cells in the bone-marrow.

- Will the work be of significance to the field and related fields? How does it compare to the established literature? If the work is not original, please provide relevant references.

The results are important in the neuroblastoma field where the effect of long-term RA treatment has been questioned lately. The present data indicates an effect of the treatment on minimal residual disease with potential removal of metastatic tumor cells from the bone marrow. This is of immense importance and might lead to RA treatment being offered to more patients at risk of relapse. Additionally, knowledge of the mechanism-of-action can potentially lead to new combined therapies as suggested in the manuscript by combining senescence inducing RA with immunotherapy targeting senescent cells. The authors suggest that “developmental hijacking” of other naturally occurring processes might be present also in other types of cancer, in particular other developmental tumors of the childhood. This could be true, and it is also possible that other tumors with difficult to treat bone/bone marrow metastasis could benefit from the knowledge presented in this manuscript, e.g. breast cancer and prostate cancer. However, that is not experimentally tested in the current manuscript.

- Does the work support the conclusions and claims, or is additional evidence needed?

The data supports the main conclusions of the paper, i.e. response to RA is dependent on BMP signaling and high BMP expression at specific sites such as in the bone marrow can lead to apoptosis/senescence (not differentiation) in neuroblastoma cells in those specific sites. However, more data would be needed to explain the underlying mechanism of preference for specific tumor cell lines to undergo either apoptosis, senescence or differentiation even in the presence of high BMP. Also, some important aspects of limitations to the study and how this affects the generalizability and clinical application of the results should be discussed more thoroughly.

- Are there any flaws in the data analysis, interpretation and conclusions?

See major and minor comments below.

- Do these prohibit publication or require revision?

Revision required.

- Is the methodology sound? Does the work meet the expected standards in your field?

Adequate controls and number of repeats.

Some of the major experiments were only performed in one or two cell lines (for example CRISPR screen). Confirmatory experiments and data analyses were performed in additional cell lines and published data, but see comment below on generalizability of the results.

More details are needed for animal experiments, such as tumor growth curves, weight of mice etc.

- Is there enough detail provided in the methods for the work to be reproduced?

The method section is detailed and extensive except for the in vivo section.

Major considerations

- The current debate of the efficacy of RA use for maintenance therapy should be acknowledged as well as the very few clinical trials investigating RA treatment in patient cohorts (eg. Peinemann et al. *Cochrane Database Syst Rev.* 2017 Aug 25;8(8):CD010685. doi: 10.1002/14651858.CD010685.pub3.)

- Row 99-101) I disagree that hijacking of normal development processes have not been described as anti-cancer mechanisms before, one example is EMT (epithelial to mesenchymal transition).

- Choice of cell lines: the molecular and clinical characteristics of the cell lines should be presented in a table (for example adding this to Table 1). This is important since the sensitivity to RA is very different between cell lines. The authors should also discuss the possibility of confounding biological factors related to RA sensitivity.

- Only RA-treatment sensitive cell lines are chosen for further work: NB13, CHP-134, D283 Med, SK-N-SH, TGW etc. It should be clearly stated that this is a choice and that this is a limiting factor when it comes to the generalizability of the results. Have the authors tried also less sensitive cell lines? How would they behave in the rest of the experiments? For example, what happens with them if exhibited to ATRA + FK506? Is it still only resulting in differentiation in those cases? Has it been tested?

- Synergy: Figure 2G and S2K/L display synergy scores for RA + FK506 or Rapamycin. It is stated that ATRA showed "synergistic effects with both compounds in all NB cell lines" with one exception. It is not clear to me how SK-N-AS and GI-ME-N treated with FK506 can be synergistic, a cut-off of what is considered synergistic scores should be added. Also in figure S2K, L, O, P, the cut-off is not clear. The additive effect is shown with a red dotted line, but some statistical testing will be necessary to claim them to be synergistic.

- Fig 4: row 356) ATRA can sometimes induce cell cycle arrest which is indicated by the CRISPR screen. However, how can you "rule out differentiation" when you have just showed that in 4A and B? Here is a logical step missing that needs to be filled or explained better.

- Chip seq, why use BE(2)-M17 cells here? Are they used as diff cells anywhere else? Again, characteristics of the different cell lines and motivation (including supporting data) should be included in a table and explained. S4 should be more detailed in calling.

- Fig 5K, the conceptual overview should be simplified to better convey the hypothesis. Also, in figure 5 there is no data relating to the senescence (TGW) or differentiation (BE(2)-M17) cell results. I think the massive S4-S7 should be reorganized and some data added from them to the main figure to support the senescence and differentiation hypotheses.

- Importantly: to the best of my understanding the analysis of chip-seq data does not provide an explanation to WHY the three different cell lines get different downstream binding patterns from SMAD4 activation. This should be further investigated or discussed as a limitation. Now it is only referred to in the sentence on row 497-499 "...active SMAD transcription factors activates apoptosis or senescence, depending on baseline binding patterns of BMP effector SMADs..."

- In vivo study: provide a rationale for including a chemotherapy treatment group. As far as I can see no results except survival is presented for this group.

- PHOX2B is used as a general NB marker in analyses of patient tumor material (fig 6). From a developmental perspective, PHOX2B is only expressed in cells of the sympathoadrenal lineage (for example Zeinelden et al. 2022, Bedoya-Reina et al. 2021). Thus, in many publications on NB cell states, PHOX2B has been used as an adrenergic marker, e.g. has been considered not to be expressed in NB cells of a more immature/mesenchymal phenotype (Boeva et al. 2017, van Groningen et al. 2017, Patel et al. 2024 preprint). Other publications have shown that treatment resistant cells can be of the more mesenchymal phenotype (Westerhout et al. 2021, Manas et al. 2022). The presence of downstream BMP targets in non-PHOX2B cells could thus be of major clinical importance. Has this been investigated? Could an additional NB cell marker be used?

- A paragraph stating the limitations of the study should be included in the discussion: limited number of cell lines, no patient-derived cells, no testing in clinical cohorts to investigate the level of expression of BMP in primary tumor sites and metastatic sites in patients etc.

- An overview of BMP expression in different organs would also strengthen the clinical application suggestions. How is the BMP expression levels in other common metastatic sites? Liver, lung etc. BMP (e.g. BMP2 and 4) could be investigated in

clinical cohorts, in a TMA for protein expression or RNAseq data for example.

- The study, and particularly the clinical application possibilities, would be strengthened by an in vivo study of a metastatic model. Or by injecting cells both sub-cutaneous/orthotopic and in mouse femur to see differences in effect in vivo at different sites. For clinical application purposes the data from PDX models under these circumstances would be informative for how heterogeneous tumors behave. Possibly a spatial transcriptomic readout could inform on differential response in different subgroups of NB cells.

Minor considerations

- Row 82) A 2012-study should not be called "recent".
- 113) Previous 3-day exposure in literature is replaced in your study by 6-days exposure motivated by "in vivo pharmacokinetic profile". How is this relevant for your in vitro exposures?
- Table S1: There are 18 cell lines here when only 16 are in the results. It should be clarified somewhere that the last two are used in specific experiments only.
- 115-117) Discussion of differentiation and the lack of differentiation after 3 days (why not 6 days here?) already in results section 1, is called in figures S3A-B. This should be presented already in Fig 1/S1 if the lack of differentiation is used as motivation for investigating apoptosis.
- 127) Figure 1E and 1D are called out of order.
- 188) Fig S2A wrongly called. It is again called wrongly in row 192. Consider the correct way to refer to this figure.
- 191) Fig 2K wrongly called.
- 444) Fig 5D should be called instead of 5C
- 491) Fig S7F should be called instead of S8F
- 571) Provide a reference or data supporting "... because BMP signaling is artificially inflated in vitro." Evidence that it is higher is provided, but according to table 1 no additional BMP is added in the medium. What makes BMP signaling high in vitro?
- Fig 6: reorganize order of photos in H.

Version 1:

Reviewer comments:

Reviewer #1

(Remarks to the Author)

The authors have addressed my comments in the revised manuscript.

Reviewer #2

(Remarks to the Author)

In this revised version of the manuscript, the authors significantly improved the coherence of the study. The connections between the topics addressed in the manuscript was more comprehensively handled and the limitations of the study were adequately pointed out and discussed. Our comments were thoroughly and satisfactorily addressed.

Suggested minor corrections:

*Minor grammatical and narrative errors should be corrected. Some examples:

-Line 126: After removal of "unbiased" the article should be "a"

-Line 212: "as a negative regulator" when mentioning plural amount of inhibitors

-Line 369: Switching from past tense to present tense ("validate")

-The treatment periods were indicated as "six days / three days" or "6 days / 3 days" in different parts of the manuscript, it should be addressed either as written or number symbols

-

*Particularly in the Introduction, Results and Discussion sections, some points were addressed with sentences that were 4-5 lines long. Although this does not affect the results, it reduces the clarity of the work.

Reviewer #3

(Remarks to the Author)

Reviewer #4

(Remarks to the Author)

Overall, the authors have addressed the majority of my concerns. However, I believe there are two critical points that require further attention prior to publication:

The authors assert that RA treatment decreases BMP signaling activity. However, the SMAD4 binding signal at the genome is markedly elevated post-RA treatment, while changes in SMAD9 binding appear to be more subtle. Given the increased nuclear SMAD4 signals following RA treatment, this observation does not support the conclusion that RA diminishes BMP signaling. To clarify this discrepancy, the authors should perform SMAD1/5 and SMAD2/3 ChIP-seq analyses to further investigate the increased SMAD4 chromatin binding.

In Figure 6J, the internal control should be total SMAD1/5/9, rather than GAPDH. Total SMAD1/5/9 protein levels are the established standard for assessing phosphorylated SMAD1/5/9 changes in BMP signaling research. While the authors state they cannot draw conclusions based on total SMAD1/5/9 in their experimental context, they should at least provide a more thorough discussion of this limitation within their study.

Additionally, the phosphorylation of SMAD1/5/9 depicted in Figure 6J does not show a significant increase following BMP1/7 treatment compared to the vehicle control. The authors should offer further explanation regarding this observation.

Reviewer #5

(Remarks to the Author)

I appreciate that the authors tried to address all comments from this reviewer. However, using ATRA as an active metabolite of 13-cisRA is misplaced and not clinically irrelevant to the treatment of high-risk neuroblastoma. Multiple clinical pharmacokinetic studies of 13-cisRA in neuroblastoma reported that 4-oxo-13-cisRA is the major metabolite in patients. Also, ATRA is detected at negligible levels. Please refer to Veal GJ, Cole M, Errington J, Pearson AD, Foot AB, Whyman G, Boddy AV; UKCCSG Pharmacology Working Group. Pharmacokinetics and metabolism of 13-cis-retinoic acid (isotretinoin) in children with high-risk neuroblastoma - a study of the United Kingdom Children's Cancer Study Group. *Br J Cancer*. 2007 Feb 12;96(3):424-31. doi: 10.1038/sj.bjc.6603554. Epub 2007 Jan 16. PMID: 17224928; PMCID: PMC2360017. Therefore, the translational implications of this study remain in question.

Reviewer #6

(Remarks to the Author)

I believe that the authors have responded to and clarified most of my concerns regarding the manuscript. I appreciate this and think that the good manuscript has become even better.

In response to my comment #5 about drug synergy, the authors have now included a definition of synergy (vs additivity) and provided results from statistical testing. This could be sufficient, but I wonder why a well established synergy score such as Bliss independence or Zip synergy score was applied? But this is up to the authors.

However, my comment regarding the details of the animal studies has been missed or disregarded. I believe that more data is needed to support the conclusions drawn from the animal models:

"- Is the methodology sound? Does the work meet the expected standards in your field?
More details are needed for animal experiments, such as tumor growth curves, weight of mice etc.

- Is there enough detail provided in the methods for the work to be reproduced?
The method section is detailed and extensive except for the in vivo section."

Some additional information about the mice has been added in Methods 1146-47 in response to another reviewer question, but tumor growth curves and mice weight are still not included.

Version 2:

Reviewer comments:

Reviewer #4

(Remarks to the Author)

SMAD9 levels suggest complex crosstalk, necessitating ChIP-seq analysis for SMAD1/5 and SMAD2/3 to determine whether BMP or TGF- β signaling is predominantly affected. Concerns about BMP signaling activation remain unresolved due to potential confounding changes in total-SMAD1/5/9 levels, which must be disentangled from phospho-SMAD increases. Discrepancies in SMAD9 signal trends between figures (e.g., Fig. S4F vs. S5A) require careful review for consistency. Additionally, the Venn diagram in Fig. 5B needs correction or clarification regarding its proportional calculations.

Below are the detailed comments:

In my previous comments, I noted that RA treatment may interfere with the TGF- β superfamily, as evidenced by the significantly increased nuclear SMAD4 signal shown in Fig. 5A. While I acknowledge that changes in SMAD4 levels alone are insufficient to confirm BMP signaling activation, I find it perplexing that the authors report an increase in SMAD4 levels alongside a subtle decrease in SMAD9 levels. This apparent discrepancy complicates the interpretation of BMP signaling

inhibition following RA treatment. To address this issue, I recommend performing ChIP-seq for SMAD1/5 and SMAD2/3 to determine whether RA treatment predominantly impacts the BMP or TGF- β signaling pathways. Although this additional experiment may require more time, it would provide critical insights into the interaction between RA treatment and these signaling pathways.

My earlier concerns regarding total-SMAD levels specifically pertain to Fig. 2D. The authors previously stated, in their response to Reviewer #1 during the earlier revision, that they could not draw definitive conclusions. However, following FK506/Rapamycin treatment, the authors observed an increase in phospho-SMAD1/5/9 levels in TGW and SK-N-SH cell lines (as well as in CHP-134 cells treated with Rapamycin), accompanied by an increase in total-SMAD1/5/9 levels. To conclusively establish BMP signaling activation, it is essential to demonstrate that the observed increase in phospho-SMAD1/5/9 levels occurs independently of changes in total-SMAD1/5/9 levels. Without such confirmation, the claim of BMP signaling activation remains speculative.

The authors stated that "the change of SMAD9 signal in our data following RA treatment is quite substantial; the decrease is striking (>10-fold) after 3 days and 6 days of treatment (see, e.g., Fig. S4F)." However, Fig. S5A appears to display a significant increase in total SMAD9 peaks after 3 days of RA treatment. I recommend carefully reviewing the heatmap in Fig. S5A to ensure consistency with the described findings.

Finally, the proportions presented in the Venn diagram in Fig. 5B do not sum to 100%. The authors should address this inconsistency or provide additional clarification regarding the methodology used to calculate these proportions.

Reviewer #5

(Remarks to the Author)

Overall, the authors' response is acceptable to the specific comment. However, there are factors to consider when utilizing ATRA vs 13-cisRA in preclinical studies: 1) why only ATRA causes pseudotumor cerebri, 2) as demonstrated, some part of 13-cisRA changes to ATRA in cells when used in in vitro settings, which was not demonstrated in humans yet. The accumulation of 13-cisRA or ATRA in keratinocytes or in sebocytes is anticipated in high concentrations as the side effects of ATRA in skin or glands are well documented, but this cannot be generalized in cancer cells. Either ways, what is it that actually show the biological activity of 13-cisRA in neuroblastoma tumor is yet to be determined, 3) 4-oxo-13-cisRA has comparable biological activity to 13-cisRA in cultured neuroblastoma cells (PMID: 25039756). Does this mean 4-oxo-13-cisRA transforms to 13-cisRA or ATRA intracellularly? This is highly unlikely as the formation of 4-oxo-13-cisRA in humans is the result of enzymatic metabolism.

Reviewer #6

(Remarks to the Author)

My questions have now been answered by the authors.

Reviewer #7

(Remarks to the Author)

With regard to the comments of Reviewer #4 that the authors have already responded to and to the current ones.

The reviewer states that the authors assert that RA treatment decreases BMP signaling activity. This is not my interpretation of what they are saying with regard to the data in Figure 5 where the authors are looking at 1 day of ATRA treatment. The authors' interpretation is that they see strongly overlapping binding profiles between RARA, SMAD4 and SMAD9 in untreated cells with the genes regulated by these transcription factors being those involved in apoptosis. After 1 day treatment with ATRA they see an increase in RARA binding at a subset of the baseline SMAD4 peaks. It is true that the intensity of the SMAD4 peaks increases in the heatmap which they do not really explain, but the SMAD9 peaks do not change. They do not say that short term treatment with ATRA decreases BMP signaling as Reviewer #4 seems to imply. At longer treatments with ATRA, 90% of the cells die and the cells that are left, show a differentiation phenotype. In this case, levels of PSMAD1/5/9 are substantially lower (Fig S5I) meaning that BMP signaling has been reduced at this later time points. The authors clearly see a decrease in SMAD9 occupancy at the classic BMP target genes ID1 and also TLX2 and SKIL at 3 and 6 days after ATRA treatment.

Although the authors don't explain why the SMAD4 peaks increase in intensity after 1 day of ATRA, I do not think that it is necessary for the authors to perform SMAD2/3 and SMAD1/5 ChIP-Seq for the current study. In their most recent comments, the reviewer in still wants the authors to perform these ChIP-seq experiments. It would be interesting to see if TGF- β signalling was upregulated by 1 day of ATRA treatment and the authors could easily investigate this by a simple PSMAD2 and PSMAD3 Western blot, as well as PSMAD1/5/9 Western blot. A ChIP-seq is not necessary in my opinion.

The reviewer points out that SMAD1/5/9 should be used as a control in the Western blots and not just GAPDH. It is true that it is important to check that the PSMAD1/5/9 does not just increase/decrease because of changes in SMAD1/5/9 levels. They have done this for the blots that the reviewer refers to. I think it would be important to ask the authors to quantitate the blots and normalise the PSMAD1/5/9 levels to the SMAD1/5/9 levels. The GAPDH levels are also important as they indicate that the same amount of extract was loaded in each lane, assuming that they are the result of a reprobe of the same membrane. The values can also be normalised to the GAPDH levels.

With regard to the issue of the compatibility of the data for SMAD9 in Figures S4 and S5, I think the only thing that needs clarifying is that the >10 fold decrease in SMAD9 at day 3 and 6 is at particular genes, whilst the slight increase in SMAD9 occupancy at day 3 is on all targets. I don't think there is any discrepancy in the manuscript concerning this point.

The reviewer says that the proportions presented in the Venn diagram in Fig. 5B do not sum to 100%. According to my calculation, it adds up to 99.9%, so I think it is just a result of rounding up/down. They have presented the data as a percentage of total peaks (RARA, SMAD4 and SMAD9) in each case.

There is one further comment that I will make about the BMP side of the paper. The authors treat the cells with BMP1, but it should be noted that this is not a BMP ligand, but rather a protease. Thus, it is not surprising that it was inactive. (see https://en.wikipedia.org/wiki/Bone_morphogenetic_protein_1)

Version 3:

Reviewer comments:

Reviewer #7

(Remarks to the Author)

The authors have now successfully addressed all the points I raised and I have no further comments on the manuscript.

Dear peer Reviewers,

We thank the 6 Reviewers for their careful assessment of our manuscript. We have carefully addressed all comments point-by-point below. All changes to the manuscript are referenced by line number, which refer to the *tracked copy of the updated manuscript*. We have included tracked and clean copies of the updated manuscript and supplementary materials. We are extremely grateful to the Reviewers for their time and expertise and believe that the updated manuscript represents a significant improvement and that our broader overall conclusions are well supported by the collective data.

REVIEWER COMMENTS

Reviewer #1 (Remarks to the Author): expert in CRISPR screens, scRNA-seq

Reviewer #1 Overall summary:

Pan et al. present an interesting study about the role of retinoic acid in neuroblastoma therapy. They integrate data from multiple sources, including Genomics of Drug Sensitivity in Cancer (GDSC1) (Iorio et al., Cell, 2016), DepMap and summarize data from their recent work (Lee et al., Nature Communications, 2023) before performing a genome-wide CRISPR screen in sensitive neuroblastoma cell lines treated with RA. They then investigate several mechanisms downstream of retinoic acid, including high BMP/SMAD signaling and its role in apoptosis/senescence, while recapitulating early findings on its role in differentiating neuroblastoma cells. They contrast this with situations of low BMP signaling, where differentiation predominates. They then explore the CHIP-seq binding profiles of SMAD4, SMAD9, and RARA in baseline and ATRA-treated cells in 3 cell lines, representing apoptosis, senescence, and differentiation. They explore the lack of efficacy of RA for mouse xenografts of neuroblastoma by performing RNA-seq, finding that BMP signaling activity decreased dramatically in xenografts. Finally, they analyze published RNA-seq data and find high BMP signaling in metastases, and confirm these findings in paired primary and bone marrow biopsies in 6 neuroblastoma patients, also finding that BMP2/4/7 enhance apoptosis downstream of ATRTA in vitro. One exciting aspect of the study is that it highlights the limitations of purely in vitro gene/drug screens by showing that highly RA-sensitive lines act very differently when xenografted in vivo. Much of the field thinks of xenografts into nude mice as purely a furry test tube and this study provides a compelling example of where that is not the case.

Overall, the authors present a compelling study suited to the broad readership of Nature Communications. However, there are a few details and experimental controls missing, summarized below:

Response to overall Reviewer summary

We thank the Reviewer for their overall positive assessment of our study and provide a point-by-point response to their additional comments below.

Reviewer #1 Major comments

Reviewer #1 Comment #1

- Western blots should show both the unmodified and modified protein. For example, western blots showing cleaved PARP1 are missing the uncleaved PARP1 band (e.g. Fig 1C, etc.) - both should be shown. Similarly, the blots showing pSMAD (Fig 2D, Fig S1G, etc) should also show the unphosphorylated protein.

Response to Reviewer #1 Comment #1

We agree with the Reviewer that these Western blots could be valuable, though we emphasize that our overarching conclusions based on cPARP (apoptosis) and pSMAD (summarizing the overall levels of BMP signaling activity) were supported by multiple orthogonal assays. We have now included these additional Western blots in Supplementary Figure S12. We additionally note that our manuscript has not drawn any conclusions on the basis of total SMAD protein levels, which can be difficult to deconvolve from BMP signaling activity—especially in the longer-term treatments relevant to study the interaction with RA—because BMP SMAD transcription factors engage in autoregulatory feedback by binding to their own promoter regions, thereby modulating their transcriptional activity in a self-regulatory manner.

Reviewer #1 Comment #2

- There are contradictory statements on the CRISPR screen, such as: “As expected, the knockout of retinoic acid receptor RARA was the strongest ATRA resistance hit [...] enrichment ACVR1 similar to that of RARA knockout (2.2 vs 1.5 log₂ fold change for ACVR1 vs RARA respectively)”. Could the authors clarify how RARA is the strongest hit but that ACVR1 is stronger (2.2 > 1.5)? There is a need for more precise language than “strongest hit”. Strongest how? Fold-change of top gRNA? Fold-change of median gRNA? P-value? Something else?

Response to Reviewer #1 Comment #2

We agree with the Reviewer that this is confusing and was an oversight on our part. This has now been clarified where we state on lines 130 that “hits ranked by RRA scores from MAGeCK, see Methods” and on line 136-137 where we explicitly state the fold change is referring to the “estimated magnitude of ACVR1’s effect”.

Reviewer #1 Comment #3

- Given the emphasis on the CRISPR screen, the Methods given for CRISPR screen analysis (“Results were analyzed using MAGeCK ver. 0.5.7 (Li, Xu et al. 2014)”) are inadequate. Full

details of preprocessing, normalization, replicate correlation, gRNA ranking and p-value computation should be provided. The main figures for the CRISPR screen should plot median/mean enrichment/depletion or RRA p-value (with appropriate multiple hypothesis testing correction); “RRA score” is not interpretable without more complete method details.

Response to Reviewer #1 Comment #3

MAGeCK is the *de facto* end-to-end software tool used for analysis of CRISPR-drug screens and RRA scores at the standard, widely accepted and robust approach for ranking these hits (the tool has >2000 citations on Google Scholar). The preprocessing, normalization, gRNA ranking and p-value calculations are performed internally to the MAGeCK algorithm, which in our case was run with default parameters. We agree with the Reviewer that it would be helpful if we were more explicit about where to find these details so we now state on line 830-832 that “Results were analyzed using the *de facto* tool MAGeCK ver. 0.5.7 (Li, Xu et al. 2014) using the default parameters (see “Overview of the MAGeCK algorithm” section from Li et al for a description of the algorithm).” We also agree with the Reviewer that it would be helpful for us to include gRNA ranking statistics, which we have now included in additional tabs in our Table S2. This table also contains all of the relevant statistics, including gene and gRNA specific RRA scores, P-values, fold changes, FDRs, etc. We thank the Reviewer for helping us improve the presentation of our CRISPR screening results.

Reviewer #1 Comment #4

- No QC plots are provided for the CRISPR screen. At a minimum, replicate and sample correlation should be shown. The authors might consider looking at published CRISPR screens to find established methods to demonstrate that the screen is of high-quality (e.g. essential gene dropout).

Response to Reviewer #1 Comment #4

We agree with the Reviewer the quality control is important when interpreting the results of the CRISPR-drug screens. We note that, in our case, we have a very strong built in positive control in the form of the retinoic acid receptor alpha (RARA), the accepted primary target of our drug, knockout of which was the top resistance hit our screens. However, we agree with the Reviewer that this could be supplemented by reporting drop out of pan-essential genes in our screens, comparing these to the 1,000 non-targeting negative control gRNAs that we also included in our screens. We retrieved a set of 55 pan-essential genes from DepMap and compared their gRNA abundances to those of the non-targeting gRNAs. As expected, the gRNAs targeting pan-essential are far more depleted than the non-targeting controls ($P < 2.2 \times 10^{-16}$ from Wilcoxon rank sum test; approximately a 5-fold difference in both cases, which is in line with estimates reported in the literature), showing that the positive and negative controls in the screens are well separated and behaving as expected (now included in Supplementary Figure S1F). We thank the Reviewer for the excellent suggestion which has significantly improved the robustness of our CRISPR screening results.

Reviewer #1 Comment #5

- The authors move from their genome-wide CRISPR screen to analyzing a previous published CRISPR screen targeting 655 genes (Lee, Wright 2023). Why are these 655 genes important (esp. in light of the results previously presented)? Does this previous screen include among the 655 genes many of the same top-ranked genes from the current work? It would be best if the targeted screen capitalized on the discovered top genes from the genome-wide screen but, absent such a justification, the authors must provide some rational basis for these 655 genes being chosen for screening in more NB cell lines.

Response to Reviewer #1 Comment #5

We thank the Reviewer for the comment and provide the following clarification: The previous paper (655 genes) is independent of this study and the 655 genes were not chosen with this study in mind, rather they were chosen because they represented a library of druggable genes. However, because retinoic acid was also included in that previous screen, and that screen was performed at large scale (18 cell lines), it turned out that the data could be informative in aspects of the current study, in particular assessing the generality of the association between FKBP1A knockout and response to retinoic acid. Thus, we were able to use those previously published findings to assess the generality of the relationship between BMP signaling and RA response, though this is also supported by numerous other lines of evidence (gene expression associations, perturbations in multiple cell lines, multiple large scale small molecule screens etc.). We thank the Reviewer for the opportunity to clarify this point.

Reviewer #1 Comment #6

- There are several quantifications missing:
 - 1 Quantification of blot results and immunofluorescence from the "three independent experiments" should be shown in addition to the representative blot/images.
 - 2 It is unclear if there are replicates of the CRISPR screens. If so, correlation between replicates should be reported. If not, replicates should be performed.
 - 3 Many results missing statistics, for example the results presented in Supp Fig 2, others. Sometimes it is unclear what statistical test was performed.

Response to Reviewer #1 Comment #6

We thank the reviewer for these excellent suggestions on the technical presentation of the data, which is now much improved in the revised paper, specifically:

1. We have now quantified the relevant Western blot results and the data are included in Supplementary Table S9. Additional IF images are included in the updated Supplementary Figure 13 and the additional Western blot images are included as Supplementary Figure 15.
2. The CHP-134/ATRA CRISPR screens were performed twice and similar results were obtained each time (Fig S1H). Pearson's and Spearman correlation are not an appropriate statistical measure of concordance for CRISPR-drug screen as the

overwhelming majority of the >20,000 gene perturbations will not have an effect on drug response; the resulting lack of variability in the data will lead to “attenuation due to restriction of range”, a statistical phenomenon that renders correlation coefficients ineffective for assessing concordance/agreement between two variables. An appropriate and interpretable alternative is using an e.g. Fisher’s exact test to compare the number of top hits shared. In our case, 5 of the top 10 hits in each screen are shared between the two screens, which is incredibly statistically unlikely under the null ($P = 2.9 \times 10^{-15}$ from Fisher’s exact test; now added to Fig. S1H). In both screens, the top resistance hit is the accepted primary drug target *RARA*. We thank the Reviewer for helping us improve the statistical rigor with which these results are reported.

3. We have endeavored to improve the reporting of statistical tests throughout the revised manuscript. We have tested the statistical significance of synergy results in Supplementary Figure 2 as suggested. We explained how these results were obtained (using a one-sample t-test), and we thank the reviewer for this helpful improvement.

Reviewer #1 Minor comments

Reviewer #1 Comment #6

- “unbiased genome-wide CRISPR knockout screen” should just be “genome-wide CRISPR knockout screen”. No biological technique is unbiased. In this case: The authors did not design the CRISPR library nor did they systematically measure knock-out efficiency for each gRNA (which certainly differs).

Response to Reviewer #1 Comment #6

We have removed the word “unbiased” (line 126).

Reviewer #1 Comment #7

- “hijacking” of a normal developmental process has been described since the very early studies with retinoic acid, where treatment of acute promyelocytic leukemia led to differentiation of leukemic cells into macrophages and neutrophils

Response to Reviewer #1 Comment #7

We agree with the Reviewer that retinoic acid leads to the differentiation/maturation of blood cells and that this has long been described in APL. We mentioned this in our Introduction and Discussion sections. However, the mechanism we are describing in neuroblastoma appears to be different, and appears to be “hijacking” an *organismal*-developmental process, directing several possible developmental-relevant cell fate decisions. We believe this is conceptually different from the maturation of blood cells. We have attempted to clarify this on line 101, referring to what we are describing as the hijacking of an “organismal developmental” process specifically. We thank the Reviewer for helping us clarify this point.

Reviewer #1 Comment #8

- There was not an insignificant amount of apoptotic cells in DMSO treated cells (Supp Fig 1B)

Response to Reviewer #1 Comment #8

We thank the Reviewer for pointing this out and we understand the concern. Under normal culture conditions, D283 MED is a cell line with mixed multicellular aggregates in suspension and some adherent cells. Some cells may be dead in large aggregates or when they are separated into single cells during sample preparation for flow cytometry. Due to this unique characteristic, it's not uncommon to see a higher apoptotic cell percentage in DMSO control in published literature. For example,

Figure 1C in Yang *et al* 2008 (PMID: 19001435, <https://www.ncbi.nlm.nih.gov/pmc/articles/PMC2592687/>) has > 20% of Annexin V+/PI+ cells under DMSO condition. We have 16.3% of Annexin V+/PI+ cells;

Figure S3D in Fitzgerald *et al* 2023 (PMID: 37898609, <https://pubmed.ncbi.nlm.nih.gov/37898609/>) has > 50% of Annexin V+ cells under DMSO condition;

Figure S2A in Levesley *et al* 2018 (PMID: 29016820, <https://www.ncbi.nlm.nih.gov/pmc/articles/PMC7059858/>) has ~20% PI+ cells under DMSO condition.

Reviewer #1 Comment #9

- Additional differentiation markers should be evaluated

Response to Reviewer #1 Comment #9

We agree with the Reviewer that multiple lines of evidence should be evaluated in order to robustly assess cellular differentiation. Further, we have endeavored to support all claims in our manuscript by multiple orthogonal lines of evidence, from multiple different types of assays and any claims we have made around differentiation are no different from this. These include:

- Additional NeuN western blots in CHP-134 cells (now included in Figure S1B).
- Western blots for MAP2 a/b MAP2 c/d (Figures 3B, E, G, S1B, S3B, C)
- IF for NEFH in CHP-134 and TGW (e.g. Figure 3A, 4A)
- Western blots for Tau, DCX in xenograft tissues and CHP-134 cells (e.g. Figure S8M)
- Images showing clear neurite outgrowth and differentiated features, e.g. Fig. 3A, 4A, S1C, Fig. 4D, F, I, K.
- We generated CHIP-seq data showing that RARA binding was occurring at differentiation-related genes (e.g. Fig. S5G-H, S6C, S7C, S7E, Tables S3-S5).
- We similarly showed RNA-seq data, showing strong enrichment of genes functionally related to neuronal differentiation (e.g. Fig. 5F, Tables S3-S5).

We thank the Reviewer for helping us clarify that our broader conclusions around retinoic acid induced differentiation are supported by the numerous lines of evidence shown in the manuscript.

Reviewer #1 Comment #10

- Some experimental/analysis details are missing. What kind of replicates in 4A? How many replicates of bulk RNA-seq? How were single cell counts converted to TPM - is each dot a cell in 5F, Supp Fig 9E-M or how was this analysis done? How old were the mice, what was the sex, and how often were they weighed?

Response to Reviewer #1 Comment #10

We thank the Reviewer for highlighting these points, which we respond to below and cite the line number for updates to the revised manuscript where appropriate:

What kind of replicates in 4A?

The replicates in 4A and 5A were biologically independent replicates. We have now stated this explicitly on line 396 and 524.

How many replicates of bulk RNA-seq?

1. There were 2 replicates for all high-throughput sequencing data, including RNA-seq, and a detailed summary of these data is presented in Table S3.

How were single cell counts converted to TPM - is each dot a cell in 5F, Supp Fig 9E-M or how was this analysis done?

TPM (Transcripts Per Million) is the most commonly applied normalization procedure for single-cell RNA-seq, which corrects the read count for each gene by the total transcript count in the cell, the gene length, then multiplies this number by 1 million. To generate Fig. 6F, for each gene, we then calculated the average TPM for all single cells within the same cluster of neuroblastoma cells. Thus, each dot in Fig 6F is a cluster of neuroblastoma cells (all single cells were clustered using the acNMF method) and the y-axis is the average expression of the indicated genes. Complete details of our single-cell analysis are in our Methods section (lines 1110-1130). We now realize this figure was not properly described in the legend, where we have now added more detail to properly explain what the axes and points represent (lines 662-664). We thank the reviewer for highlighting this problem.

How old were the mice, what was the sex, and how often were they weighed?

All the mice were purchased from Charles River (strain code 553), female and approximately 12 weeks in age. They were weighed weekly. These details are now included in the Methods on line 1147-1148.

Reviewer #1 Comment #11

- Addgene plasmid numbers are missing for some plasmids (e.g. lentiCas9-Blast)

Response to Reviewer #1 Comment #11

We thank the Reviewer for noticing this omission, which has now been included (line 816, Addgene #52962).

Reviewer #1 Comment #12

- Minor typographical and citation errors
 - Order of figure panels in text (1E then 1D), Fig 2K should be 2B
 - "exprcted" should be "extracted", "AVCR1" is "ACVR1"
 - Cangelosi et al. 2020 was used but not cited

Response to Reviewer #1 Comment #12

We thank the Reviewer for noticing these, each of which has now been fixed (lines 129, 196, 825, 1133) and the Cangelosi reference has been added (line 1264).

Overall, we thank Reviewer #1 for their very careful and helpful assessment of our paper, which have greatly improved the quality of the overall work and helped us find several key areas which we were able to improve. We again thank them for their time, and hope that we have been able to address their concerns.

Reviewer #2 (Remarks to the Author): expert in ATRA/RA, Epigenetics

Reviewer #2 Overall summary:

In this study, Pan & Zhang et al. investigated the effects of ATRA in neuroblastoma cells. The authors employed a comprehensive range of methods to identify and examine the impact of BMP signaling on ATRA-induced differentiation and apoptosis/senescence. We have some comments:

Reviewer #2 Comment #1:

- In the manuscript, Genomics of Drug Sensitivity in Cancer (GDSC1) by Iorio, Knijnenburg et al. 2016 was referenced and NB-13 was pointed out as a retinoic acid sensitive cell line with IC50 of 0.26 μ M. The authors then moved onto analyze 16 cell lines themselves by treating cell lines with ATRA for 6 days, which they state as a more clinically relevant treatment duration. According to Supplementary Figure 1A and Supplementary Table 1, under their 6 days treatment protocol IC50 values of NB-13, CHP-154 and D283 Med are 287.7, 2.507 and 4.399 nM respectively. In the following parts of the manuscript, NB-13, CHP-134 and D283 Med are treated with 2 μ M ATRA, which is 2-3 orders of magnitude higher than the calculated IC50 values. Could this effect or explain the induction of apoptosis over differentiation? Does

isotretinoin have a comparable effect? Some experiments could be added testing lower ATRA concentrations.

Response to Reviewer #2 Comment #1:

We thank the Reviewer for their insightful questions, which we address below:

1. Could ATRA concentrations effect or explain the induction of apoptosis over differentiation?

We tested ATRA at several different concentrations in the manuscript, not just 2 μ M. For example, in Figure 3I, we tested cell apoptosis with lower ATRA concentrations (right panel, 0.05 μ M and 0.1 μ M ATRA for 6 days). We also tested cell differentiation under the same conditions. While we can see these low concentrations of ATRA still induced apoptosis, no differentiation (increase of MAP2 expression) was observed (the data shown below). This indicates that regardless of the concentrations of ATRA, apoptosis is the favored cell fate decision over differentiation.

2. Does isotretinoin have a comparable effect?

Yes. We compared cell sensitivity to ATRA and isotretinoin (in CHP-134, TGW, BE3C, BE(2)-M17), which is shown in Fig. S8C. We did not observe significant differences between the two isoforms. Additionally, in our mouse experiments, isotretinoin was quickly converted to ATRA, and in the mouse CHP-134 tumors, the concentrations of ATRA were actually higher than the concentrations of isotretinoin, as measured by HPLC (Fig. S8D, consistent with the dominant idea in the literature that the activity of isotretinoin is largely as a pro-drug, which is converted to ATRA by both enzymatic and non-enzymatic processes, see e.g. PMID15826600, PMID15714209).

Reviewer #2 Comment #2:

- pSMAD 1/5/9 protein levels in Western blots are shown as an indicator of activated BMP signaling. To interpret phosphoprotein levels total SMAD1/5/9 levels should have been assessed in parallel.
- The same applies to the analysis of c-PARP as a marker for apoptosis induction. Total PARP

levels should be included.

Response to Reviewer #2 Comment #2:

We agree with the Reviewer that these Western blots could be valuable, though we emphasize that our overarching conclusions based on cPARP (apoptosis) and pSMAD (summarizing the overall levels of BMP signaling activity) were supported by multiple orthogonal assays. We have now included these additional blots in Supplementary Figure S12. We additionally note that our manuscript has not drawn any conclusions on the basis of total SMAD protein levels, which can be difficult to deconvolve from BMP signaling activity—especially in the longer-term treatments relevant to study the interaction with RA—because BMP SMAD transcription factors engage in autoregulatory feedback by binding to their own promoter regions, thereby modulating their transcriptional activity in a self-regulatory manner.

Reviewer #2 Comment #3:

- Although FK506 and rapamycin can enhance BMP signaling activity by tethering FKBP1A to calcineurin and by tethering FKBP1A to mTOR, respectively, both are known to have a broad spectrum of effects. For instance, mTOR inhibitors have been demonstrated to synergize with ATRA in AML, as evidenced by their impact on MYC. This could be discussed.

Response to Reviewer #2 Comment #3:

We agree with the Reviewer's concern here. On line 232-235 we state, "The max synergy scores for RA+FK506 and RA+Rapamycin were significantly correlated across tested cell lines (Fig. 2H), suggesting that, while both FK506 and rapamycin affect other cellular processes, their shared common effect on amplifying BMP signaling contributed to the synergy with ATRA.". For these results to be meaningfully informative as to the effect of BMP signaling, it was important that we test 2 compounds with a shared effect on BMP signaling, as each of these compounds has its own orthogonal effects, which would otherwise have confounded these results.

Reviewer #2 Comment #4:

- To analyze interactions between BMP and RA signaling transcription factors, the authors performed CHIP-seq binding profiles of SMAD4, SMAD9, and RARA baseline and ATRA-treated cells. Their data indicate high amount of overlaps between peaks (such as RARA peaks overlapping 87% of SMAD9 peaks and 98% of SMAD9 peaks overlapping with SMAD4 peaks). In our hands, RARA ChIPs are usually of very low quality, although we never tested the here used antibody. Was the specificity of the antibodies assessed and blacklisted known phantom peaks excluded? One could also assess canonical binding motives or known targets.

Response to Reviewer #2 Comment #4:

We used the human RARA antibody # PP-H1920-00 from R&D Systems, which had been previously used in Fang et al, 2012 (PMID 22383578), yielding reliable results. In our hands this antibody also performed well, but we agree with the Reviewer that motif enrichment at the identified RARA binding sites could bolster confidence in the peaks we have called. We have

now included motif enrichment analysis at RARA binding regions in Table S6. The binding motifs of RARA, RXRB, RXRG were among the top 3 ranked motifs (all $P < 10^{-15}$), supporting the validity of this antibody.

Additionally, we had also removed blacklisted regions during our peak calling. We used the blacklist provided by the ENCODE project for hg38 and filtered out these blacklisted regions. In addition, for the neuroblastoma cell line specific problematic highly repetitive regions, such as MYCN region, which show abnormally high signal, we also generated a custom blacklist. We thank the Reviewer for highlighting this important detail, which we omitted from our original description of the Methods, but is now included in the revised Methods section (line 1059-1062).

Reviewer #2 Comment #5:

- It has been previously shown that fetal calf serum contains high amounts of BMPs. <https://doi.org/10.1186/1471-2121-10-20>.) Could this be skewing the results authors get in vitro towards a “bone marrow metastasis” like phenotype and the reason why in vitro dynamics cannot be replicated in primary tumor sites?

Response to Reviewer #2 Comment #5:

Yes, this is a very insightful comment and is an important point to discuss in more detail. As the Reviewer has suggested, we also believe that BMP ligands in the cell culture media is causing a “bone marrow-like” environment for the cells *in vitro*. We have included this citation and additional discussion of this idea that access to BMP ligands is likely to partially explain the context specific activity of BMP signaling and thus retinoic acid response (lines 700-701). We thank the reviewer for helping us better express this important point.

Reviewer #2 Comment #6:

- In experiments that involve viral transduction (Such as Figure 2K) comparisons are made between control groups, there is no information regarding how empty controls compare to parental cells.

Response to Reviewer #2 Comment #6:

We thank the Reviewer for their feedback regarding the inclusion of parental cells in our experiments involving viral transduction of shRNAs (Figure 2K). We appreciate the opportunity to clarify our experimental design. We employed a scrambled shRNA control, which serves as a non-targeting sequence that does not affect the expression of any specific genes. This control is widely recognized as a robust standard in shRNA experiments because it accounts for any potential effects associated with the viral transduction process itself, including viral entry, integration, and expression of shRNA sequences. By using the scrambled shRNA control, we ensure that any observed effects are due to the specific knockdown of our target gene rather than off-target effects or the transduction procedure. Using parental cells (cells not subjected to viral transduction) would introduce additional variables unrelated to the specific gene knockdown, such as differences in cellular responses to viral exposure or selection markers. The scrambled shRNA control more closely mirrors the experimental conditions of the target

shRNA-transduced cells, providing a more accurate baseline for the comparisons of interest. We thank the Reviewer for the opportunity to clarify our rationale, which we believe supports the conclusions we have drawn based on these experiments/data. We also included the comparison between parental cells and scrambled shRNA control cells in the bar chart below, where we did not find substantial difference between the two groups.

Reviewer #2 Comment #7:

- Considering the author's aim to explain the RA's lack of effect on primary human tumors and its efficacy in eliminating neuroblastoma cells from the bone marrow during post-chemo maintenance, co-culture systems between bone marrow stromal / immune cells and neuroblastoma cells would allow authors to enlighten the effect of bone marrow microenvironment on the apoptosis / differentiation dynamics in neuroblastoma cells.

Response to Reviewer #2 Comment #7:

We agree with the Reviewer, and we did in fact previously consider exactly these experiments. However, this was not as straightforward as we anticipated, because bone marrow stromal cells growing in culture shut down the expression of BMP ligands (see table below, where we tested gene expression of BMP ligands in these cells). It is plausible this is response to BMP in the cell culture media. Thus, we chose to instead use the strategy of directly treating the neuroblastoma cells with BMP ligands (Fig 6K-L, Fig S10B-C). Arguably, this is also a cleaner experiment, as the co-culture system alone would not necessarily have pinpointed the specific factor underpinning the interaction, because bone marrow stromal cells will secrete additional factors other than BMP.

Column1	hTERT-MSC_1	hTERT-MSC_2	hTERT-MSC_22
Gene	SJNORM048145_C19	SJNORM048145_C20	SJNORM048145_C21
BMP10	-1	-1	-1
BMP15	-1	-1	-1
BMP2	-0.884034753	-0.924061239	-0.910642803
BMP2K	1.647780538	1.711926579	1.807429194
BMP3	-0.974940836	-1	-1
BMP4	1.806957722	1.816890836	1.733130574
BMP5	-1	-0.964906454	-1
BMP6	1.211640477	1.101103544	1.153347015
BMP7	-0.973492682	-0.987101912	-0.989824653
BMP8A	-0.916782439	-0.735411227	-0.936043024
BMP8B	-0.857029319	-0.796667457	-0.787050307
BMPER	2.355878592	2.162920237	2.063225031
BMP1	5.132469177	4.986493111	4.886711121
BMPR1A	2.445197344	2.411584139	2.48474431
BMPR1B	0.42182228	0.305858433	0.360346258
BMPR2	2.803089857	2.541646242	2.784680605

Figure legend: Log₂ TPM values for various BMP ligands for (hTERT-MSC) bone marrow stromal cells growing in vitro. This compares to log₂ TPM values of 6-8 for the similar genes for bone marrow stromal cells in neuroblastoma patient data.

Reviewer #2 Comment #8:

- In Supplementary Figure 3B what kind of treatment DMSO Day 0 represent?

Response to Reviewer #2 Comment #8:

We thank the Reviewer for this comment. Now the details are included in the figure legend (now Figure S1C): Day 0 indicates DMSO treatment (< 0.01% of DMSO, the same concentration as in RA treated cells) for 13 days.

Reviewer #2 Comment #9:

- In Figure D, the authors show Enrichr clustergram of CRISPR screen results and state that the input is top and bottom 1% genes. The figure could be more clear on which genes are top and bottom.

Response to Reviewer #2 Comment #9:

We thank the Reviewer for this valuable comments. We have now included these missing details in this figure legend (lines 171-172) and the gene list and directionality is now added included in Table S2.

Reviewer #2 Comment #10:

- In order to confirm their high throughput CRISPR screen, the authors knocked down the BMP type I receptor ACVR1, with three independent shRNAs, and consistent with the screen. When describing their following approach of using selective BMP inhibitor K02288, they state that “due to functional redundancy, a single gene knockout is usually not sufficient to fully inactivate BMP signaling”. While the statement itself might be true, it causes a confusion between CRISPR KO and the technique they actually used (shRNA mediated knockdown).

Response to Reviewer #2 Comment #10:

We agree with the Reviewer that this text is confusing as worded. We have now reworded this as “silencing a single gene” (line 150), which should be more accurate and consistent with the experimental result reported.

Overall, we thank Reviewer #2 (and co-Reviewer #3, whose comments are also addressed above) for spending their time to review our paper in detail. They have provided us with multiple very helpful comments which have greatly improved the clarity and quality of the revised manuscript. We hope that we have been able to comprehensively address their comments in full and that they agree our primary conclusions are well supported by the data shown.

Reviewer #3 (Remarks to the Author): ECR, co-reviewed with rev#2

I co-reviewed this manuscript with one of the Reviewers who provided the listed reports. This is part of the Nature Communications initiative to facilitate training in peer review and to provide appropriate recognition for Early Career Researchers who co-review manuscripts.

Response to Reviewer #3

We thank Reviewer #3 for the time, attention to detail, and rigor in co-reviewing this manuscript. We have addressed their comments along with our responses to Reviewer #2 above.

Reviewer #4 (Remarks to the Author): expert in BMP signaling

Reviewer #4 Summary

Pan et al., demonstrated that retinoic acid (RA) treatment can induce apoptosis or senescence in neuroblastoma cells. Through a genome-wide CRISPR screen, they identified the bone morphogenetic protein (BMP) signaling pathway as a key regulator of RA-induced apoptosis in these cells. Furthermore, they observed that neuroblastoma cells derived from bone marrow exhibit heightened BMP activity, offering a novel perspective on the mechanisms of action (MoA) of RA in the clinical treatment of neuroblastoma patients.

Response to Reviewer #4 Summary

We thank the Reviewer for their kind summary and overall positive assessment of our work.

Reviewer #4 Major points:

Reviewer #4 Comment #1

1. The authors observed that certain pediatric cancer cell lines exhibit exceptionally low IC50 values in response to RA treatment, as shown in Figure 1A. It would be beneficial to include clinical data on these cell lines, such as the presence of driver mutations and potential mutations in BMP signaling pathway components.

Response to Reviewer #4 Comment #1

We agree with the Reviewer. Indeed, we had also originally suspected that some of these highly responsive cell lines may have contained mutational events in key genes in the BMP signaling or retinoic acid signaling pathways. Thus, very early in this project, we performed whole genome sequencing on the cell lines CHP-134 and NB-13, but we identified no clear mutational events targeting BMP signaling, nor retinoic acid signaling. This is consistent with whole genome-sequencing analysis of neuroblastoma patient cohorts, where recurrent mutations targeting these signaling pathways have never been identified (see Brady *et al* for the largest reported analysis: <https://www.nature.com/articles/s41467-020-18987-4> (additional note: these sequencing analyses were performed by colleagues in our department at St. Jude, with whom we have corroborated these conclusions)). Thus, mutational events do not explain the strong responses to retinoic acid in some of these cell lines. However, we agree with the Reviewer that this will be a common question from readers, and it would be helpful for us to report these results in more detail. Thus, we have now included this information on common neuroblastoma driver mutational events in the cell lines used in this study Supplementary Table S1. We thank the reviewer for the insightful and helpful comment.

Reviewer #4 Comment #2

2. The authors employed K02288 to inhibit BMP signaling in neuroblastoma cells, reporting that BMP activity became undetectable after three days of treatment. However, phosphorylation of SMAD1/5/8 typically peaks within 15 minutes, with downstream targets activated within 2 hours in the presence of BMP ligands. Standard BMP signaling pathway inhibitors also block SMAD1/5/8 phosphorylation within hours, not days. Therefore, to more accurately assess the inhibition of BMP signaling activity, a shorter treatment duration should be used instead of the three-day period implemented in this study.

Response to Reviewer #4 Comment #2

We thank the Reviewer for their comment regarding the duration of perturbations to BMP signaling used in our study. We appreciate the opportunity to clarify our experimental design and the rationale for using longer duration (e.g. three-day) treatments. Our study focuses on the interplay between BMP signaling and retinoic acid (RA) signaling in the context of neuroblastoma cells specifically. While it is accurate that BMP signaling can respond to ligands within minutes and that inhibitors like K02288 can block this phospho-SMAD rapidly, the downstream biological effects we are investigating require a much longer timeframe to manifest.

This is because retinoic acid signaling affects cellular responses over days, or even weeks (see e.g. Fig. S3B). The phenotypic changes associated with RA treatment develop progressively and cannot be adequately assessed within a few hours. Therefore, to effectively evaluate the impact of BMP inhibition on RA response, it is essential to maintain BMP pathway inhibition throughout the entire period of RA treatment. By extending the K02288 treatment to e.g. three days, we ensured consistent suppression of BMP signaling during the critical window of RA-mediated effects. This prolonged inhibition allows us to capture the cumulative and synergistic interactions between the BMP and RA pathways, providing a more accurate representation of their interplay in the specific physiological context that we are studying. We thank the Reviewer for their insightful comment and hope we have clarified our methodology and the necessity of longer treatment periods in our specific experimental setup and the specific case of probing interactions with retinoic acid signaling.

Reviewer #4 Comment #3

3. The authors assert that BMP signaling is essential for retinoic acid (RA)-induced apoptosis in neuroblastoma (NB) cells. They demonstrate that inhibition of BMP signaling with K02288 reduces NB cell sensitivity to RA, as illustrated in Figure 1F, and prevents the increase in cleaved PARP levels induced by ATRA, shown in Figure 1G. This observation is noteworthy. The authors should further investigate whether RA treatment activates BMP signaling or coordinates with BMP signaling pathways to facilitate apoptosis in NB cells.

Response to Reviewer #4 Comment #3

We thank the Reviewer for their comment regarding the interaction between retinoic acid (RA) treatment and BMP signaling in facilitating apoptosis in neuroblastoma (NB) cells. Our data suggest that RA treatment can decrease BMP signaling activity. These are shown in Figure S5I, Figure 6D (d0 vs d6), 6J (DMSO vehicle vs ATRA vehicle). The latter part of our manuscript includes an extensive section specifically dedicated to investigating how RA treatment influences BMP signaling and how these pathways coordinate to determine cell fate in NB cells. Specifically, the section titled "Downstream BMP and RA transcription factors cooperate to determine cell fate following exposure to RA" (Figure 5) addresses this question in detail. In this section, we performed comprehensive analyses, including:

- ChIP-seq profiling: We analyzed the binding profiles of SMAD4 and SMAD9 (key BMP pathway transcription factors) and RARA (the RA receptor) in multiple NB cell lines representing different cell fate outcomes (apoptosis, senescence, and differentiation).
- Overlap of transcription factor binding: We found significant overlap in the binding sites of SMAD4, SMAD9, and RARA, indicating that RA and BMP signaling pathways converge on common target genes involved in apoptosis.
- Pathway enrichment analyses: Gene Set Enrichment Analysis (GSEA) revealed that genes co-bound by these transcription factors are enriched in apoptosis-related pathways. This suggests that RA treatment may activate BMP signaling components or that these pathways work in concert to promote apoptosis.

- Temporal changes over 6 days upon RA treatment: Our time-course experiments showed dynamic changes in transcription factor binding and gene expression profiles following RA treatment, further supporting the coordinated interaction between RA and BMP signaling in determining cell fate.
- Functional validation: Overexpression and knockdown studies of key genes (such as the ID family genes) demonstrated that modulating BMP signaling components affects NB cell sensitivity to RA-induced apoptosis (other similar perturbational results were included Figures 3 and 4).

Collectively, we believe these findings provide substantial details on how RA treatment interacts with BMP signaling to facilitate apoptosis in NB cells and we thank the Reviewer for the opportunity to clarify these additional aspects of our results.

Reviewer #4 Comment #4

4. FKBP1A is shown to inhibit not only BMP signaling but also TGF β 3 signaling. Consequently, FK506 or rapamycin, which act as FKBP1A inhibitors, enhance both BMP and TGF β 3 signaling. Therefore, in Figure 2D-2H, the author should exercise caution in concluding that BMP signaling is the sole target of these small molecules.

Response to Reviewer #4 Comment #4

We agree with the Reviewer's concern here and we have not concluded that BMP signaling is the sole target of these molecules nor have we drawn any broad conclusions of this type based on this result in isolation. On line 232-235 we state, "The max synergy scores for RA+FK506 and RA+Rapamycin were significantly correlated across tested cell lines (Fig. 2H), suggesting that, while both FK506 and rapamycin affect other cellular processes, their shared common effect on amplifying BMP signaling contributed to the synergy with ATRA." For these results to be meaningfully informative as to the effect of BMP signaling, it was important that we test 2 compounds with a shared effect on BMP signaling, as each of these compounds has its own orthogonal effects, which would otherwise have confounded these results. We thank the Reviewer for helping us clarify this point.

Reviewer #4 Comment #5

5. The authors utilized SMAD9 knockout (KO) to investigate neuroblastoma (NB) cell differentiation in the context of BMP signaling inhibition, as shown in Figures 3G and S3D. First, the authors should provide evidence of the extent of SMAD9 KO to confirm its effectiveness, given that SMAD9 expression levels are typically low in human cells. Additionally, to achieve more comprehensive inhibition of BMP signaling, it is recommended that the authors also conduct SMAD1/5 KO instead of single gene KO (SMAD1 KO). Alternatively, the authors should present data demonstrating the inhibition of BMP downstream targets following SMAD9 knockout.

Response to Reviewer #4 Comment #5

We thank the Reviewer for pointing out these important. First, we confirmed SMAD9 knockout by western blot (Fig S2F, results shown for single gRNAs as well as multiple gRNAs combined, in both CHP-134 and TGW cells). Additionally, per the Reviewer's excellent suggestion, in addition to our genome-wide CRISPR screens, we did perform more comprehensive inhibition of BMP signaling using various orthogonal approaches, which included shRNA targeting of ACVR1 (Fig. 2K, Fig. S1I-J), treatment with K02288 (Fig. 1F-G, Fig. S1G, Fig. 3B-E, Fig. 4A-F) the knockout of SMAD1 (Fig. S2I-J) and treatment with noggin (Fig 3E-F). Given the numerous orthogonal assays we used to suppress BMP signaling broadly, as well as the various strategies we employed to amplify BMP signaling, and the consistency of the results we observed across these various assays, we believe our broader conclusions are strongly supported by the collective data we have shown.

Reviewer #4 Comment #6

6. The authors conducted chromatin immunoprecipitation sequencing (ChIP-seq) to assess the genomic binding patterns of RARA, SMAD4, and SMAD9 in neuroblastoma (NB) cell lines following all-trans retinoic acid (ATRA) treatment. Interestingly, the analysis revealed a significant increase in SMAD4 binding across the genome post-ATRA treatment, contrasting with a slight decrease observed in SMAD9 binding. This discrepancy warrants further elucidation from the authors to provide a comprehensive understanding of the differential regulatory roles of SMAD4 and SMAD9 in response to ATRA.

Response to Reviewer #4 Comment #6

We thank the Reviewer for pointing out this important point. SMAD4 activity is not unique to BMP signaling and is also a core component of the canonical TGF-beta/activin pathway. Thus, SMAD4 expression is not alone a reliable proxy for BMP signaling activity, which is why we have used e.g. pSMAD1/5/9 and the expression of the BMP family of ID target genes as our main tools to read out the activity of BMP signaling in our paper. Our manuscript does not conclude that we have understood every aspect of differential regulation of canonical TGF-beta/activin signaling and BMP signaling activity in these contexts. These relationships are themselves an entire active field of research, which can be reviewed in e.g. PMID38514615. We agree that further understanding these relationships, in a range of contexts, will provide valuable foundations for future research projects. We thank the reviewer for helping us clarify this point.

Reviewer #4 Comment #7

6. The genome-wide increase in SMAD4 binding does not account for the decreased expression of ID genes following ATRA treatment from the perspective of BMP signaling. The authors should differentiate between the activation or inhibition of BMP and TGF₄₃ signaling pathways. TGF₄₃ signaling is known to inhibit ID gene expression, whereas BMP signaling promotes the expression of these genes in certain cellular contexts. Therefore, the GSEA analysis in Figure

5C should focus specifically on BMP signaling rather than the broader TGF- β superfamily.

Response to Reviewer #4 Comment #7

We thank the Reviewer for the important comment on the role of SMAD4. As a point of clarification, Fig. 5C does not deliberately focus on the TGF-beta superfamily (which includes both BMP signaling and TGF-beta/activin signaling). The gene sets listed on Fig. 5C are the top 4 ranked gene sets in the analysis, and it happens that the “TGF-beta signaling pathway” superfamily (which includes BMP signaling) is ranked #4 (of the gene sets included in the Bioplanet 2019 pathways dataset, which is the basis for the analysis). Because this ambiguity can cause confusion, we have generally emphasized the ID family of BMP target genes as the primary means to estimate BMP signaling activity in gene expression space (e.g. Fig. 5G, Fig. 6B-C, 6F). We apologize for the misunderstanding and note that the entire sets of GSEA results is included in Table S4 (the enrichment result of Fig. 5C is also shown for CHP-134 RARA&SMAD9 in Table S4).

Reviewer #4 Comment #8

7. Transcriptional regulation in BMP signaling typically occurs within hours. In the present study, the authors assessed gene expression changes over several days, which may primarily reflect secondary effects. It is strongly recommended that the authors investigate gene expression alterations over shorter time intervals, such as 1 hour or 4 hours, to capture the more immediate effects of BMP signaling.

Response to Reviewer #4 Comment #8

To recap, we are interested in the interactions between BMP signaling and RA signaling, whose effects take days to weeks to manifest. This is discussed in detail in “Response to Reviewer #4 Comment #2”. We thank the reviewer for allowing us to clarify this important detail, relevant in this experimental system.

Reviewer #4 Minor points:

Reviewer #4 Comment #9

1. In Figure 1B, the proportion of Annexin V+/PI- cells decreases on Day 6 in the D283 Med cell line following ATRA treatment, which contrasts with the observations in the other two cell lines. The authors should provide an explanation for this phenomenon.

Response to Reviewer #4 Comment #9

Yes, we agree with the Reviewer that this could seem unusual at first glance. However, the reason for this is that this cell line is extremely sensitive to RA induced apoptosis and almost all the cells are dead by 6 days (please see the underlying data in Fig. S1D).

Reviewer #4 Comment #9

2. The authors conducted a Western blot analysis to assess cleaved PARP expression following ATRA treatment in two neuroblastoma cell lines and a medulloblastoma cell line, as depicted in Figure 1C. However, it is noteworthy that the loading controls used differ between the neuroblastoma cell lines (13-Actin) and the D283-Med cell line (GAPDH). The authors should provide an explanation for this.

Response to Reviewer #4 Comment #9

We thank the Reviewer for pointing this out, we have now included a Western blot result in Supplementary Figure 12 using 13-Actin as a loading control for D283 cell line. Both 13-Actin and GAPDH are good housekeeping genes that can be used as loading control. The main reasons why we used GAPDH instead of 13-Actin are (1) on the Western blot membrane, 13-Actin is too close to other proteins that we also want to study in D283 MED cell line; (2) antibodies for other proteins showed nonspecific bands at a similar molecular size as 13-Actin on the western blot membrane for this cell line, when we blot the membrane with those antibodies first.

Reviewer #4 Comment #10

3. The legend of Fig. 1D should indicate the meaning of the panel color (the red color indicates the screen hits?).

Response to Reviewer #4 Comment #10

We thank the Reviewer for this valuable comments. Now we included more details in the figure legend (lines 172) and the gene list and directionality is added in Table S2.

Reviewer #4 Comment #11

4. The authors conducted an unbiased genome-wide CRISPR screen using a neuroblastoma cell line to compare the enrichment and depletion of gene knockouts in cells treated with ATRA for six days versus mock-treated cells. It is recommended that the authors include a schematic diagram to illustrate this experimental strategy.

Response to Reviewer #4 Comment #11

We thank the Reviewer for the helpful suggestion. We have now included a schematic in Supplementary Figure 1E and referenced this from the main text on line 128. This has greatly improved the interpretability of these results.

Reviewer #4 Comment #12

5. Line 131: SMAD4 and SMAD9 are not downstream targets of BMP signaling. The authors should revise this description accordingly.

Response to Reviewer #4 Comment #12

We thank the Reviewer for helping us clarify our message. On line 131 we had intended to communicate that SMAD4 and SMAD9 were targets of BMP receptors, not downstream targets

of BMP signaling. We have now worded this more explicitly stating that SMAD4 and SMAD9 are the “targets of these [BMP] receptors” (line 135).

Reviewer #4 Comment #13

6. Figure S2B does not specify the units for the x-axis. Does “time” refer to years? The authors should clarify this in the figure legend.

Response to Reviewer #4 Comment #13

Yes we realize this was ambiguous, it had intended to be “years”. We thank the Reviewer for noticing the omission and have updated Figure S2B to indicate this.

Reviewer #4 Comment #14

7. Line 193: If the authors intend to describe all SMAD proteins involved in the BMP signaling pathway, they should also analyze the correlation of Co-SMAD (SMAD4) and I-SMAD (SMAD6/7) with the ATRA drug response. Or the authors want to emphasize the R-SMADs (SMAD1/5/8), they should correct the statements of “BMP SMADs”.

Response to Reviewer #4 Comment #14

We agree with the Reviewer that this requires further clarification and we have now updated this to the intended and more specific term “R-SMADs” (line 198). We thank the reviewer for helping us improve the terminology.

Reviewer #4 Comment #15

8. The authors should use total SMAD1/5/9 as the loading control for detecting pSMAD1/5/9 levels in Figures S1G, 2D, S5I, and 6J.

Response to Reviewer #4 Comment #15

We agree with the Reviewer that these Western blots could be valuable for evaluation of BMP activity, though we emphasize that our overarching conclusions based on pSMAD (summarizing the overall levels of BMP signaling activity) were supported by multiple orthogonal assays. To understand SMAD1/5/9 phosphorylation more precisely, we have now included additional blots showing total SMAD1/5/9 expression in Supplementary Figure S12. We additionally note that our manuscript has not drawn any conclusions on the basis of total SMAD protein levels, which can be difficult to deconvolve from BMP signaling activity—especially in the longer-term treatments relevant to study the interaction with RA—because BMP SMAD transcription factors engage in autoregulatory feedback by binding to their own promoter regions, thereby modulating their transcriptional activity in a self-regulatory manner.

Reviewer #4 Comment #16

9. Lines 226-228: The authors demonstrated that FK506 and rapamycin both enhance BMP signaling, contributing to their synergistic effects with ATRA. To further elucidate this, the authors should conduct RNA-seq or qPCR analyses targeting BMP signaling pathways in neuroblastoma (NB) cell lines treated with FK506 + RA and rapamycin + RA. This would allow for a comparative assessment of the shared effects of these treatments on BMP signaling.

Response to Reviewer #4 Comment #16

We agree with the Reviewer that RNA-seq analysis is valuable to further understand the shared effects of these treatments and their relationship to BMP signaling. We utilized the Bliss δ synergy scores for both the FK506+RA and the rapamycin+RA treatments and identified the association between these summary statistics and genome-wide gene expression, estimated by RNA-seq, across our entire drug-screened cohort. Interestingly, of the BMP factors investigated, δ scores for both the FK506+RA and rapamycin+RA combinations are most correlated with the expression of SMAD9 (Supplementary Figure S2M-N), which is reminiscent of the expression correlations against the baseline RA-treated single-agent tumors, where SMAD9 is also the top correlation (Fig. 2A-B). Overall, the data suggest that baseline BMP signaling activity is also a determinant of the response to these combinations. We thank the Reviewer for helping us clarify how our experimental design supports the paper's broader conclusions.

Reviewer #4 Comment #17

10. Line288-290: The authors demonstrated that the combination of ATRA and K02288 induced differentiation more quickly than K02288 alone in Fig. 3B. They should perform a time-course western blot to check the induction of differentiation between the combination and K02288 alone.

Response to Reviewer #4 Comment #17

We thank the Reviewer for highlighting this discrepancy and agree that this was worded such that it was not entirely consistent with the data shown. We have re-written this to correct the discrepancy, now reading "The combination of ATRA and K02288 induced differentiation even more intensely than RA or K02288 alone" (line 296-297), which is consistent with the data we have shown (which as the reviewer pointed out is not a time course) and is consistent with the paper's overall conclusions.

Reviewer #4 Comment #17

11. The gene symbols should be italic in Fig. 5G.

Response to Reviewer #4 Comment #17

We thank the Reviewer for spotting this oversight and have now updated Fig. 5G.

Reviewer #4 Comment #18

12. Compared to the DMSO control condition, SMAD9 binding at TLX2 and ID1 appears to increase after 1 day of ATRA treatment, followed by a subsequent decrease over time, as depicted in Figure S4F. The authors should provide an explanation for this dynamic change in BMP signaling activity.

Response to Reviewer #4 Comment #18

We appreciate the Reviewer's attention to detail. However, based on our comparison of SMAD9 binding peaks around genes (as shown in Figure S4F) between the 1-day ATRA treatment and control samples, the magnitude of these differences is very small, and there is no statistically significant difference after 1 day of ATRA treatment (see table below). Therefore, it is not possible to conclude that SMAD9 binding increases following 1 day of treatment. Our data only supports a decrease in SMAD9 binding after 3 days of ATRA treatment.

Region	Closest_Gene	Gencode_id	log2FoldChange	pvalue
chr13:36919909-36920623	SMAD9	ENSG00000120693.13	0.065291733	0.819581885
chr2:74512430-74513960	TLX2	ENSG00000115297.11	0.123655065	0.374327193
chr20:31603798-31605585	ID1	ENSG00000125968.9	-0.014683942	0.921922836
chr20:31608242-31609032	ID1	ENSG00000125968.9	-0.301141173	0.087317887
chr3:170355762-170358570	SKIL	ENSG00000136603.14	0.286304289	0.066763726

Reviewer #4 Comment #19

14. In Figure 6F, the authors used the mean expression levels of ID genes to assess BMP signaling activity. However, given that ID1, ID2, and ID3 may exhibit extreme variations in expression across different cells or tissues, calculating mean values (ID1 + ID2 + ID3) may not accurately reflect BMP signaling activity. It is recommended that the expression levels of ID1, ID2, and ID3 be compared individually to provide a more precise analysis.

Response to Reviewer #4 Comment #19

We agree with the Reviewer and indeed the expression levels of the individuals ID genes were included in Figures S9E-G, where the trends for each gene was consistent with the overall set of trends observed at the level of the mean.

Overall, we thank Reviewer #4 for taking the time to review our manuscript in detail and for their excellent comments. We thank the Reviewer for identifying several issues and helping us improve the overall quality and rigor of the manuscript. We are hopefully that we have adequately addressed their concerns and believe that the revised manuscript represents is

much improved and we believe that our overall conclusions are well supported by the data and analyses.

Reviewer #5 (Remarks to the Author): expert in neuroblastoma and ATRA/RA

Reviewer #5 Summary

Pan et al. report the mechanism of retinoic acid in eliminating post-chemotherapy neuroblastoma cells. In a previous publication, Iorio et al. showed that two neuroblastoma cell lines are >100-fold sensitive to all-trans-retinoic acid compared to other cell lines in *in vitro* activity testing. In repeat experiments of six-day incubation with ATRA, the authors found two sensitive cell lines were apoptotic. CRISPR knockout screen showed that BMP signaling was one of the enriched pathways associated with ATRA sensitivity. Using BMP inhibitors and knock-down experiments, they showed that BMP contributes to ATRA-induced apoptosis. They also reported that BMP inhibition induced differentiation of neuroblastoma cells by ATRA. However, in the cell-line xenograft model of the same cell line that was utilized for the *in vitro* studies, the BMP signaling pathway was not enriched. In investigating the reason, the authors reported that the bone marrow stromal cells obtained from patients have high BMP protein expression. Therefore, the ATRA effect in eliminating neuroblastoma cells in bone marrow is due to the tumor microenvironment with high BMP expression. There are major and minor comments on the manuscript to improve the quality of the study.

Response to Reviewer #5 Summary

We thank the Reviewer for their summary of our paper and have endeavored to address their comments below.

Reviewer #5 Comment #1

1. The authors utilized all-trans-retinoic acid instead of 13-cis retinoic acid for no clear reasons. Although they are enantiomers, they have different metabolic profiles. Also, to make the experiments clinically relevant, 13-cis retinoic acid should have been used. Alternatively, if the use of ATRA was inevitable for experimental convenience, a few key additional experiments should have been included to demonstrate that they are biologically equivalent.

Response to Reviewer #5 Comment #1

We completely agree with the Reviewer that the decisions to use ATRA for the *in vitro* experiments and 13-cis retinoic acid for our mouse experiments should have been further justified in our text, and we appreciate the opportunity to clarify. We very carefully considered these decisions at the beginning of this study, which are justified based on the existing literature, and are further supported by additional experiments we performed ourselves.

Firstly, based on several decades of study, 13-cis RA is most widely accepted as a pro-drug, which has a better *in vivo* pharmacokinetics profile than ATRA, but whose biological activity *in vivo* stems primarily from its conversions (enzymatic or by other mechanisms) to the active

product ATRA. These ideas can be reviewed in e.g. Armstrong et al (BJC, 2005, PMID15714209), who conclude “Several lines of evidence suggest that 13cisRA acts as a prodrug for ATRA. 13cisRA has a much lower binding affinity than ATRA for the retinoic acid receptor (RAR) family of ligand-activated transcription factors”; many additional published findings support this stance (see e.g. PMID15826600). Thus, given some of this enzymatic conversion for 13-cis-RA to ATRA occurs in e.g. the liver, or non-tumor tissue, it made sense for us to remove this ‘enzymatic conversion’ variable from the *in vitro* mechanistic dissection of retinoic acid’s activity by treating directly with ATRA, as differences in enzymatic conversion (or e.g. total lack of the necessary metabolic enzyme) would have the potential to severely confound our results, potentially obscuring any mechanistic insights into the behavior of the pro-drug’s active product. Additional difficulties can also be introduced by 13-cis-RA to ATRA conversion upon exposure to small amounts of light or heat, introducing further difficult-to-control experimental confounding. Regardless, while we believe we have chosen the most rational approach given the existing evidence, we did also treat several of our cell lines (CHP-134, TGW, BE2C, BE(2)-M17) with both ATRA and 13-cis-RA, which is shown in Fig. S8C, where we observed almost no difference between the two drug enantiomers activities in these cell lines *in vitro*, suggesting that ATRA/13-Cis-RA could have largely been used interchangeably in the key cell lines used in our study.

Additionally, while we treated our mice with 13-cis-RA, we did also retrieve these tumor tissues after the study and used high-performance liquid chromatography (HPLC) to directly quantify the levels of both ATRA and 13-cis-RA in the tumors, where, consistent with existing reports on the interconversion of 13-cis-RA to ATRA, the average concentration of ATRA in the retrieved CHP-134 tumors was actually higher than the concentration of 13-cis-RA (Fig. S8D), again consistent with the idea that our decisions were reasonable and relevant to the activities of these compounds *in vivo*.

Notably, our choice to use ATRA in our *in vitro* experiments is not unique, neither in the study of retinoic acid in neuroblastoma (see e.g. Thiele et al, Nature, PMID3855502), nor the study of cancer pro-drugs more broadly. For example, similar strategies are applied when studying e.g. irinotecan, where most *in vitro* mechanistic studies treat with the most widely accepted active product, SN-38, rather than irinotecan itself, but *in vivo* studies treat mice directly with the irinotecan pro-drug (see e.g. Stewart et al, Cell Reports 2014 (PMID25437539)).

Finally, we absolutely agree with the Reviewer that these decisions are complex and arguably have a degree of subjectivity. Given this, we have now included a section in our Methods titled “Treatments with 13-cis-RA vs ATRA” (lines 1190-1211), which provides clear justification of our decisions on this important matter. We thank the Reviewer for raising this important point and hope we have clarified our experimental choices.

Reviewer #5 Comment #2

2. It is stated that CHP-134 is more sensitive than NB13 to ATRA, meaning sub micromolar IC50. However, Figure 1F shows that both 3-day- and 6-day-incubation show >1 uM.

Response to Reviewer #5 Comment #2

The values on the X-axis of these figures are on a log₁₀ scale and thus the IC50s of ATRA alone in CHP-134 (black curve) are substantially lower than 1 μ M. It is possible that the Reviewer has mistakenly read the IC50s from the red curve, which are ATRA + K02288 (the BMP inhibitor). However, the IC50s for CHP-134 are indeed sub micromolar. We hope we have clarified this important point.

Reviewer #5 Comment #3

3. What is the common gene mutation al status of the 16 cell lines (or the 12 neuroblastoma cell lines) that were used for the project? SMAD9 is also regulated by MYCN, PHOX2B, etc. Some cell lines responding better to ATRA than others can be also due to several other reasons. It may better explain if other clinical and genetic features of the cell lines are considered. For example, are any of the cell lines retinoic acid naive? What is the status of common mutations of neuroblastoma?

Response to Reviewer #5 Comment #3

We agree with the Reviewer. Indeed, we had also originally suspected that some of these highly responsive cell lines may have contained mutational events in key genes in the BMP signaling or retinoic acid signaling pathways. Thus, very early in this project, we performed whole genome sequencing on the cell lines CHP-134 and NB-13, but we identified no clear mutational events targeting BMP signaling, nor retinoic acid signaling. This is consistent with whole genome-sequencing analysis of neuroblastoma patient cohorts, where recurrent mutations targeting these signaling pathways have never been identified (see Brady *et al* for the largest reported analysis: <https://www.nature.com/articles/s41467-020-18987-4> (additional note: these sequencing analyses were performed by colleagues in our department at St. Jude, with whom we have corroborated these conclusions)). Thus, mutational events do not explain the strong responses to retinoic acid in some of these cell lines. However, we agree with the Reviewer that this will be a common question from readers, and it would be helpful for us to report these results in more detail. Thus, we have now included this information on common neuroblastoma driver mutational events in the cell lines used in this study Supplementary Table S1. We thank the reviewer for the insightful and helpful comment.

Reviewer #5 Comment #4

4. Apoptosis by ATRA was only observed in vitro, but not in xenograft tumors even at clinically relevant concentrations of ATRA, and apoptosis in metastatic neuroblastoma by ATRA was not shown, which is the key experiment for the project. 1) Therefore, the title that BMP signaling determines neuroblastoma cell fate and sensitivity to retinoic acid is limited to cell culture and largely speculative in clinical/in vivo settings. 2) Would this be related to the oxygen tension that was used in cell culture? Retinoic acid analogs are known to have different activity depending on oxygen tension (20% vs hypoxia). The apoptotic effect shown in 20% oxygen may diminish

when the cells are cultured in hypoxia. This issue is worth investigating, especially based on the data shown in Figure 6F.

Response to Reviewer #5 Comment #4

We thank the Reviewer for the insightful comments regarding hypoxia, which is known to affect response to many cancer drugs. Notably, we did not claim that factors other than BMP signaling may also in some circumstances modulate response to retinoic acid in neuroblastoma cells *in vivo* or *in vitro*. Like all other cancer drugs, several additive effects and interactions between factors would be necessary to explain all of the variation in retinoic acid response. We thank the Reviewer for pointing out the caveat, which we have now acknowledged explicitly in our Discussion section (lines 766-773) and will form the basis for future research in this area.

However, one of our main goals was to try to explain the existing clinical observation of differential RA activity at bone marrow metastatic sites in patients, relative to primary tumors. Thanks to the Reviewer's suggestion that hypoxia may also play a role, we have used the existing single-cell RNA-seq datasets to probe this idea, by assessing differential expression of known target genes of the hypoxia induced factors (HIF1A and HIF1B; whose target genes include VEGFA/B/C and SLC2A1 and whose mRNA levels can act as a proxy for hypoxia (proposed in PMID8855223). However, there was no evidence of substantial differences between the expression of these genes in neuroblastoma primary patient tumors vs neuroblastoma cells at bone marrow metastatic sites (see Figure below), thus it is not likely that differential hypoxia explains the existing clinical observation of site-specific RA activity, meaning this must be predominantly determined by other factors.

In contrast, as we showed in Fig. 6F (copied below), and with orthogonal assays, BMP signaling activity is strikingly different between neuroblastoma cells at bone marrow metastatic sites and primary tumors. Thus, given the collective body of evidence, we have concluded that this differential BMP signaling activity across different tumor sites provides a plausible explanation for the differential activity of retinoic acid at different tumor sites in patients. While we of course acknowledge that other factors also influence RA response, which is now discussed (lines 769-773), we believe that our primary conclusions are well supported by the existing data.

Reviewer #5 Comment #5

5. Figure 2: SMAD9, not other SMADs, was highly correlated with ATRA IC50. However, **(1)** the scope of the investigation includes SMAD4 in Figure 5 without a plausible explanation (it showed in ChIP-seq, and what is the explanation?). And **(2)** the summary illustration includes SMAD1/5/9 for apoptosis, but SMAD1 data has not been included. Also, **(3)** there is no evidence that SMAD4 contributed to apoptosis in response to ATRA.

Response to Reviewer #5 Comment #5

We thank the Reviewer for these insightful comments and value the opportunity to clarify these important details and components of our experimental design.

1. Given SMAD9 is somewhat understudied compared to other components of BMP signaling, we were surprised to learn there was no existing SMAD9 ChIP-seq antibody. We could obtain these data in CHP-134 by adding an endogenous HA-tag to SMAD9, which is a technical genome editing experiment, is slow, expensive, will not work in many cell lines, and is not a scalable solution. Thus, we only applied this strategy to CHP-134 initially. Given we found that the information from SMAD9 and SMAD4 were largely redundant, we applied only SMAD4 to the other cell lines, as there is a good ChIP anti-body for SMAD4 and this represented a scalable solution that could be broadly applied.

2. The summary figure includes phospho-SMAD1/5/9, for which data was collected in several of the preceding figures. The schematic was not a summary of just the data in Figure 5, but rather is an attempt to summarize the collective data and everything we had learned mechanistically to that point in the paper.
3. If BMP is contributing to ATRA response, SMAD4 is obligated to be involved because SMAD4 is an indispensable core component of this pathway (see e.g. PMID25401122).

We appreciate the Reviewer's attention to detail and hope we have been able to clarify these experimental decisions and technical details.

Reviewer #5 Comment #6

6. Figure 6: several types of primary samples, cultured cell lines and xenograft models of neuroblastoma were investigated. However, again their genetic and clinical features are not taken into consideration.

Response to Reviewer #5 Comment #6

We agree that this information has the potential to be informative to readers and we thank the Reviewer for the suggestion. Where available, we have now included, in Table S7, the clinical features of all samples profiled. Within each of these types of samples, there is a strong diversity of different genomic and clinical features (MYCN amplification, ATRX mutations, diagnosis/relapse etc.), but per Fig. 6F, the dominant determinant of BMP signaling activity remains the cellular context, e.g. bone marrow vs primary tumor. We thank the Reviewer for the helpful addition which will improve the transparency and scientific rigor of the manuscript.

Reviewer #5 Comment #7

7. Line 355 – 356: it is stated that the authors could rule out differentiation, but it is unclear how this was ruled out.

Response to Reviewer #5 Comment #7

We agree with the Reviewer that this was confusing and not clearly articulated. We have modified the text to now read "Thus, we speculated ATRA may sometimes induce cell cycle arrest, and *given that differentiation was associated with more viable cells* (Fig. 1F, Fig. 3B, Fig. 4B-C, ATRA + K02288 induced more differentiation and led to more viable cells than ATRA alone), we hypothesized these effects may be mediated by senescence" (line 362365), which should provide a better summary of the data and a clearer rationale for investigating RA-induced senescence in these cell lines. We thank the reviewer for helping us clarify this point.

Overall, we thank Reviewer #5 for their time, attention to detail, and expertise in reviewing the manuscript. They have made several important and insightful comments that have helped us greatly improve the rigor and quality of the revised paper. We hope that we have addressed

their concerns, and that the collective data broadly support the manuscript's primary conclusions.

Reviewer #6 (Remarks to the Author): expert in neuroblastoma models

Reviewer #6 overall summary

- What are the noteworthy results?

The manuscript by Pan, Zhang et al. provides experimental evidence of a previously undescribed mechanism-of-action for retinoic acid (RA) maintenance therapy in neuroblastoma (NB). Their data show that despite differentiation occurring in tumor cells subjected to RA (as previously known), induction of apoptosis and/or senescence by RA might be the main cause of its anti-neoplastic effect in specific sites. Moreover, the dependence of the RA effect on BMP signaling provides a plausible explanation for the site-specific effect of RA therapy on metastatic NB cells in the bone-marrow.

- Will the work be of significance to the field and related fields? How does it compare to the established literature? If the work is not original, please provide relevant references.

The results are important in the neuroblastoma field where the effect of long-term RA treatment has been questioned lately. The present data indicates an effect of the treatment on minimal residual disease with potential removal of metastatic tumor cells from the bone marrow. This is of immense importance and might lead to RA treatment being offered to more patients at risk of relapse. Additionally, knowledge of the mechanism-of-action can potentially lead to new combined therapies as suggested in the manuscript by combining senescence inducing RA with immunotherapy targeting senescent cells.

The authors suggest that "developmental hijacking" of other naturally occurring processes might be present also in other types of cancer, in particular other developmental tumors of the childhood. This could be true, and it is also possible that other tumors with difficult to treat bone/bone marrow metastasis could benefit from the knowledge presented in this manuscript, e.g. breast cancer and prostate cancer. However, that is not experimentally tested in the current manuscript.

- Does the work support the conclusions and claims, or is additional evidence needed?

The data supports the main conclusions of the paper, i.e. response to RA is dependent on BMP signaling and high BMP expression at specific sites such as in the bone marrow can lead to apoptosis/senescence (not differentiation) in neuroblastoma cells in those specific sites. However, more data would be needed to explain the underlying mechanism of preference for specific tumor cell lines to undergo either apoptosis, senescence or differentiation even in the presence of high BMP. Also, some important aspects of limitations to the study and how this affects the generalizability and clinical application of the results should be discussed more thoroughly.

- Are there any flaws in the data analysis, interpretation and conclusions?

See major and minor comments below.

- Do these prohibit publication or require revision?

Revision required.

- Is the methodology sound? Does the work meet the expected standards in your field?

Adequate controls and number of repeats.

Some of the major experiments were only performed in one or two cell lines (for example CRISPR screen). Confirmatory experiments and data analyses were performed in additional cell lines and published data, but see comment below on generalizability of the results.

More details are needed for animal experiments, such as tumor growth curves, weight of mice etc.

- Is there enough detail provided in the methods for the work to be reproduced?

The method section is detailed and extensive except for the in vivo section.

Response to Reviewer #6 overall summary

We thank the Reviewer for the kind summary and general comments on our paper. We have addressed each of the specific considerations in comments below.

Reviewer #6 major comments

Reviewer #6 comment #1

- The current debate of the efficacy of RA use for maintenance therapy should be acknowledged as well as the very few clinical trials investigating RA treatment in patient cohorts (eg. Peinemann et al. Cochrane Database Syst Rev. 2017 Aug 25;8(8):CD010685. doi: 10.1002/14651858.CD010685.pub3.)

Response to Reviewer #6 comment #1

We agree with the Reviewer that we could have added additional detail on this point, which was certainly a major factor in the initial motivation for our study. We have now included this additional information in the paper's Discussion stating "Indeed, this lack of clarity around RA's mechanism, combined with a relatively small number of sufficiently powered clinical studies, has fueled recent debate about RA's effectiveness in neuroblastoma (Peinemann, van Dalen et al. 2017)" (line 693-695).

Reviewer #6 comment #2

- Row 99-101) I disagree that hijacking of normal development processes have not been described as anti-cancer mechanisms before, one example is EMT (epithelial to mesenchymal transition).

Response to Reviewer #6 comment #2

We thank the Reviewer for the important comment and value the opportunity to clarify our point. We had intended to be more specific, that we are referring to the pharmacological induction of an organismal-developmental process, which we have now been clearer about (line 100-101). Our understanding is that EMT's canonical role in cancer is promoting metastasis. We are not aware of existing reports of pharmacological induction of EMT having a successful documented anticancer effect. If such a reference exists, we are of course happy to cite this and incorporate this idea into our discussion/conclusions. We agree with the reviewer that it is very important that we accurately place our work into the context of the existing literature, and we value the opportunity to improve this aspect of our paper.

Reviewer #6 comment #3

- Choice of cell lines: the molecular and clinical characteristics of the cell lines should be presented in a table (for example adding this to Table 1). This is important since the sensitivity to RA is very different between cell lines. The authors should also discuss the possibility of confounding biological factors related to RA sensitivity.

Response to Reviewer #6 comment #3

We agree with the Reviewer. Indeed, we had also originally suspected that some of these highly responsive cell lines may have contained mutational events in key genes in the BMP signaling or retinoic acid signaling pathways. Thus, very early in this project, we performed whole genome sequencing on the cell lines CHP-134 and NB-13, but we identified no clear mutational events targeting BMP signaling, nor retinoic acid signaling. This is consistent with whole genome-sequencing analysis of neuroblastoma patient cohorts, where recurrent mutations targeting these signaling pathways have never been identified (see Brady *et al* for the largest reported analysis: <https://www.nature.com/articles/s41467-020-18987-4> (additional note: these sequencing analyses were performed by colleagues in our department at St. Jude, with whom we have corroborated these conclusions)). Thus, mutational events do not explain the strong responses to retinoic acid in some of these cell lines. However, we agree with the Reviewer that this will be a common question from readers, and it would be helpful for us to report these results in more detail. Thus, we have now included this information on common neuroblastoma driver mutational events in the cell lines used in this study Supplementary Table S1. We thank the reviewer for the insightful and helpful comment.

Reviewer #6 comment #4

- Only RA-treatment sensitive cell lines are chosen for further work: NB13, CHP-134, D283

Med, SK-N-SH, TGW etc. It should be clearly stated that this is a choice and that this is a limiting factor when it comes to the generalizability of the results. Have the authors tried also less sensitive cell lines? How would they behave in the rest of the experiments? For example, what happens with them if exhibited to ATRA + FK506? Is it still only resulting in differentiation in those cases? Has it been tested?

Response to Reviewer #6 comment #4

We agree with the Reviewer that it was relevant to include some RA-resistant cell lines in some of our experiments. While the primary goal of our manuscript was to understand the extreme responses to retinoic acid, we had indeed included several RA-resistant cell lines in the manuscript, when relevant to our primary conclusions. This includes the drug combination screens (numerous comparatively RA resistant cell lines, including GIMEN ($\log_{10}(\text{IC}_{50}(\text{nM})) = 4.02$), CHP212 ($\log_{10}(\text{IC}_{50}(\text{nM})) = 3.8$), compared to 0.39 for CHP-134. We also interrogated baseline genome-wide gene expression and its correlates with both RA and the combination responses for these cell lines. We also generated/assembled a very comprehensive dataset on the comparatively resistant ($\log_{10}(\text{IC}_{50}(\text{nM})) = 4.39$) BE2 cells (BE2C and BE(2)-M17) (dataset details in Table S2) and presented a detailed summary of these findings Supplementary Figure S7, generating ChIP-seq to study the behaviors of retinoic acid and BMP transcription factors and gene expression changes over several time points in this resistant cell line. Much of this work is in the supplement, because while it provides important additional context, it is less relevant to the paper's core conclusions relative to the results in the hyper-responsive cell lines. We value the opportunity to clarify these details of our manuscript.

Reviewer #6 comment #5

- Synergy: Figure 2G and S2K/L display synergy scores for RA + FK506 or Rapamycin. It is stated that ATRA showed "synergistic effects with both compounds in all NB cell lines" with one exception. It is not clear to me how SK-N-AS and GI-ME-N treated with FK506 can be synergistic, a cut-off of what is considered synergistic scores should be added. Also in figure S2K, L, O, P, the cut-off is not clear. The additive effect is shown with a red dotted line, but some statistical testing will be necessary to claim them to be synergistic.

Response to Reviewer #6 comment #5

We thank the reviewer for highlighting the need for further quantification of our drug synergy analyses. We have revised the manuscript to clearly define the criteria used to determine synergy and have included statistical analyses to support our claims. We now specify that a synergistic effect was considered significant when two criteria were met:

1. Combined treatment effect exceeds additive expectation: The observed effect of the combination treatment is greater than the sum of the effects of each treatment alone, represented by the red dotted line in updated Figures S2K, S2L, S2O, and S2P.
2. Statistical significance: The difference between the combined treatment effect and the additive expectation is statistically significant, with a $P < 0.05$. We have conducted statistical tests (two-tailed one-sample Student's t-tests) to evaluate the significance of

the synergistic effects. The *P*-values for each combination treatment are now also provided in Supplementary Table S1 and depicted in Supplementary Figure S2. This addition ensures that our claims of synergy are supported by rigorous statistical evidence.

Based on these updated criteria, ATRA still showed synergistic effects with both compounds in the vast majority of the neuroblastoma cell lines. Those cases with no significant synergy are NGP treated with ATRA + FK506 and ATRA + rapamycin, SK-N-AS with ATRA + FK506, and SK-N-FI with ATRA + rapamycin. This updated information is now detailed in the Results (lines 223–226) and the Methods section (lines 996–999). We appreciate the Reviewer's insightful comments, which strengthen the validity of our findings and enhance the rigor of the manuscript.

Reviewer #6 comment #6

- Fig 4: row 356) ATRA can sometimes induce cell cycle arrest which is indicated by the CRISPR screen. However, how can you “rule out differentiation” when you have just showed that in 4A and B? Here is a logical step missing that needs to be filled or explained better.

Response to Reviewer #6 comment #6

We agree with the Reviewer that this was confusing and not clearly articulated. We have modified the text to now read “Thus, we speculated ATRA may sometimes induce cell cycle arrest, and given that differentiation was associated with more viable cells (Fig. 1F, Fig. 3B, Fig. 4B-C, ATRA + K02288 induced more differentiation and led to more viable cells than ATRA alone), we hypothesized these effects may be mediated by senescence” (356357), which should provide a better summary of the data and a clearer rationale for investigating RA-induced senescence in these cell lines. We thank the reviewer for helping us clarify this.

Reviewer #6 comment #6

- Chip seq, why use BE(2)-M17 cells here? Are they used as diff cells anywhere else? Again, characteristics of the different cell lines and motivation (including supporting data) should be included in a table and explained. S4 should be more detailed in calling.

Response to Reviewer #6 comment #6

The BE2 cell line (and derivatives, BE2C and BE(2)-M17) are used as a comparatively RA resistant models, which differentiate upon RA treatment. They were included in the initial drug screening data and comprehensively profiled in our Fig. 5 and Fig. S7. Retinoic acid-induced differentiation of these cells is well established, previously reported in, for example, (Andres, Keyser et al. 2013; PMID23597229), which we have now explicitly cited on line 437. Thus, these are a rational choice for these experiments with good support in the existing literature.

Reviewer #6 comment #7

- Fig 5K, the conceptual overview should be simplified to better convey the hypothesis. Also, in

figure 5 there is no data relating to the senescence (TGW) or differentiation (BE(2)-M17) cell results. I think the massive S4-S7 should be reorganized and some data added from them to the main figure to support the senescence and differentiation hypotheses.

Response to Reviewer #6 comment #7

We thank the Reviewer for the comment. We agree with the Reviewer that we have generated a lot of high throughput data, which has helped us understand the mechanistic basis for the interactions between BMP signaling and retinoic acid. We agree it is often difficult to display such data within the constraint of a conventional academic manuscript, for example, our Figure 5 already contains 11 dense panels of data. Thus, given we feel it is very important to show as much of these data as possible, to be as transparent as possible about the basis for our conclusions, it was necessary for us to place much of these results in the supplement, as they clearly cannot fit in the main figure. We endeavored, as best we could, to place the highest priority results in the main figure, though we realize this comes with a high degree of subjectivity. We understand that it is important to create a simplified schematic. We spent considerable time working on simplifying this schematic, but it is not clear to us how the schematic we are showing could be further simplified at this point and still maintain the minimal necessary information, which seems highly subjective, however we believe the schematic strikes as good balance as we were able to achieve.

Reviewer #6 comment #8

- Importantly: to the best of my understanding the analysis of chip-seq data does not provide an explanation to WHY the three different cell lines get different downstream binding patterns from SMAD4 activation. This should be further investigated or discussed as a limitation. Now it is only referred to in the sentence on row 497-499 "...active SMAD transcription factors activates apoptosis or senescence, depending on baseline binding patterns of BMP effector SMADs..."

Response to Reviewer #6 comment #8

We thank the Reviewer for the comment. The current manuscript we did not claim/conclude to have worked out exactly why the three cell lines have different downstream binding patterns. We stated in the Discussion, for example, that "the relationship between apoptosis and senescence is multifaceted and still poorly understood (Childs, Baker et al. 2014), and it is likely additional factors, including TP53 mutation (Tavana, Benjamin et al. 2010), which is evident in e.g. our TGW cell line, may also contribute to tipping neuroblastoma cells towards this outcome.". This is an active field research and can be reviewed in (Childs 2014, PMID25312810). Rather, the current manuscript focuses on the broader interactions between BMP signaling and retinoic acid, but we agree that, while not within the scope of the current study, further dissecting the basis for these binding patterns will form a valuable basis for many future studies.

Reviewer #6 comment #9

- In vivo study: provide a rationale for including a chemotherapy treatment group. As far as I can

see no results except survival is presented for this group.

Response to Reviewer #6 comment #9

We thank the Reviewer for pointing out that we did not explicitly state the rationale for this treatment group. In the original study design, the chemo-treated mice were intended as a positive-control reference point for the retinoic acid group (the vehicle-treated and chemo-treated mice can be thought of as positive and negative control groups). We have now included this in the figure legend on line 649-650 stating that “The IRN+TMZ group is intended as a standard-chemo positive control.”. We thank the Reviewer for helping us clarify this detail.

Reviewer #6 comment #10

- PHOX2B is used as a general NB marker in analyses of patient tumor material (fig 6). From a developmental perspective, PHOX2B is only expressed in cells of the sympathoadrenal lineage (for example Zeinelden et al. 2022, Bedoya-Reina et al. 2021). Thus, in many publications on NB cell states, PHOX2B has been used as an adrenergic marker, e.g. has been considered not to be expressed in NB cells of a more immature/mesenchymal phenotype (Boeva et al. 2017, van Groningen et al. 2017, Patel et al. 2024 preprint). Other publications have shown that treatment resistant cells can be of the more mesenchymal phenotype (Westerhout et al. 2021, Manas et al. 2022). The presence of downstream BMP targets in non-PHOX2B cells could thus be of major clinical importance. Has this been investigated? Could an additional NB cell marker be used?

Response to Reviewer #6 comment #10

We thank the Reviewer for the comment and yes this was investigated. PHOX2B is currently the most widely accepted marker of neuroblastoma cancer cells, and it is used as a diagnostic in routine clinical pathology, including at bone marrow metastatic sites (see PMID18838715). In neuroblastoma single-cell RNA-seq analysis of bone marrow samples, there was no evidence reported for mesenchymal-like cells in any of the samples analyzed (PMID37365178). Our re-analysis of this dataset reached identical conclusions, which motivated our choice of the PHOX2B as the conventional marker of neuroblastoma cells in the IF study. Thus, while it is impossible to claim any marker gene is definitively expressed in every cell, based on the existing evidence from single-cell RNA-seq and the clinical literature, PHOX2B is an appropriate marker for neuroblastoma cells in bone marrow.

Reviewer #6 comment #11

- A paragraph stating the limitations of the study should be included in the discussion: limited number of cell lines, no patient-derived cells, no testing in clinical cohorts to investigate the level of expression of BMP in primary tumor sites and metastatic sites in patients etc.

Response to Reviewer #6 comment #11

We agree with the Reviewer that additional details on the limitations of the current study and the scope for future studies could represent valuable additions to our Discussion section (line 762773). We thank the Reviewer for helping us further clarify these important points.

Reviewer #6 comment #12

- An overview of BMP expression in different organs would also strengthen the clinical application suggestions. How is the BMP expression levels in other common metastatic sites? Liver, lung etc. BMP (e.g. BMP2 and 4) could be investigated in clinical cohorts, in a TMA for protein expression or RNAseq data for example.

Response to Reviewer #6 comment #12

We thank the Reviewer for highlighting that neuroblastoma also metastasizes to other sites, where BMP may be variable. However, the existing clinical observations in neuroblastoma suggest that RA response is selective to the bone marrow (see e.g. Villablanca et al, PMID7707116, for an early example of this). While we agree in principle that investigating BMP signaling at many or all metastatic sites in neuroblastoma patients could be interesting, these investigations are currently limited by our access to the appropriate data and patient samples. Notably, our manuscript does not preclude nor state one way or another the expectation around BMP signaling at other metastatic sites. We have now modified our Discussion section to state this explicitly (lines 768-772). However, we reiterate that the relevant existing clinical observations pertain to the bone marrow, which we have thus investigated in detail, and supports the primary conclusions of this paper.

Reviewer #6 comment #13

- The study, and particularly the clinical application possibilities, would be strengthened by an in vivo study of a metastatic model. Or by injecting cells both sub-cutaneous/orthotopic and in mouse femur to see differences in effect in vivo at different sites. For clinical application purposes the data from PDX models under these circumstances would be informative for how heterogeneous tumors behave. Possibly a spatial transcriptomic readout could inform on differential response in different subgroups of NB cells.

Response to Reviewer #6 comment #13

We agree with the Reviewer that this would represent an excellent future direction. Indeed, we hope to secure R01 level funding and assemble an interdisciplinary team to pursue a similar direction over the next 5-7 years. Currently, animal models of disseminated bone marrow disease in neuroblastoma are very challenging, as the prevailing observation is that the mice die of other causes before developing robust bone marrow metastases that can be robustly studied, unlike in humans where this site is colonized sooner. Indeed, the bone marrow models of neuroblastoma, which have been reported to have good penetrance, achieve this by interfering specifically with apoptosis (see e.g. PMID23536557), which would render those models inappropriate for our study of RA-induced apoptosis. We agree with the Reviewer that working out how to develop robust genomically diverse bone marrow metastatic mouse models of

neuroblastoma patient derived samples, and profiling these with spatial transcriptomics under RA treatment will be an excellent direction for the future, which we now Discuss on lines 771-773. However, given the resources required and technical/interdisciplinary challenges, these results will likely be much higher profile than the current manuscript.

Reviewer #6 Minor considerations

Reviewer #6 comment #14

- Row 82) A 2012-study should not be called “recent”.

Response to Reviewer #6 comment #14

We thank the Reviewer for highlighting this inconsistency. We have now changed “recent” to “previous” (line 82).

Reviewer #6 comment #15

- 113) Previous 3-day exposure in literature is replaced in your study by 6-days exposure motivated by “in vivo pharmacokinetic profile”. How is this relevant for your in vitro exposures?

Response to Reviewer #6 comment #15

Retinoic acid is less toxic than most standard chemotherapeutics and thus, pharmacokinetics studies have shown that it can be tolerated for longer duration exposures in patients (usually a 6 day cycle), compared to conventional cytotoxic chemotherapeutics, such as e.g. cisplatin. Thus, in vitro studies of retinoic acid consider a 6-day treatment to be more pharmacologically relevant than a 3-day treatment, which is more appropriate for cytotoxic drugs like DNA damaging agents. The pharmacokinetics data can be reviewed in PMID17224928, which we have cited to support this statement. We thank the Reviewer for the opportunity to clarify this point

Reviewer #6 comment #16

- Table S1: There are 18 cell lines here when only 16 are in the results. It should be clarified somewhere that the last two are used in specific experiments only.

Response to Reviewer #6 comment #16

We thank the Reviewer for the helpful comment and have now included a footnote in Table S1 stating that these cell lines were only used in the synergy experiments.

Reviewer #6 comment #17

- 115-117) Discussion of differentiation and the lack of differentiation after 3 days (why not 6 days here?) already in results section 1, is called in figures S3A-B. This should be presented already in Fig 1/S1 if the lack of differentiation is used as motivation for investigating apoptosis.

Response to Reviewer #6 comment #17

We thank the Reviewer for this suggestion, which we agree is an improvement. The previous Figure S3A-B has been moved to current Figure S1B-C now.

Reviewer #6 comment #18

- 127) Figure 1E and 1D are called out of order.

Response to Reviewer #6 comment #18

We thank the Reviewer for spotting this detail, which has now been fixed (line 129).

Reviewer #6 comment #19

- 188) Fig S2A wrongly called. It is again called wrongly in row 192. Consider the correct way to refer to this figure.

Response to Reviewer #6 comment #19

We thank the Reviewer for spotting this oversight. We have now updated this to correctly call this figure in using the correct context (line 194-195).

Reviewer #6 comment #20

- 191) Fig 2K wrongly called.

Response to Reviewer #6 comment #20

We thank the Reviewer for spotting this typo which has now been updated to the correct label "2B".

Reviewer #6 comment #21

- 444) Fig 5D should be called instead of 5C

Response to Reviewer #6 comment #21

This has been updated to the correct panel "5D" (line 455).

Reviewer #6 comment #22

- 491) Fig S7F should be called instead of S8F

Response to Reviewer #6 comment #22

This has been corrected to S7F (line 502).

Reviewer #6 comment #23

- 571) Provide a reference or data supporting "... because BMP signaling is artificially inflated in

vitro.” Evidence that it is higher is provided, but according to table 1 no additional BMP is added in the medium. What makes BMP signaling high in vitro?

Response to Reviewer #6 comment #23

We stated here “These mouse data could plausibly indicate that the extreme/acute RA responses observed in cell lines are not relevant in vivo (nor clinically) because BMP signaling is artificially inflated in vitro”. This was intended as a postulate used to motivate the results that follow this, rather than being intended as a conclusion. We now realize that this was confusing and have reworded this to be clearer, now stating “These mouse data could plausibly indicate that the extreme/acute RA responses observed in cell lines are not relevant in vivo (nor clinically), which could arise if, hypothetically for example, BMP signaling is artificially inflated in vitro” (lines 584-585). We thank the Reviewer for helping us to clarify this point.

Reviewer #6 comment #24

- Fig 6: reorganize order of photos in H.

Response to Reviewer #6 comment #24

We thank the Reviewer for the helpful suggestion. We have now re-organized the images in Fig. 6H, which we now realize were in the incorrect order.

Overall, we thank Reviewer #6 for their time and expertise and their extremely helpful and detailed review of the manuscript, which is now much improved. We hope that we have addressed the concerns, and we believe that our manuscript’s primary conclusions are well supported by the data we have shown.

In conclusion, we thank the Reviewers for their time and efforts in helping us improve our manuscript, which is now much better. Importantly, we believe our manuscript’s main claims and conclusions are well supported by the collective data we have generated. We hope we have addressed all concerns and we look forward to a positive decision on the resubmission.

Sincerely,

Paul Geeleher, PhD
Associate Professor
Comprehensive Cancer Center
Dept. of Comp Biology
School of Biomedical Sciences
St Jude Children's Research Hospital,
Memphis TN, USA
paul.geeleher@stjude.org

Min Pan, PhD
Comprehensive Cancer Center
Dept. of Comp Biology
School of Biomedical Sciences
St Jude Children's Research Hospital,
Memphis TN, USA
Min.pan@stjude.org

John Easton, PhD
Director, Genomics Laboratory,
Department of Comp Biology
St Jude Children's Research Hospital,
Memphis TN, USA
John.easton@stjude.org

Dear 6 peer Reviewers,

We thank the Reviewers for their assessment of our revised manuscript. We have carefully addressed the final comments point-by-point below. We have included both tracked and clean copies of the updated manuscript and supplementary materials. All changes to the manuscript are referred to by their *line number in the tracked copy of the new updated manuscript*. We are extremely grateful to the Reviewers for their time and expertise and value the opportunity to address the final comments, where we have clarified some minor remaining concerns.

REVIEWER COMMENTS:

Reviewer #1 (Remarks to the Author)

The authors have addressed my comments in the revised manuscript.

Response to Reviewer #1 final comments:

We thank Reviewer #1 for their final comments on the paper and we are glad our revisions have been able to address their concerns.

Reviewer #2 (Remarks to the Author)

Reviewer #2 Overall summary:

In this revised version of the manuscript, the authors significantly improved the coherence of the study. The connections between the topics addressed in the manuscript was more comprehensively handled and the limitations of the study were adequately pointed out and discussed. Our comments were thoroughly and satisfactorily addressed.

Response to Reviewer #2 Overall summary:

We thank the reviewer for identifying a few additional minor typographical and grammatical errors, which we have now fixed and addressed individually below.

Reviewer #2 suggested minor corrections:

Reviewer #2 minor correction #1:

*Minor grammatical and narrative errors should be corrected. Some examples:
-Line 124: After removal of “unbiased” the article should be “a”

Response to Reviewer #2 minor correction #1:

We thank the reviewer for identifying this typographical error. We have corrected “an” to “a” on line 124 of the manuscript.

Reviewer #2 minor correction #2:

-Line 210: “as a negative regulator” when mentioning plural amount of inhibitors

Response to Reviewer #2 minor correction #2:

We thank the reviewer for identifying this typographical error. We have corrected “a negative regulator” to “negative regulators” on line 210 of the manuscript.

Reviewer #2 minor correction #3:

-Line 368: Switching from past tense to present tense (“validate”)

Response to Reviewer #2 minor correction #3:

We thank the reviewer for identifying this typographical error. We have corrected “validate” to “validated” on line 368 of the manuscript.

Reviewer #2 minor correction #4:

-The treatment periods were indicated as “six days / three days” or “6 days / 3 days” in different parts of the manuscript, it should be addressed either as written or number symbols

Response to Reviewer #2 minor correction #4:

We thank the reviewer for highlighting this formatting inconsistency. We have now standardized the treatment periods as number symbols throughout the manuscript. Specifically, we have changed “six days” to “6 days” on lines 112, 284, 296, and 298; “three days” to “3 days” on lines 116, 119, 142, 151, 295, 543, and 675; and “one day” to “1 day” on lines 455 and 793. Additionally, we have made corresponding changes in the legends of the supplementary figures: “one day” to “1 day” in the legend of Supplementary Figure 5, “three days” to “3 days” in the legends of Supplementary Figures 1, 4, 5, and 6, and “six days” to “6 days” in the legends of Supplementary Figures 5 and 8.

Reviewer #2 minor correction #5:

*Particularly in the Introduction, Results and Discussion sections, some points were addressed with sentences that were 4-5 lines long. Although this does not affect the results, it reduces the clarity of the work.

Response to Reviewer #2 minor correction #4:

We agree with the reviewer, and we have now endeavored to break up some of the longer sentences added during the first round of revisions, such as that on lines 191 and on line 361.

Overall, we thank Reviewer #2 for their rigorous and fair approach to reviewing this manuscript. We hope that we have addressed the final concerns and that our manuscript is now much improved and ready for publication.

Reviewer #3 (Remarks to the Author)

Response to Reviewer #3

We thank Reviewer #3 for their efforts in co-reviewing this manuscript and we are glad we could address their concerns.

Reviewer #4 (Remarks to the Author)

Overall, the authors have addressed the majority of my concerns. However, I believe there are two critical points that require further attention prior to publication:

Reviewer #4 comment #1:

The authors assert that RA treatment decreases BMP signaling activity. However, the SMAD4 binding signal at the genome is markedly elevated post-RA treatment, while changes in SMAD9 binding appear to be more subtle. Given the increased nuclear SMAD4 signals following RA treatment, this observation does not support the conclusion that RA diminishes BMP signaling. To clarify this discrepancy, the authors should perform SMAD1/5 and SMAD2/3 ChIP-seq analyses to further investigate the increased SMAD4 chromatin binding.

Response to Reviewer #4 comment #1:

We value the opportunity to clarify some important points related to (1) SMAD4 levels and (2) SMAD9 levels during retinoic acid treatment in our experiments.

- (1) SMAD4 levels alone should never be used as a readout of BMP signaling activity. The reason for this is that SMAD4 is ***not*** unique to the BMP signaling pathway. SMAD4 levels change in response to other signaling pathways, for example, the TGF-beta/activin pathway (for a detailed review, please see PMID 37714133) and overall SMAD4 levels alone cannot and should not be used to infer BMP signaling activity levels. There are numerous other sources of this information, including established textbooks such as *The Molecular Biology of the Cell* (described on page 865 of the sixth edition). Our study has used the widely accepted *de facto* assays for establishing BMP signaling activity, specifically, the levels of phospho-SMAD1/5/9 and the expression of the canonical BMP-target ID family of genes, which are the standard approaches applied throughout the literature, the results of which are always consistent with our study's main conclusions.
- (2) While SMAD9 levels alone are also ***not*** a reliable means to estimate overall BMP-signaling activity, the change of SMAD9 signal in our data following RA treatment is quite substantial; the decrease is striking (>10-fold) after 3 days and 6 days of treatment (see e.g. Fig. S4F).

We thank the reviewer for their attention to detail and helping us improve the clarity of the manuscript. We hope we have clarified our experimental decisions and why we have reached our main conclusions, which are well supported by the data we have shown.

Reviewer #4 comment #2:

In Figure 6J, the internal control should be total SMAD1/5/9, rather than GAPDH. Total SMAD1/5/9 protein levels are the established standard for assessing phosphorylated SMAD1/5/9 changes in BMP signaling research. While the authors state they cannot draw conclusions based on total SMAD1/5/9 in their experimental context, they should at least provide a more thorough discussion of this limitation within their study.

Response to Reviewer #4 comment #2:

The results with total SMAD1/5/9 as an internal control are now included in Supplementary Figure 12. We thank the Reviewer again for giving us this opportunity to improve the manuscript.

Reviewer #4 comment #3:

Additionally, the phosphorylation of SMAD1/5/9 depicted in Figure 6J does not show a significant increase following BMP1/7 treatment compared to the vehicle control. The authors should offer further explanation regarding this observation.

Response to Reviewer #4 comment #3:

We thank the reviewer for the important question. There is no expectation that every BMP ligand will have the same effect on every cell type. There are at least 14 different BMP ligands that have different affinities for at least 7 different BMP receptors. These receptors can be expressed at different levels in different cells, as can intermediate factors with which they interact, as well as downstream targets and interacting partners in the signaling cascade. Thus, there are a multitude of factors that can determine how any given BMP ligand effects a specific cell and it is impossible to expect these effects would be identical in different cell lines. Importantly, given the complexity of these cell-specific effects our manuscript (and its core conclusions) has focused on the broader/general behavior of BMP signaling overall in determining retinoic acid response, but dissecting these specific details will be an excellent platform for future studies. We thank the reviewer for the opportunity to clarify this important aspect of our study design and for the excellent suggestion as regards directions for future work.

Overall, we thank Reviewer #4 for their time and expertise and value the opportunity to clarify the important points that they have raised. We hope that we have addressed the final concerns and that our manuscript is now much improved and ready for publication.

Reviewer #5 (Remarks to the Author)

I appreciate that the authors tried to address all comments from this reviewer. However, using ATRA as an active metabolite of 13-cisRA is misplaced and not clinically irrelevant to the treatment of high-risk neuroblastoma. Multiple clinical pharmacokinetic studies of 13-cisRA in neuroblastoma reported that 4-oxo-13-cisRA is the major metabolite in patients. Also, ATRA is detected at negligible levels. Please refer to Veal GJ, Cole M, Errington J, Pearson AD, Foot AB, Whyman G, Boddy AV; UKCCSG Pharmacology Working Group. Pharmacokinetics and metabolism of 13-cis-retinoic acid (isotretinoin) in children with high-risk neuroblastoma - a study of the United Kingdom Children's Cancer Study Group. Br J Cancer. 2007 Feb 12;96(3):424-31. doi: 10.1038/sj.bjc.6603554. Epub 2007 Jan 16. PMID: 17224928; PMCID: PMC2360017. Therefore, the translational implications of this study remain in question.

Response to Reviewer #5 final comment:

We thank the reviewer for the important reference to this pharmacokinetics study, which we have now cited. This pharmacokinetics study measured various 13-cis-RA metabolites in

patients at 1-day and at 2-weeks post-treatment, where 4-oxo-13-cisRA levels were reported as substantially higher than ATRA levels (i.e. the study reported that 4-oxo-13-cisRA is the “major” metabolite of 13-cis-RA; “major” implying steady state *abundance*). However, this pharmacokinetics study does not claim—nor does it provide any evidence—that 4-oxo-13cisRA is the primary *active* metabolite of 13-cisRA. Indeed, the study states that “there was a greater likelihood of relapse for patients with higher day 14 peak 4-oxo-13cisRA plasma concentrations (P=0.014; Cox regression analysis)”—the exact opposite of what would be expected if 4-oxo-13cisRA was a primary *active* metabolite of 13-cisRA. Indeed, the elimination half-life of ATRA is much shorter than 13-cis-RA or 4-oxo-13cisRA (45 minutes vs 20 hours, see e.g. PMID 10388008), thus it is not expected that ATRA would accumulate at steady state to the same degree as 4-oxo-13cisRA. Critically, the cited pharmacokinetics study also measured retinoid concentrations in *plasma*, not in tumor tissue; *tumor* would be the relevant site for anti-neuroblastoma activity, which we measured by HPLC in our Fig. S8D. This is particularly important here because isomerization of 13-cis-RA to ATRA has been shown to predominantly occur within cells and is negligible in extracellular environments like plasma (PMID 11841795). We reiterate that in our study we did also treat some of our cell lines with 13-cis-RA, and that 13-cis-RA’s activity in neuroblastoma cells was virtually identical to ATRA (Fig. S8C). We also reiterate that numerous published lines of evidence, from both cancer studies and model organisms, point towards ATRA as the primary *active* metabolite of 13-cis-RA (e.g. PMID 1352213, PMID 10951254, detailed review related to pediatric oncology in PMID 15826600). Thanks to the reviewer’s excellent suggestion, we have now further discussed these important details, citing the paper showing the accumulation of 4-oxo-13cisRA in patient’s plasma observed during a 13-cis-RA pharmacokinetic study (lines 1208-1210). We agree that the literature around retinoids and their metabolites is complex, with some conflicting/confusing claims, and that clearly presenting these points is very important for the readership of our paper. Indeed, we have now dedicated a full section of the paper to these ideas (“Treatments with 13-cis-RA vs ATRA”, lines 1203-1227). We thank the reviewer for helping us further clarify these details, and value the opportunity to improve the rigor of our paper.

Reviewer #6 (Remarks to the Author)

I believe that the authors have responded to and clarified most of my concerns regarding the manuscript. I appreciate this and think that the good manuscript has become even better.

Reviewer #6 comment #1

In response to my comment #5 about drug synergy, the authors have now included a definition of synergy (vs additivity) and provided results from statistical testing. This could be sufficient, but I wonder why a well established synergy score such as Bliss independence or Zip synergy score was applied? But this is up to the authors.

Response to Reviewer #6 comment #1:

We thank the reviewer for the comment. We emphasize that we did in fact calculate Zip synergy scores from the data (Fig. 2, Supplementary Figure 2), but note that the Zip software does not calculate *P*-values along with these scores (nor do Bliss scores), thus, we have calculated the *P*-values using the one-sample Student’s t-test, testing the deviation from the expected synergy score. We thank the reviewer for the helpful suggestion.

Reviewer #6 comment #2

However, my comment regarding the details of the animal studies has been missed or disregarded. I believe that more data is needed to support the conclusions drawn from the animal models:

- *Is the methodology sound? Does the work meet the expected standards in your field?*

More details are needed for animal experiments, such as tumor growth curves, weight of mice etc.

- *Is there enough detail provided in the methods for the work to be reproduced?*

The method section is detailed and extensive except for the *in vivo* section." Some additional information about the mice has been added in Methods 1146-47 in response to another reviewer question, but tumor growth curves and mice weight are still not included.

Response to Reviewer #6 comment #2:

We thank the reviewer for the additional helpful comments regarding our animal study. As a point of clarification, tumor growth curves were shown in Supplementary Figure 8, which we had collected by the Xenogen *in vivo* imaging system. We have now added the mice weights for the 13-cis-RA and Irinotecan + Temozolomide treated groups for both the CHP-134 and BE2 cell lines, now shown in Supplementary Table 7. We thank the reviewer for noticing this omission, which was an oversight on our part. Note: in both treatment groups the weight of the mice was stable across the course of treatment, indicating good tolerability. We have also added additional text describing the evidence informing our dosing/schedule for 13-cis-RA in mouse (lines 1155-1171), which was informed by existing pharmacokinetics studies in reported in humans and animal models. We thank the reviewer for helping us bolster the rigor with which this important aspect of our study is presented.

In conclusion, we thank the 6 Reviewers for their time and efforts in helping us to further improve our manuscript and clarify the final remaining points. Importantly, we believe our manuscript's main claims and conclusions are well supported by the collective data we have generated. We hope we have addressed all concerns and we look forward to a positive decision on the final resubmission.

Sincerely,

Paul Geeleher, PhD
Associate Professor
Comprehensive Cancer Center
Dept. of Comp Biology
School of Biomedical Sciences
St Jude Children's Research Hospital,
Memphis TN, USA
paul.geeleher@stjude.org

Min Pan, PhD
Comprehensive Cancer Center
Dept. of Comp Biology
School of Biomedical Sciences
St Jude Children's Research Hospital,
Memphis TN, USA
Min.pan@stjude.org

John Easton, PhD
Director, Genomics Laboratory,
Department of Comp Biology
St Jude Children's Research Hospital,
Memphis TN, USA
John.easton@stjude.org

Dear Peer Reviewers,

We thank the reviewers for their detailed assessment of the paper and we have addressed the final comments. As requested by Reviewer #7, we have added additional results estimating TGF- β signaling activity under treatment with RA, and additional quantification and normalization for the pSMAD1/5/9 Western Blots. We have discussed all remaining issues raised by the Reviewers. We have included both tracked and clean copies of the updated manuscript and supplementary materials. All changes to the manuscript are referred to by their *line number in the tracked copy of the new updated manuscript*. We thank the Reviewers for their assessment of the paper and we hope the final manuscript is much improved. We provide a detailed point-by-point response below.

REVIEWER COMMENTS

Reviewer #4 (Remarks to the Author):

Reviewer #4 comment #1

SMAD9 levels suggest complex crosstalk, necessitating CHIP-seq analysis for SMAD1/5 and SMAD2/3 to determine whether BMP or TGF- β signaling is predominantly affected.

Response to Reviewer #4 comment #1

We thank the reviewer for the important comment. As we mentioned in the previous rounds of review, our manuscript concluded that BMP signaling activity determines response to retinoic acid. Our manuscript drew no conclusions about the activity of TGF- β , nor any other specific signaling pathways. However, per the recommendation of Reviewer #7 we have now also estimated TGF-beta signaling activity in response to RA in our CHP-134 cell lines. Based on the expression of the pathway's target genes, TGF-beta signaling activity was largely unchanged during 6 days of retinoic acid treatment (see new Supplementary Figure S5I). We value the opportunity to clarify this point.

Reviewer #4 comment #2

Concerns about BMP signaling activation remain unresolved due to potential confounding changes in total-SMAD1/5/9 levels, which must be disentangled from phospho-SMAD increases.

Response to Reviewer #4 comment #2

We thank the reviewer for their important comment. As we mentioned in both previous rounds of review, blots for total SMAD1/5/9 were added to Supplementary Figure S12. We also quantitated BMP signaling activity by complementary orthogonal assays (e.g. the expression level of the ID family of BMP target genes). Per the recommendation of Reviewer #7 we have now also normalized the pSMAD1/5/9 levels to the total SMAD1/5/9 levels and included this

additional quantitation in Table S10. We thank the reviewer for their important comment. We value the opportunity to clarify this important point.

Reviewer #4 comment #3

Discrepancies in SMAD9 signal trends between figures (e.g., Fig. S4F vs. S5A) require careful review for consistency.

Response to Reviewer #4 comment #3

We thank the reviewer for the comment. Per the recommendation of Reviewer #7, we value the opportunity to clarify: These figures show different things (examples of SMAD9 peaks at a few specific genes vs SMAD9 peaks globally). It is common to show both of these views of ChIP-seq data. As mentioned in the previous rounds of review, neither should be used to infer overall BMP signaling activity. We value the opportunity to clarify this important point.

Reviewer #4 comment #3

Additionally, the Venn diagram in Fig. 5B needs correction or clarification regarding its proportional calculations.

Response to Reviewer #4 comment #3

As pointed out by Reviewer #7, the proportions in the Venn diagram add up to 99.9% (i.e. there is rounding error). We thank the reviewer for their attention to detail.

Reviewer #4 comment #4

Below are the detailed comments: In my previous comments, I noted that RA treatment may interfere with the TGF- β superfamily, as evidenced by the significantly increased nuclear SMAD4 signal shown in Fig. 5A.

Response to Reviewer #4 comment #4

We thank the reviewer for the important question. Per the recommendation of Reviewer #7 we have now also estimated TGF-beta signaling activity in response to RA in our CHP-134 cells. Based on the expression of the pathway's target genes, TGF-beta signaling activity was largely unchanged during 6 days of retinoic acid treatment (see new Supplementary Figure S5I). We value the opportunity to clarify this important point.

Reviewer #4 comment #5

While I acknowledge that changes in SMAD4 levels alone are insufficient to confirm BMP signaling activation, I find it perplexing that the authors report an increase in SMAD4 levels alongside a subtle decrease in SMAD9 levels. This apparent discrepancy complicates the interpretation of BMP signaling inhibition following RA treatment. To address this issue, I recommend performing ChIP-seq for SMAD1/5 and SMAD2/3 to determine whether RA

treatment predominantly impacts the BMP or TGF- β signaling pathways. Although this additional experiment may require more time, it would provide critical insights into the interaction between RA treatment and these signaling pathways.

Response to Reviewer #4 comment #4

Per the recommendation of Reviewer #7 we have now also estimated TGF-beta signaling activity in response to RA in our CHP-134 cell lines. Based on the expression of the pathway's target genes, TGF-beta signaling activity was largely unchanged during 6 days of retinoic acid treatment (see new Supplementary Figure S5I). We thank the reviewer for the excellent suggestion.

Reviewer #4 comment #5

My earlier concerns regarding total-SMAD levels specifically pertain to Fig. 2D. The authors previously stated, in their response to Reviewer #1 during the earlier revision, that they could not draw definitive conclusions. However, following FK506/Rapamycin treatment, the authors observed an increase in phospho-SMAD1/5/9 levels in TGW and SK-N-SH cell lines (as well as in CHP-134 cells treated with Rapamycin), accompanied by an increase in total-SMAD1/5/9 levels. To conclusively establish BMP signaling activation, it is essential to demonstrate that the observed increase in phospho-SMAD1/5/9 levels occurs independently of changes in total-SMAD1/5/9 levels. Without such confirmation, the claim of BMP signaling activation remains speculative.

Response to Reviewer #4 comment #4

We thank the reviewer for the excellent suggestion. As mentioned in the previous rounds of review, these additional Western blots were performed and are now included in Supplementary Figure S12. We also quantitated BMP signaling activity by complementary orthogonal assays (e.g. expression level of the ID family of BMP target genes). Per the recommendation of Reviewer #7 we have now also normalized the pSMAD1/5/9 levels to the total SMAD1/5/9 levels and included this additional quantitation in Table S10. We thank the reviewer for their important comment. We value the opportunity to clarify this important point.

Reviewer #4 comment #5

The authors stated that “the change of SMAD9 signal in our data following RA treatment is quite substantial; the decrease is striking (>10-fold) after 3 days and 6 days of treatment (see, e.g., Fig. S4F).” However, Fig. S5A appears to display a significant increase in total SMAD9 peaks after 3 days of RA treatment. I recommend carefully reviewing the heatmap in Fig. S5A to ensure consistency with the described findings.

Response to Reviewer #4 comment #5

Per the recommendation of Reviewer #7, we value the opportunity to clarify: These figures show different things (examples of SMAD9 peaks at a few specific genes vs SMAD9 peaks globally). It is common to show both of these view of ChIP-seq data. As mentioned in the previous rounds

of review, neither should be used to infer overall BMP signaling activity. We value the opportunity to clarify this important point.

Reviewer #4 comment #6

Finally, the proportions presented in the Venn diagram in Fig. 5B do not sum to 100%. The authors should address this inconsistency or provide additional clarification regarding the methodology used to calculate these proportions.

Response to Reviewer #4 comment #6

As pointed out by Reviewer #7, the proportions in the Venn diagram add up to 99.9% (i.e. there is rounding error). We thank the reviewer for their attention to detail.

In summary, we thank Reviewer #4 for their excellent comments and suggestions and attention to detail, which have dramatically improved the quality of the manuscript.

Reviewer #5 (Remarks to the Author):

Overall, the authors' response is acceptable to the specific comment. However, there are factors to consider when utilizing ATRA vs 13-cisRA in preclinical studies: 1) why only ATRA causes pseudotumor cerebri, 2) as demonstrated, some part of 13-cisRA changes to ATRA in cells when used in in vitro settings, which was not demonstrated in humans yet. The accumulation of 13-cisRA or ATRA in keratinocytes or in sebocytes is anticipated in high concentrations as the side effects of ATRA in skin or glands are well documented, but this cannot be generalized in cancer cells. Either ways, what is it that actually show the biological activity of 13-cisRA in neuroblastoma tumor is yet to be determined, 3) 4-oxo-13-cisRA has comparable biological activity to 13-cisRA in cultured neuroblastoma cells (PMID: 25039756). Does this mean 4-oxo-13-cisRA transforms to 13-cisRA or ATRA intracellularly? This is highly unlikely as the formation of 4-oxo-13-cisRA in humans is the result of enzymatic metabolism.

Response to Reviewer #5

We thank the reviewer for their excellent additional points. We agree that there are still unanswered questions about 13-cis-RA metabolites and their possible activity in neuroblastoma cells, which will form an interesting basis for future research. We have now further expanded on these ideas in our dedicated section "Treatments with 13-cis-RA vs ATRA" where we have now included the Reviewer's additional citation (PMID: 25039756) and further commented on the caveats that the Reviewer has pointed out (lines 1209-1210). We thank the Reviewer for their attention to detail and their expert knowledge of the existing literature, which has substantially improved our manuscript.

Reviewer #6 (Remarks to the Author):

My questions have now been answered by the authors.

Response to Reviewer #6

We thank the Reviewer for their time in carefully reviewing our paper, and their helpful comments, which have substantially improved the quality of the paper.

Reviewer #7 (Remarks to the Author):

With regard to the comments of Reviewer #4 that the authors have already responded to and to the current ones.

Reviewer #7 comment #1

The reviewer states that the authors assert that RA treatment decreases BMP signaling activity. This is not my interpretation of what they are saying with regard to the data in Figure 5 where the authors are looking at 1 day of ATRA treatment. The authors' interpretation is that they see strongly overlapping binding profiles between RARA, SMAD4 and SMAD9 in untreated cells with the genes regulated by these transcription factors being those involved in apoptosis. After 1 day treatment with ATRA they see an increase in RARA binding at a subset of the baseline SMAD4 peaks. It is true that the intensity of the SMAD4 peaks increases in the heatmap which they do not really explain, but the SMAD9 peaks do not change. They do not say that short term treatment with ATRA decreases BMP signaling as Reviewer #4 seems to imply. At longer treatments with ATRA, 90% of the cells die and the cells that are left, show a differentiation phenotype. In this case, levels of PSMAD1/5/9 are substantially lower (Fig S5I) meaning that BMP signaling has been reduced at this later time points. The authors clearly see a decrease in SMAD9 occupancy at the classic BMP target genes ID1 and also TLX2 and SKIL at 3 and 6 days after ATRA treatment.

Response to Reviewer #7 comment #1

This Reviewer's interpretation of our results and conclusions is correct.

Reviewer #7 comment #2

Although the authors don't explain why the SMAD4 peaks increase in intensity after 1 day of ATRA, I do not think that it is necessary for the authors to perform SMAD2/3 and SMAD1/5 ChIP-Seq for the current study. In their most recent comments, the reviewer in still wants the authors to perform these ChIP-seq experiments. It would be interesting to see if TGF- β signalling was upregulated by 1 day of ATRA treatment and the authors could easily investigate this by a simple PSMAD2 and PSMAD3 Western blot, as well as PSMAD1/5/9 Western blot. A ChIP-seq is not necessary in my opinion.

Response to Reviewer #7 comment #2

We agree with Reviewer #7 that it would be interesting to investigate whether TGF- β signaling

was upregulated by 1 day of ATRA treatment. Figure 1 below (obtained from KEGG) shows that pSMAD2/3 (red circles) is downstream of TGF- β (yellow circle), but also downstream of both the Activin (green circle) and Nodal (blue circle) signaling pathways, meaning pSMAD2/3 would provide an ambiguous readout of TGF- β activity. As an alternative (and similarly to an approach we used to quantify BMP signaling activity (Figure 2 below)) we have calculated the expression of the canonical TGF- β signaling target genes (orange circle, Figure 3). Those results indicate little change in TGF- β signaling activity in ATRA-treated CHP-134 cells over 6 days (Figure 3), in contrast to a substantial loss in BMP signaling at 6 days (Figure 2; this is consistent with all of our other data shown in the paper). The only gene representing an exception is RBL1 (a.k.a. p107), which is also cell cycle responsive. We have now included the expression of the TGF- β target genes in Supplementary Figure S5I (now referenced in main text line 469). We thank the Reviewer for their additional input, and we hope that these results have further clarified our paper's overall conclusions.

Reviewer #7 comment #3

The reviewer points out that SMAD1/5/9 should be used as a control in the Western blots and not just GAPDH. It is true that it is important to check that the PSMAD1/5/9 does not just increase/decrease because of changes in SMAD1/5/9 levels. They have done this for the blots that the reviewer refers to. I think it would be important to ask the authors to quantitate the blots and normalise the PSMAD1/5/9 levels to the SMAD1/5/9 levels. The GAPDH levels are also important as they indicate that the same amount of extract was loaded in each lane, assuming that they are the result of a reprobe of the same membrane. The values can also be normalised to the GAPDH levels.

Response Reviewer #7 comment #3

We thank the reviewer for the additional input. We have now included these results quantitating the relative levels of phosph-SMAD1/5/9 and total SMAD1/5/9 (Table S10). The results do not change our core conclusions, which depend on the relative change of phosph-SMAD1/5/9 over the course of treatment with retinoic acid. We thank the Reviewer for the helpful suggestion which has improved the rigor of our paper.

Reviewer #7 comment #4

With regard to the issue of the compatibility of the data for SMAD9 in Figures S4 and S5, I think the only thing that needs clarifying is that the >10 fold decrease in SMAD9 at day 3 and 6 is at particular genes, whilst the slight increase in SMAD9 occupancy at day 3 is on all targets. I don't think there is any discrepancy in the manuscript concerning this point.

Response to Reviewer #7 comment #4

We agree with the Reviewer that these data are consistent with our paper's overall conclusions.

Reviewer #7 comment #5

The reviewer says that the proportions presented in the Venn diagram in Fig. 5B do not sum to 100%. According to my calculation, it adds up to 99.9%, so I think it is just a result of rounding up/down. They have presented the data as a percentage of total peaks (RARA, SMAD4 and SMAD9) in each case.

Response to Reviewer #7 comment #5

We agree with Reviewer #7.

Reviewer #7 comment #6

There is one further comment that I will make about the BMP side of the paper. The authors treat the cells with BMP1, but it should be noted that this is not a BMP ligand, but rather a protease. Thus, it is not surprising that it was inactive.

(see https://en.wikipedia.org/wiki/Bone_morphogenetic_protein_1)

Response to Reviewer #7 comment #6

We thank the reviewer, this is an excellent catch. We did not understand why BMP1 was inactive in this assay but included these data anyway. We thank the Reviewer for explaining this result. We have now altered our presentation of these data, which better summarizes the result, given that BMP1 is not a BMP ligand (lines 621-622).

Overall, we thank Reviewer #7 for their rational and fair approach to the review of our paper and we appreciate their ability to comprehend the large collective body of data we have presented. We hope they appreciate that our overall conclusions are consistent with the data we have shown.

Sincerely,

Paul Geeleher, PhD
Associate Professor
Comprehensive Cancer Center
Dept. of Comp Biology
School of Biomedical Sciences
St Jude Children's Research Hospital,
Memphis TN, USA
paul.geeleher@stjude.org

Min Pan, PhD
Comprehensive Cancer Center
Dept. of Comp Biology
School of Biomedical Sciences
St Jude Children's Research Hospital,
Memphis TN, USA
Min.pan@stjude.org

John Easton, PhD
Director, Genomics Laboratory,
Department of Comp Biology
St Jude Children's Research Hospital,
Memphis TN, USA
John.easton@stjude.org

Dear Peer Reviewers,

We thank the reviewers for their detailed assessment of the paper and we have addressed the final comments. We thank the Reviewers for their assessment of the paper and we hope the final manuscript is much improved.

REVIEWER COMMENTS

Reviewer #7 (Remarks to the Author):

The authors have now successfully addressed all the points I raised and I have no further comments on the manuscript.

Response to Reviewer #7:

We are glad we have addressed all of Reviewer #7's concerns. Overall, we thank Reviewer #7 for their rational and fair approach to the review of our paper.

Sincerely,

Paul Geeleher, PhD
Associate Professor
Comprehensive Cancer Center
Dept. of Comp Biology
School of Biomedical Sciences
St Jude Children's Research Hospital,
Memphis TN, USA
paul.geeleher@stjude.org

Min Pan, PhD
Comprehensive Cancer Center
Dept. of Comp Biology
School of Biomedical Sciences
St Jude Children's Research Hospital,
Memphis TN, USA
Min.pan@stjude.org

John Easton, PhD
Director, Genomics Laboratory,
Department of Comp Biology
St Jude Children's Research Hospital,
Memphis TN, USA
John.easton@stjude.org